# Porosity and Permeability Prediction through Forward Stratigraphic Simulations Using GPM™ and Petrel™: Application in Shallow Marine Depositional Settings.

Daniel Otoo and David Hodgetts

Department of Earth and Environmental Sciences, University of Manchester, Manchester, M13 9PL, United Kingdom.

*Correspondence to*: Daniel Otoo (daniel.otoo@manchester.ac.uk)

**Abstract**

The forward stratigraphic simulation approach is applied to predict porosity and permeability distribution. Synthetic well logs from the forward stratigraphic model served as secondary data to control porosity and permeability representation in the reservoir model. Building a reservoir model that fits data at different locations comes with high levels of uncertainty. Therefore, it is critical to generate an appropriate stratigraphic framework to guide lithofacies and associated porosity/permeability simulation. The workflow adopted for this task is in three parts; first, simulation of twenty scenarios of sediment transportation and deposition using the geological process modeling (GPM™) software developed by Schlumberger. Secondly, an estimation of the extent and proportion of lithofacies units in the stratigraphic model using the property calculator tool in Petrel™. Finally, porosity and permeability values are assigned to corresponding lithofacies-units in the forward stratigraphic model to produce a forward stratigraphic-based porosity and permeability model. Results show a forward stratigraphic-based lithofacies model, which depends on sediment diffusion rate, sea level variation, sediment movement, wave processes, and tectonic events. This observation is consistent with the natural occurrence, where variation in sea level, sediment supply, and accommodation control stratigraphic sequences, and therefore, facies distribution in a geological basin. Validation wells VP1 and VP2 showed a notable match after a comparing the original and forward stratigraphic-based porosity models. However, a significant discrepancy is recorded in the permeability estimates. These results suggest that the forward stratigraphic modeling approach can be a practical addition to geostatistical-based workflows for realistic prediction of porosity and permeability.

# 1    Introduction

The distribution of reservoir properties such as porosity and permeability is a direct function of a complex

combination of sedimentary, geochemical, and mechanical processes (Skalinski & Kenter, 2014). The

impact of reservoir petrophysics on well planning and production strategies makes it imperative to use

reservoir modeling techniques that present realistic property variations via 3-D models (Deutsch and

Journel, 1999; Caers and Zhang, 2004; Hu & Chugunova, 2008). Typically, reservoir modeling requires

continued property modification until an appropriate match to subsurface data. Meanwhile, subsurface

data acquisition is expensive, thus restricts data collection and accurate subsurface property modeling.

Several studies, Hodgetts et al. (2004) and Orellana et al. (2014) have demonstrated how stratigraphic

patterns, and therefore petrophysical attributes in seismic data, outcrops, and well logs are applicable in

subsurface modeling. However, the absence of detailed 3-dimensional depositional frameworks to guide

property modeling inhibits the use of stratigraphic patterns to capture subsurface property variations

(Burges et al. 2008). Reservoir modeling techniques with the capacity to integrate forward stratigraphic

simulation outputs with stochastic modeling techniques for subsurface property modeling will improve

reservoir heterogeneity characterization, because they more accurately produce geological realism than

the other modeling methods (Singh et al. 2013). The use of geostatistical-based methods to represent

spatial variability of reservoir properties has been in many exploration and production projects (Kelkar

and Godofredo, 2002). In the geostatistical modeling method, an alternate numerical 3-D model

(realizations) shows different property distribution scenarios that are most likely to match well data

(Ringrose & Bentley, 2015). However, due to cost, reservoir modeling practitioners continue to encounter

the challenge of obtaining adequate subsurface data to deduce reliable variograms for geostatistical-based

subsurface modeling, therefore introducing a significant level of uncertainty in reservoir models (Orellena

et al. 2014). The advantages of applying geostatistical modeling approaches to represent subsurface

properties in models are discussed in studies by Deutsch and Journel (1999), Dubrule, (1998). A notable

disadvantage is that the geostatistical modeling method tends to confine reservoir property distribution to

subsurface data and rarely produces geological realism to capture sedimentary events that led to reservoir

formation (Hassanpour et al. 2013). In effect, the geostatistical modeling technique does not reproduce long-range continuous reservoir properties, which are essential for generating realistic reservoir connectivity models (Strebelle & Levy, 2008). In their work, Christ et al. (2016) illustrate the use of forward stratigraphic modeling for reconstructing subsurface patterns. The forward stratigraphic modeling method operates on the guiding principle that multiple sedimentary process simulations in a 3-D framework will provide geologic details to improve the modeling of stratigraphic sequences, and therefore facies and petrophysical property distribution in an existing basin model. Given this, the forward stratigraphic simulation approach was applied in this contribution to forecast lithofacies, porosity, and permeability in a reservoir model. A significant aspect of this work is using variogram parameters from forward stratigraphic-based synthetic wells to simulate porosity and permeability trends in the reservoir model.

The geological process modeling GPM$^{TM}$ software (Schlumberger, 2017) is used to replicate sediment depositional processes in the model area to realize realistic stratigraphic sequences for porosity and permeability prediction. The reservoir interval understudy is within the Hugin formation. Studies by Varadi et al. (1998); Kieft et al. (2011) indicate that the Hugin formation was formed through a complex depositional architecture of waves, tidal, and fluvial processes. This knowledge suggests that a single depositional model will not be adequate to produce a realistic lithofacies or petrophysical distributions model of the area. Furthermore, the indication of a complicated Syn-depositional rift-related faulting system by Milner and Olsen, 1998, significantly influences the stratigraphic architecture of the model area. Therefore, the contribution seeks to produce a depositional sequence, which captures subsurface attributes observed in seismic and well data to guide porosity and permeability modeling.

**Study Area**

The Volve field (Figure 1), located in Block 15/9 south of the Norwegian North Sea, has the Hugin Formation as the reservoir interval from which hydrocarbons are produced (Vollset and Dore, 1984). The Hugin formation, which is Jurassic in age (late Bajocian to Oxfordian), is made up of shallow marine to marginal marine sandstone deposits, coals, and a significant influence of wave events that tend to control

lithofacies distribution in the formation (Varadi et al. 1998; and Kieft et al. 2011). Studies by Sneider et al. (1995) and Husmo et al. (2003) associate sediment deposition into the study area to rift-related subsidence and successive flooding during a large transgression of the Viking Graben within the Middle to Late Jurassic period. Also, Cockings et al. (1992), Milner and Olsen (1998) indicate that the Hugin formation comprises of marine shoreface, lagoonal and associated coastal plain, back-stepping delta-plain, and delta front. However, recent studies by Folkestad and Satur (2006) also provide evidence of a high tidal event, which introduces another dimension that requires attention in any subsurface modeling task in the study area. The thickness of the Hugin formation is estimated between 5 m and 200 m, but can be thicker off-structure and non-existent on structurally high segments due to post-depositional erosion (Folkestad and Satur, 2006).

A summarised sedimentological delineation within the Hugin formation is presented based on studies by Kieft et al. (2011). In **Table 1,** lithofacies-association codes A, B, C, D, and E represent bay fill units, shoreface sandstone facies, mouth bar units, fluvio-tidal channel fill sediments, and coastal plain facies units, respectively. Additionally, a lithofacies association prefixed code F, which consists of open marine shale units, mudstone. Within it are occasional siltstone beds, parallel laminated soft sediment deformation that locally develop at bed tops. The lateral extent of the code F lithofacies package in the Hugin formation is estimated to be 1.7 km to 37.6 km, but the total thickness of code F lithofacies is not known (Folkestad & Satur, 2006).

# Data and Software

This work is based on the description and interpretation of petrophysical datasets in the Volve field by Equinor. Datasets include 3-D seismic and a suite of 24 wells that consist of formation pressure data, core data, petrophysical and sedimentological logs. Previous studies by Folkestad & Satur (2006) and Kieft et al., (2011) in this reservoir interval show varying grain size, sorting, sedimentary structures, bounding contacts of sediment matrix. Grain size, sediment matrix, and the degree of sorting will typically drive the volume of the void created, and therefore the porosity and permeability attributes. Wireline-log attributes such as gamma-ray (GR), sonic (DT), density (RHOB), and neutron-porosity (NPHI)

distinguish lithofacies units, stratigraphic horizons, and zones that are essential for building the 3-D

property model in Schlumberger's Petrel[TM] software. Besides, this study also seeks to produce a realistic

depositional model like the natural stratigraphic framework in a shallow marine depositional setting.

Therefore, obtaining a 3-dimensional stratigraphic model that shows a similar stratigraphic sequence

observed in the seismic data allows us to deduce variogram parameters to serve as input in actual

subsurface property modeling.

Twenty forward stratigraphic simulations were produced in the geological process modeling (GPM[TM])

software to illustrate depositional processes that resulted in the build-up of the reservoir interval under

study. By the fourth simulation, there was a development of stratigraphic patterns that shows similar

sequences as those observed in seismic, hence the decision to constrain the simulation to twenty scenarios.

Delft3D-Flow[TM] and DIONISOS[TM] are examples of subsurface process modeling software used in

previous studies such as Rijin & Walstra, (2003) and Burges et al. (2008). The availability of the GPM[TM]

software license and the capacity to integrate stratigraphic simulation outputs in the property modeling

workflow in Petrel[TM] is the reason for using the geological process modeling software in this study.

## Methodology

The workflow (Figure 2a) combines the stratigraphic simulation capacity of GPM[TM] in different

sedimentary processes and the property modeling tools in Petrel[TM] to predict the distribution of porosity

and permeability properties away from known data. This involves three broad steps: (i) forward

stratigraphic simulation in GPM[TM] (2019.1 version), (ii) lithofacies classification using the calculator tool

in Petrel[TM], and (iii) porosity and permeability modeling in Petrel[TM] (2019.1 version).

## Forward Stratigraphic Simulation in GPM[TM]

GPM[TM] is commercial software developed by Schlumberger to simulate clastic and carbonate

sedimentation in a deep or shallow marine environment. GPM[TM] consists of geological processes such as

steady flow, sediment diffusion, tectonics, and sediment accumulation that rely on physical equations and

assumptions to replicate the process of sedimentation in a geological basin. A realistic realization of a

stratigraphic pattern as observed in seismic or well data provides a 3-dimensional framework to constrain subsurface property representation that conforms with the real-world property distribution trends. In clastic sedimentation, the movement of sediments relies on equations from the original SEDSIM developed in Stanford University (Harbaugh, 1993). Sediment movement, erosion, and deposition is governed by a simplified Navier Stokes equation. "Simplified" because the Navier-Stokes equation in its original form define sediment movement in a 3-dimensions differential form, while the flow equation in GPM$^{TM}$ is 2-dimensional with an arbitrary input of flow depth. Kieft et al. (2011) describe the influence of a combination of fluvial and wave processes in the genetic structure of sediments in the Hugin formation. These geological processes are rapid, depending on accommodation generated by sea-level variation and or sediment composition and flow intensity. The deposition of sediments into a geological basin and its response to post-depositional sedimentary or tectonic processes are significant in the ultimate distribution of subsurface lithofacies and petrophysics. Therefore, several input parameters for the forward simulation to attain a stratigraphic output that fits existing knowledge of paleo-sediment transportation and deposition into the study area (see Table 2). The forward simulation at all stages portrayed geological realism concerning stratigraphic sequence, but it also revealed some limitations, such as instability in the simulator when more than three geological processes run concurrently. Given this, the diffusion and tectonic processes remained constant whiles varying the steady flow, unsteady flow, and sediment accumulation processes in each simulation run.

**Steady & Unsteady Flow Process**

The steady flow process in GPM simulates flows that change slowly over a period, or sediment transport scenarios where flow velocity and channel depth do not vary abruptly e.g. rivers at a normal stage, deltas, and sea currents. Considering the influence of fluvial activities during sedimentation in the Hugin formation, it is significant to capture its impact on the resultant simulated output.

The unsteady flow process can simulate periodic flows such as turbidites where the occurrence is not regular, and the velocity of flow changes abruptly over time. The unsteady flow process applies several fluid elements driven by gravity and friction against the hypothetical topographic surface. Otoo and

Hodgetts (2019) illustrate how the unsteady process in GPM$^{TM}$ attains realistic distribution of lithofacies

units in a turbidite fan system. Although the steady and unsteady flow governing equations distantly rely

on the Navier-Stokes equations, the steady flow is quite distinct, as it uses a finite difference numerical

method for faster computation and to also illustrate the frequency of flow that is characteristic in channel

flow such as rivers. The finite difference method applies an assumption that flow velocity is constant

from channel bottom to surface. In contrast, the unsteady flow uses the particle method from SEDSIM3

to solve the sediment concentration in flow and sediment transport capacity (Tetzlaff & Harbaugh 1989).

The simplified equation in GPM$^{TM}$ attempts to solve the problem of "shallow-water free-surface flow"

over an arbitrary topography surface (Tetzlaff, D. personal communication, February 2021). "Shallow

water" indicates the instance where only the vertically-averaged flow velocity and flow depth are applied

and kept track of as a function of two horizontal coordinates.

The equation that control steady and unsteady flow is expressed through:

$$\frac{\partial h}{\partial t} + \nabla . hQ = 0 \qquad (1)$$

Where: h is flow depth, t is time, and Q the horizontal flow velocity vector.

$$(\frac{\partial Q}{\partial t} = -(g\nabla)H + \frac{c_2}{\rho}\nabla^2 Q - \frac{c_2 Q/Q/}{h} \qquad (2)$$

Where: $\frac{\partial Q}{\partial t}$ is the Lagrangian derivative of flow relative to time, g is gravity, H is the water surface

elevation, $c_2$ is the fluid friction coefficient, $\rho$ is the water density, $c_1$ is the water friction coefficient and

147    h is the flow depth.

The Manning's equation is applied to relate flow, slope, flow depth and hydraulic radius channels with a

constant cross-section for the steady flow process. Manning's formula states:

$$V = \frac{k}{n} R_h{}^{2/3} S^{1/2} \qquad (3)$$

Where: V is the flow velocity, k is the unit conversion factor, n is the Manning's coefficient which

depends on channel rugosity, $R_h$ is the hydraulic radius and S is the slope.

As mentioned earlier, the unsteady flow process uses the particle method equation, which relies on the

assumption that erosion and deposition depend on the balance between the flow's transport capacity and

the "effective sediment concentration". The equation for multiple-sediment transport in flow is given as

follows:

$$A_{em} = \sum_{k_s} \frac{l_{Ks}}{f_{1k_s}} \qquad (4)$$

Where: $A_{em}$ is the effective sediment concentration of mixture, $l_{ks}$ is the sediment concentration of each

type, and $f_1,k_s$ is the transportability of each sediment type.

The transport capacity of a sediment type is expressed by equations (5) and (6). Let consider

$$R = (A - A_{em})f_2,k_s \qquad (5)$$

Where $f_2,k_s$ is the erosion-deposition rate coefficient for sediment type $k_s$. For every sediment type ,$k_s$,

the formula for transporting sediment of different grain sizes is given as:

$$(H - Z)\frac{Dl_{Ks}}{Dt} = \begin{cases} R & if\ R > 0\ and\ \tau_0\ \geq f_{3,k_s}\ and\ k(x,y,z) =\ K_s \\ & or\ R < 0\ and\ K_s = 1\ or\ l_{k_s-1} = 0 \\ 0 & otherwise \end{cases} \qquad (6)$$

Where;

H is the free surface elevation to sea level, Z is the topographic elevation for sea level, $K_s$ is the sediment

type, $l_{ks,}$ is the volumetric sediment concentration of a specific type (k).

**Sediment Diffusion Process**

The diffusion process replicates sediment movement from a higher slope (source location) and deposition

into a lower elevation of the model area. Sediment diffusion runs on the assumption that sediments are

transported downslope at a proportional rate to the topographic gradient, making fine-grained sediments

easily transportable than coarse-grained sediments. Sediment diffusion depends on three parameters: (i)

sediment grain size and turbulence in the flow, (ii) diffusion curve that serves as a unitless multiplier in

the algorithm and, (iii) diffusion coefficient. The diffusion coefficient depends, among other variables on

the type of sediment and "energy" of the depositional environment. In this contribution, the highest depth-dependent diffusion coefficient occurs near sea level, where the "energy" is highest over a geological time (Dashtgard et al. 2007).

In GPM$^{TM}$, sediment diffusion is calculated using a simplified expression:

$$\frac{\partial z}{\partial t} = D_i \nabla^2 z + S_n \qquad (7)$$

where **z** is topographic elevation, $D_i$ is the diffusion coefficient, **t** for time, and $\nabla^2 z$ is the laplacian of z, and $S_n$ is the sediment source term.

Sediment diffusion ($D_i$) is estimated by assuming that the grain size for each sediment component (coarse sand, fine sand, silt, and clay) are known. Also an assumption that these sediment types have a uniform diameter (D) in the flow mix (Dade & Friend 1998; and Zhong 2011). In that case, external fore ($F_e$), which consist of drag, lift, virtual mass, and Basset history force is given as:

$$F_e = \alpha_e M_e + \alpha_e \Phi_D . \frac{U_{fi} - U_{ei}}{T_p} \qquad (8)$$

$M_e$ is the resultant force of other forces with the exception of drag force, $T_p$ stokes relation time, expressed as: $T_p = \rho_\rho D^2/(18\rho_f V_f)$, with $\rho_f$ and $V_f$ as density and viscosity of fluid respectively. $\Phi_D$ is a coefficient that accounts for the non-linear dependence of drag force on grain slip Reynolds number ($R_p$).

$$\Phi_D = \frac{R_p}{24} C_D \qquad (9), \text{ with } C_D \text{ sediment grain coefficient.}$$

With the flow component in place, the diffusion coefficient ($D_i$) is deduced from the Einstein equation. Using an assumption that the diffusion coefficient decreases with increasing grain size and rise in temperature, and that the coefficient f is known, the expression for $D_i$ is:

$$D_i = \frac{K_B . T}{f} \qquad (10)$$

Meanwhile, f is a function of the dimension of the spherical particle involved at a particular time (t). In accounting for f, the equation for $D_i$ changes into:

$$D_i = \frac{K_B.T}{6.\pi.\eta_o.r} \qquad (11)$$

## Sediment Accumulation

The sediment accumulation process in GPM is designed to generate an arbitrary amount of sediment

representing the artificial vertical thickness of a lithology as interpreted in a well or outcrop data (Tetzlaff,

D., personal communication, February 2021). The areal input rates for each sediment type (coarse-

grained, fine-grained sediments) use the value of the map surface at each cell in the model and multiply

it by a value from a unitless curve at each time step in the simulation to estimate the thickness of sediments

accumulated or eroded from a cell in the model. Sediment accumulation in the GPM software requires

other processes such as steady flow and diffusion to account for sediment transport (sediment entering or

leaving a cell) before a deposition/year (mm/yr) function to artificially produce the height of sediment

deposited per cell. The accumulation of sediments in GPM is expressed as:

$$A_T = \sum_{S=1}^{n} \left[ (M_{v1} * S_{c1}), \_\_ n \right] \qquad (12)$$

Where;

$A_T$ is the total sediment accumulated in a cell over a period, S is the sediment type, $M_v$ is the map value

of sediment in each cell, and $S_C$ is the sediment supply curve as a function of topographic elevation.

## Boundary Conditions for Forward Stratigraphic Simulation

Realistic reproduction of stratigraphic patterns in the model area requires input parameters (initial

conditions), such as paleo-topography, sea-level curves, sediment source location, and distribution curve,

tectonic event maps (subsidence and uplift), and sediment mix velocity. The application of these input

parameters in GPM$^{TM}$ and their impact on the resultant stratigraphic framework is below.

**Hypothetical Paleo-Surface:** The hypothetical paleo-topographic for the stratigraphic simulation is from the

seismic data (Figure 3), using the  assumption that the present day stratigraphic surface (paleo shoreline in Figure

4a) occurred as a result of basin filling over geological time. Since the surface obtained from the seismic section

have undergone various phases of subsidence and uplifts, it is significant to note that the paleo topographic surface

used in this work does not represent an accurate description of the basin at the period of sediment deposition; thus

presenting another level of uncertainty in the simulation. To derive an appropriate paleo-topographic for this task, five paleo topographic surfaces (TPr) were generated, by adding or subtracting elevations from the inferred paleo topographic surface (see Figure 4g) using the equation:

$$TPr = Sbs + EM \qquad (13)$$

where, Sbs is the base surface scenario (in this instance, scenario 6), and EM an elevation below and above the base surface.

The paleo-topographic surface in scenario 3 (figure 4d) is selected because it produced a stratigraphic sequences that fit the depositional patterns interpreted from the seismic section (Figure 5d).

**Sediment Source Location:** Based on regional well correlations in Kieft et al. 2011, and seismic interpretation of the basin structure, the sediment entry point is placed in the north-eastern section of the hypothetical paleo-topography surface. The exact sediment entry point into this basin is unknown, so three entry points were placed at a 4 km radius around the primary location (Figure 3c) to capture possible sediment source locations in the model area. The source position is a  positive integer (values greater than zero) to enable sediment movement to other parts of the topographic surface.

**Sea Level:** The sea-level curve is deduced from published studies and facies description in shallow marine depositional environments (e.g. Winterer and Bosellini, 1981). To sea level was constrained 30 m for short simulation runs (5000 to 20000 years), but varied with the increasing duration of the simulation (see Table 2). The peak sea-level in the simulation depicts the maximum flooding surface (Figure 5d), and therefore the inferred sequence boundary in the geological process model.

**Diffusion and Tectonic Event Rates:** The sediment mix proportion, diffusion rate, and tectonic event functions are from studies such as Walter, (1978), Winterer and Bosellini, (1981), and Burges et al., (2008). The diffusion and tectonic event rates were increased or reduced to produce a stratigraphic model that fit our knowledge of basin evolution in the study area. For example, in scenario 1 (Figure 6a), the early stages of clinoform development show resemblance to interpreted trends in the seismic section (**Figure 3b**). The process commenced with a diffusion coefficient of 8 m2/a, but it varied at each scenario to obtain diffusion coefficients to improve the model. Excluding the initial topography (Figure 4d), input

parameters in geological processes such as wave events, steady/unsteady flow, diffusion, and tectonic

events used curve functions to provide variations in the simulation.

The sensitivity of input parameters in the forward stratigraphic simulation is notable when there is a

change of value in sediment diffusion, and tectonic rates or dimension of the hypothetical topography.

For example, a change in sediment source position affects the extent and depth of sediments deposition

in the simulation. Shifting the source point to the mid-section of the topography (the mid-point of the

topography in a basin-ward direction) resulted in the accumulation of distal elements identical to turbidite

lobe systems. This output is consistent with morphodynamic experiments by de Leeuw et al. (2016),

where sediment discharge from the basin slope leads to the build-up of basin floor fan units.

## Property Classification in Stratigraphic Model

In our opinion, the most appropriate output is the stratigraphic model in **Figure 5d**. This point of view is

because, compared to the depositional description in studies such as Folkestad and Satur (2006); Kieft et

al. (2011), and the seismic interpretation presents a similar stratigraphic sequence. Sediment distribution

in each time step of the simulation was stacked into a single zone framework to attain a simplified model.

This strategy assumes that sedimentary processes that lead to the final build-up of genetic related units

within zones of the model will not vary significantly over the simulation period. The stratigraphic model

(**Figure 5d**) was converted into a 3-D format (20 m x 20 m x 2 m grid cells) for the property modeling in

Petrel$^{TM}$.

Facies, porosity, and permeability representation in the stratigraphic model was done via a rule based

approach in Petrel$^{TM}$ (see **Table 3**). The classification is driven by depositional depth, geologic flow

velocity, and sediment distribution patterns as indicated in **Figure 7**. Lithofacies representation in the

stratigraphic model relied on the sediment grain size pattern and proximity to sediment source. For

example, shoreface lithofacies units are medium-to-coarse grained sediments, which accumulate at a

proximal distance to the sediment source. In contrast, mudstone units are confined to fine-grained

sediments in the distal section of the simulation domain.

Using knowledge from published studies by Kieft et al. (2011) and wireline-log attributes such as gamma ray, neutron, sonic, and density logs, porosity and permeability variations in the stratigraphic model are estimated (Table 1). In previous studies on the Sleipner Øst, and Volve field (Equinor, 2006; Kieft et al. 2011), shoreface deposits make up the best reservoir units, whiles lagoonal deposits formed the worst reservoir units. With this guide, shoreface sandstone units and mudstone/shale units in the forward stratigraphic model are best and worst reservoir units respectively. The porosity and permeability values in Table 4 are from equations in Statoil's petrophysical report of the Volve field (Equinor, 2016):

$$\emptyset_{er} = \emptyset_D + \alpha \cdot (NPHI - \emptyset_D) + \beta \qquad (14)$$

where $\emptyset_{er}$ is the estimated porosity range, $\emptyset_D$ is density porosity, $\alpha$ and $\beta$ are regression constants; ranging between -0.02 – 0.01 and 0.28 – 0.4 respectively, $NPHI$ is neutron porosity. In instances where NPHI values for lithofacies units is not available from the published references, an average of 0.25 was used.

$$KLOGH_{er} = 10^{(2 + 8 * PHIF - 5 * VSH)} \qquad (15)$$

where $KLOGH_{er}$ is the estimated permeability range, $VSH$ is the volume of clay/shale in the lithofacies unit, and $PHIF$, the fractured porosity. The $VSH$ range between 0.01 – 0.12 for the shoreface units, and 0.78 – 0.88 for lagoonal deposits.

## Property Modeling in Petrel$^{TM}$

The workflow (**Figure 2b**) used for subsurface property modeling in Petrel$^{TM}$ is applied to represent lithofacies, porosity, and permeability properties in the stratigraphic model. These processes involve:

(1) Structure modeling: identified faults within the study area are modeled together with interpreted surfaces from seismic and well correlation to generate the main structural framework, within which the property model is built. Here, fault pillars and connecting fault bodies are linked to obtain the kind of fault framework interpreted from the seismic data.

(2) Pillar gridding: building a "grid skeleton" made up of a top, middle and base architectures. Typically, pillars join corresponding corners of every grid cell of the adjacent grid to form the

foundation for each cell within the model. The prominent orientation of faults (I-direction) within the model area was in an N-S and NE-SW direction, so the "I-direction" was set to NNE-SSW to capture the general structural description of the area.

(3) Horizons, Zones, and Vertical Layering: stratigraphic horizons and subdivisions (zones) delineate the geological formation's boundaries. As stratigraphic horizons are introduced into the model grid, the surfaces are trimmed iteratively and modified along faults to correspond with displacements across multiple faults. Vertical layering shows the thicknesses and orientation between the layers of the model. Layers refers to significant changes in particle size or sediment composition in a geological formation. Using a vertical layering scheme makes it possible to honor the fault framework, pillar grid, and horizons. A constant cell thickness of 1 m is used in the model to control the vertical scale of lithofacies, porosity, and permeability modeling.

(4) Upscaling: involves the substitution of smaller grid cells with coarser grid cells. Here, log data is transformed from 1-dimensional to a 3-dimensional framework to evaluate which discrete value suits selected data point in the model. One advantage of the upscaling procedure is to make the modeling process faster.

**Porosity and Permeability Modeling**

The Volve field petrophysical model from Equinor is the base (reference) model in this work. The model, which covers 17.9 km$^2$ was generated with the reservoir management software (RMS), developed by Irap and Roxar (Emerson$^{TM}$). The petrophysical model has a grid dimension of 108 m x 100 m x 63 m and was compressed by 75.27% of cell size from an approximated cell size of 143 m x 133 m x 84 m. To achieve a comparable model resolution as the Volve field porosity and permeability model, the forward stratigraphic output, which had an initial resolution of 90 m x 78 m x 45 m, is upscaled to a grid of 107 m x 99 m x 63 m. Variograms being a critical aspect of this work, we submit two options to extrapolate variogram parameters from the forward stratigraphic-based porosity and permeability models. In Option 1, the porosity and permeability values were assigned to the synthetic lithofacies wells that correlate with known facies-association in the study area (see **Table 4)**.

The pseudo wells comprising porosity and permeability are situated in-between well locations to guide porosity and permeability simulation in the model. For option 2, the best-fit forward stratigraphic model changes by assigning porosity and permeability attribute using the general stratigraphic orientation captured in the seismic data (NE-SW; 240º). Porosity and permeability pseudo (synthetic) logs were then extracted from the forward stratigraphic output to build the porosity and permeability models (**Figure 8**). Porosity modeling is through normal distribution, whiles the permeability models were produced using a log-normal distribution and the corresponding porosity property for collocated co-kriging.

Considering that vertical trends in options 1 and 2 will be similar within a sampled interval, option 2 presented a viable 3-D representation of property variations in the major and minor directions of the forward stratigraphic model. Ten synthetic wells (SW), ranging between 80 m and 120 m in total depth (TD), are positioned in the forward model to capture the vertical distribution of porosity-permeability at different sections of the forward stratigraphic-based models.

The synthetic wells (**Figure 9 c**) with porosity and permeability data were upscaled, and distributed into the original structural model using the sequential Gaussian simulation method. The synthetic wells derived from the stratigraphic model served as an additional control for porosity and permeability modeling in the Volve field. Because the variogram-based modeling approach is efficient in subsurface data conditioning, this idea presents an opportunity to get more wells at no additional cost to control porosity and permeability distribution. The variogram model (**Figure 10**) of dominant lithofacies units in the stratigraphic model served as a guide in estimating variogram parameters for porosity and permeability modeling. The variogram has major and minor range of 1400 m and 400 m respectively, and an average sill value of 0.75. Six out of fifty model realizations that show some similarity to the original porosity and permeability model formed the basis of our analysis (**Figure 11**). The selection of six realizations was on a visual and statistical comparison of zones in the original Volve field model and the stratigraphic-based porosity/permeability model. The statistical approach involved summary statistics from the reference model and the stratigraphic-based porosity/permeability model. In contrast, the visual evaluation compared the geological realism of forward stratigraphic-based realizations to the base model.

# Results

The stratigraphic model in stage 4 (**Figure 5d iv**) shows the final geometry after 700,000 years of simulation time. The initial stratigraphic simulation produced a progradation sequence with foreset-like features (**Figure 5d i**) and a sequence boundary, which separates the initial simulated output from the next prograding phase (**Figure 5d ii**). An aggradational stacking pattern commences and becomes prominent in stage 3 (**Figure 5d iii**). These aggradational sequences observed in the forward stratigraphic model are consistent with natural events where sediment supply matchup with accommodation due to sea-level rise within a geological period (Muto and Steel, 2000; Neal and Abreu, 2009).

Impact of the forward stratigraphic simulation on porosity and permeability representation in the reservoir model is evident by comparing its outcomes to the Volve field porosity and permeability models by using two synthetic well (VP1 and VP2); sampled at a 5 m vertical interval. Taking into account the fact that the Volve field petrophysical model (**Figure 11a**) went through various phases of history matching to obtain a model to improve well planning and production strategies, it is reasonable to assume that porosity and permeability distribution in the petrophysical model will be geologically realistic and less uncertain. This view formed the basis for using the porosity and permeability models developed by Equinor as a reference for comparing outputs in the stratigraphic model. **Table 5a** shows an almost good match in porosity at different intervals in the forward stratigraphic-based models (i.e. R14, R20, R26, R36, R45, and R49). An analysis of the well logs in the model area shows that a large proportion of reservoir porosity is between $0.18 - 0.24$. Also, the analysis of the forward stratigraphic-based porosity model is consistent with the porosity range in the Volve field model (see Figure 12).

A notable limitation with this approach is the assumption that variogram parameters and stratigraphic inclination within zones remained constant throughout the simulation. The difference in permeability attributes between the original permeability model and the forward stratigraphic-based type is the application of other measured parameters in the original model (**Table 5b**). Typically, a petrophysical model like the Sleipner Øst and Volve field model will factor in other datasets such as special core analysis (SCAL) and level of cementation, which enhances reservoir petrophysics assessment. Bearing in mind

that the forward stratigraphic model did not involve some of this additional information from the

reservoir, it is practicable to suggest that results obtained in the forward stratigraphic-based porosity and

permeability models have adequately conditioned to known subsurface data.

## Discussion

Results show the influence of sediment transport rate (or diffusion rate), initial basin topography, and

sediment source location on the stratigraphic simulation in in GPM$^{TM}$. Compared to studies such as Muto

& Steel (2000) and Neal & Abreu (2009), we observed that a variation in sea-level controls the volume

of sediment that is retained or transported further into the basin, therefore controlling the resultant

stratigraphic sequences. In related work, Burges et al. (2008) suggest that a sediment-wedge topset width

connects directly to the initial bathymetry, in which the sediment-wedge structure develops, and the

correlation between sediment supply and accommodation rate. This opinion is in line with observations

in this study, where the initial sediment deposit controls the geometry of subsequent phases of depositions

in the hypothetical basin. The uncertainty of initial conditions used in this work led to the generation of

multiple forward stratigraphic scenarios to account for the range of bathymetries that may have influenced

sediment transportation to form the present-day reservoir units in the Volve field.

The simulation produced well-defined sloping depositional surfaces in a stratigraphic architecture

(clinoforms) and sequence boundaries that depict patterns seen in the seismic data. In their work, Allen

and Posamentier (1993); Ghandour and Haredy (2019) explained the importance of sequence stratigraphy

in lithofacies characterization, and therefore petrophysical property distribution in sedimentary systems.

Also, sediment deposition into a geological basin in the natural order is controlled by mechanical and

geochemical processes that modify petrophysical attributes (Warrlich et al. 2010); therefore, using

different geological processes and initial conditions to generate depositional scenarios in 3-dimension

provides a framework to analyse property variations in a hydrocarbon reservoir. The approach produces

a porosity-permeability model comparable to the original petrophysical model using synthetic porosity

and permeability logs from the forward stratigraphic model as input datasets. As mentioned, this work

did not include variations in the layering scheme that develops in different zones of the stratigraphic

model. Under this circumstance, there is a possibility to overestimate and or underestimate porosity and permeability property in some sampled intervals in the validation wells. Therefore, we suggest that the forward stratigraphic simulation outputs such as the example presented in this contribution serve as additional data to understand sediment distribution patterns and associated vertical and horizontal petrophysical trends in the depositional environment, and not as absolute conditioning data in subsurface property modeling.

The assumptions made concerning the type of geological processes and input parameters in the stratigraphic simulation certainly differ from what existed during sediment deposition. So, applying stratigraphic models that fit a basin-scale description to a relatively smaller scale reservoir presents another level of uncertainty in this approach. This opinion agrees with Burges et al., (2008), where they indicate that the diffusion geological process simulation fits the description of large-scale sediment transportation. This view further buttresses the point that integrating forward stratigraphic simulation into a well-scale framework has a high chance of producing outcomes that deviate from the real-world subsurface description. In line with observations in Bertoncello et al. (2013); Aas et al. (2014); and Huang et al. (2015) in relations to limitations in the forward stratigraphic simulation method, it is advisable to use its outputs cautiously in reservoir modeling; as such outputs from forward stratigraphic models could lead to an increase in property representation bias in a model.

The correlation between reservoir lithofacies and petrophysics, and its prediction through reservoir models, have been extensively examined in several studies (Falivene et al.,2006; Hu and Chugunova,2008). Meanwhile, the predicted outputs most often do not depict the actual reservoir character due to the absence of a realistic 3-D stratigraphic framework to guide reservoir property representation in geological models. The forward stratigraphic modeling method, notwithstanding its limitations, provides reservoir modeling practitioners an platform to generate subsurface models that reflect the natural variation of reservoir properties.

## Conclusion

In this paper, synthetic well data from a forward stratigraphic simulation are combined with well data from the Volve field to predict porosity and permeability distribution. The forward stratigraphic modeling scenarios presented in this work do not prove that forward stratigraphic outputs should be used as absolute input parameters for a real-world reservoir modeling task. Considering the uncertainties highlighted in the choice of initial boundary conditions and geological processes for the stratigraphic simulation, it is notable that the simulation produced a depositional architecture that is geologically realistic and comparable to the stratigraphic correlation suggested in published studies of the study area. The match in porosity obtained by comparing validation wells in the original and stratigraphic-based petrophysical model indicates that it is practical to use variogram parameters and or well data from forward stratigraphic simulations for reservoir property modeling. This work also made two key findings:

1. For specific stratigraphic simulation in GPM$^{TM}$ and a range of model parameters, sediment transportation and deposition is based on diffusion rate and proximity to sediment source. This opinion agrees with several published works on sequence stratigraphy and or system tracts in shallow marine settings. However, further work with different forward stratigraphic modeling simulators could mitigate some of the challenges faced in this work.

2. A lithofacies distribution that is consistent with previous studies was produced in the stratigraphic model. This is evident in model scenarios where sediment distribution vertically matches with lithofacies variation in a sampled interval in an actual well log.

Geologically feasible stratigraphic patterns generated in the forward stratigraphic model provide an additional layer of confidence in representing facies distribution, and therefore porosity/permeability variations in a subsurface model. Furthermore, the resultant forward stratigraphic-based porosity and permeability model suggests that forward stratigraphic modeling can be integrated into geostatistical modeling workflows to improve subsurface property modeling and well planning.

## Data and Code Availability

The dataset for this work is from Equinor (Volve field, Norway), and was made available to the public in 2018. The data include 24 suits of well logs, and 3-D reservoir models in Eclipse and RMS formats. The data, models (eclipse and RMS formats), and the rule-based calculation script to generate lithofacies and porosity/permeability proportions are archived on Zenodo as Otoo & Hodgetts, (2020).

### GPM$^{TM}$ Software

The (2019.1) version of GPM$^{TM}$ software was used in completing this work after an initial 2018.1 version. Available on: https://www.software.slb.com/products/gpm. The software license and code used in the GPM$^{TM}$ cannot be provided, because Schlumberger does not allow the code for its software to be shared in publications.

### Model Availability in Petrel$^{TM}$

The work started in Petrel$^{TM}$ software (2017.1), but it was completed with Petrel$^{TM}$ software (2019.1). The software is available on: https://www.software.slb.com/products/petrel. The software runs on a Windows PC with the following specifications: Processor; Intel Xeon CPU E5-1620 v3 @3.5GHz 4 cores-8 threads, Memory; 64 GB RAM. The computer should be high end, because a lot of processing time is required for the task. The forward stratigraphic models are in Zenodo as Otoo & Hodgetts, (2020).

## Author Contribution

Daniel Otoo designed the model workflow, conducted the simulation using the GPM$^{TM}$ software, evaluated the results, and drafted the manuscript. David Hodgetts converted the Volve field data into Petrel compactible format and assisted in the revision of the manuscript.

## Acknowledgement

Thanks to Equinor for making available the Volve field dataset. Also, thanks to Schlumberger for providing GPM$^{TM}$ software license. A special thanks to Mostfa Legri and Daniel Tetzlaff (Schlumberger) for their technical support in the use of GPM$^{TM}$. Finally, to the Ghana National Petroleum Corporation (GNPC) for sponsoring this research.

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

## List of Figures

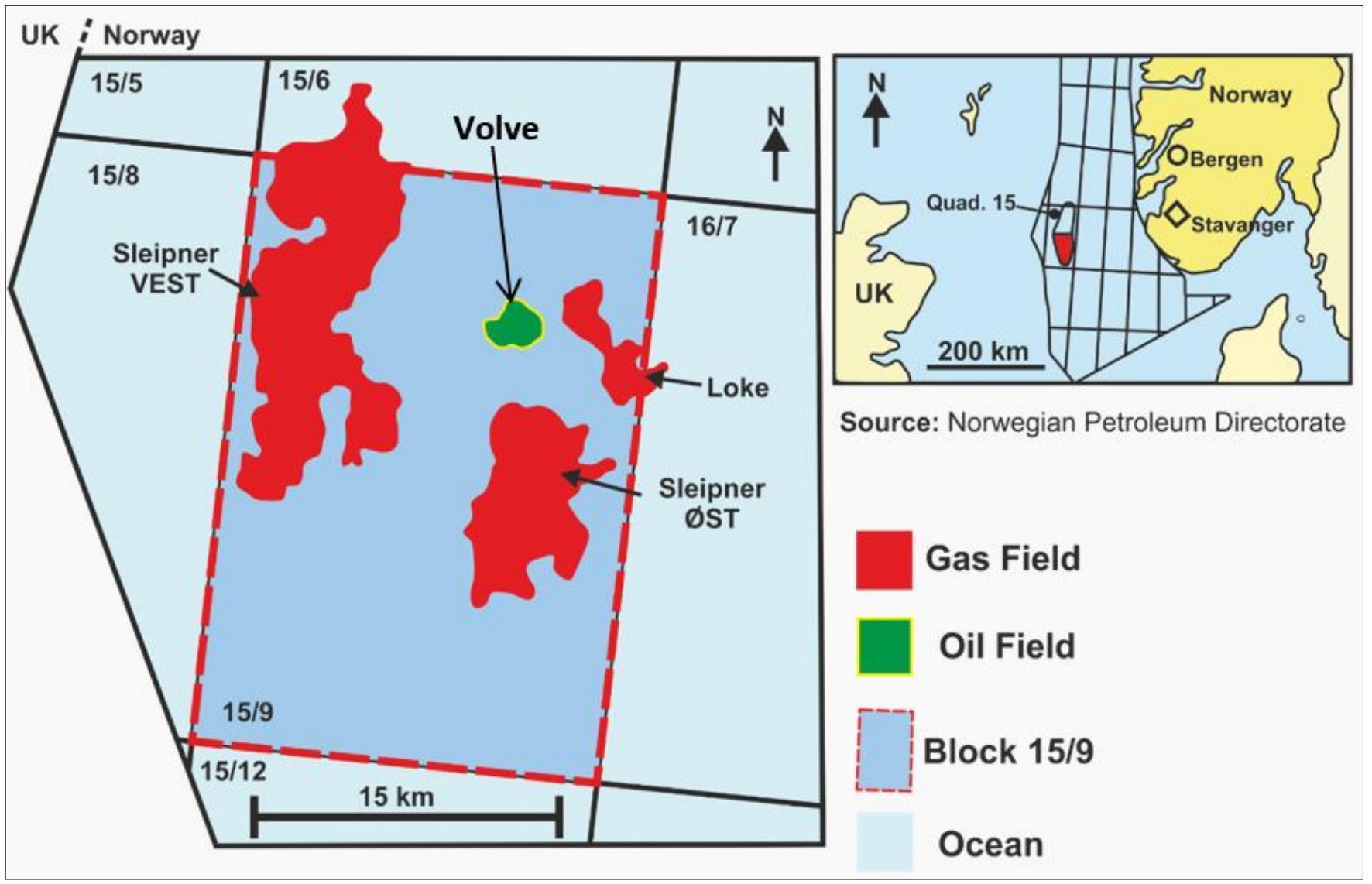

**Fig 1.** Location map of the Volve field; showing gas and oil fields in quadrant 15/9, Norwegian North Sea (from Ravasi et al., 2015).

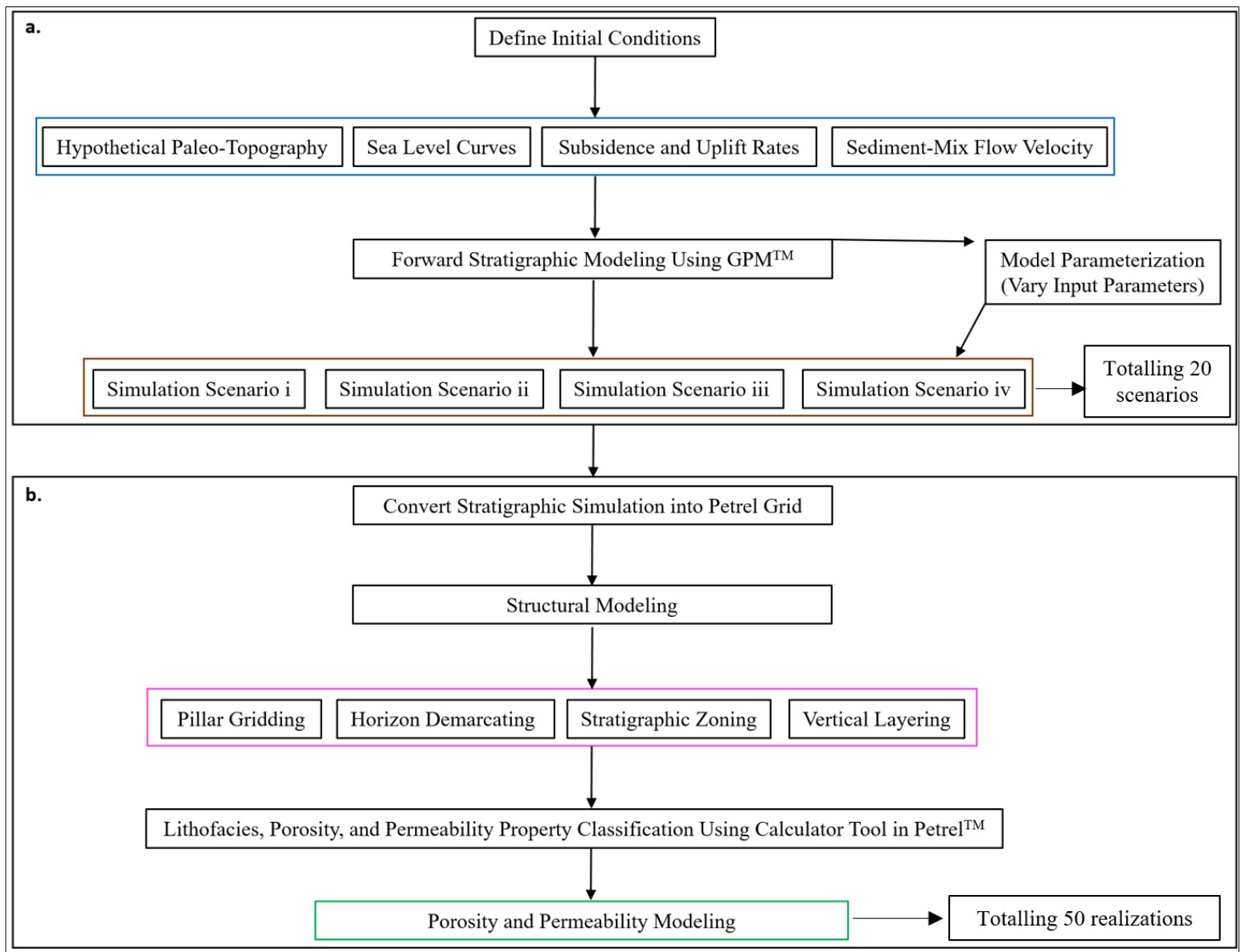

**Fig 2**. Schematic workflow of processes involved in this work. a. information of boundary conditions (input parameters) used for the forward stratigraphic simulation in GPM[TM]; b. illustrate the use of forward stratigraphic models in Petrel[TM] for porosity and permeability modeling.

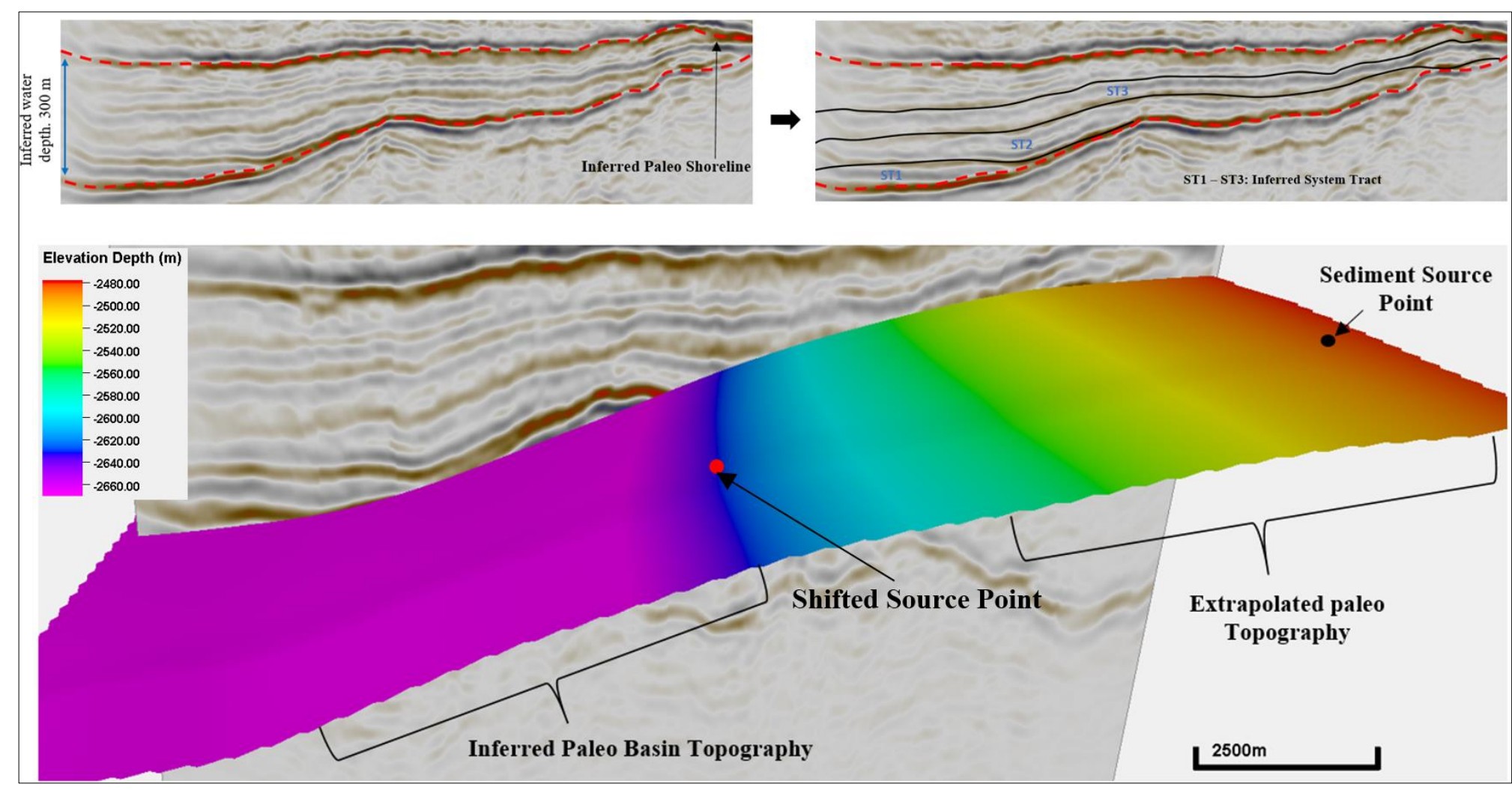

**Fig 3**. 3-D seismic section of the study area. Hypothetical topographic surface is derived from present-day base of reservoir. The sediment entry point into the basin is located in the North Eastern section (based on Kieft et al. 2011).

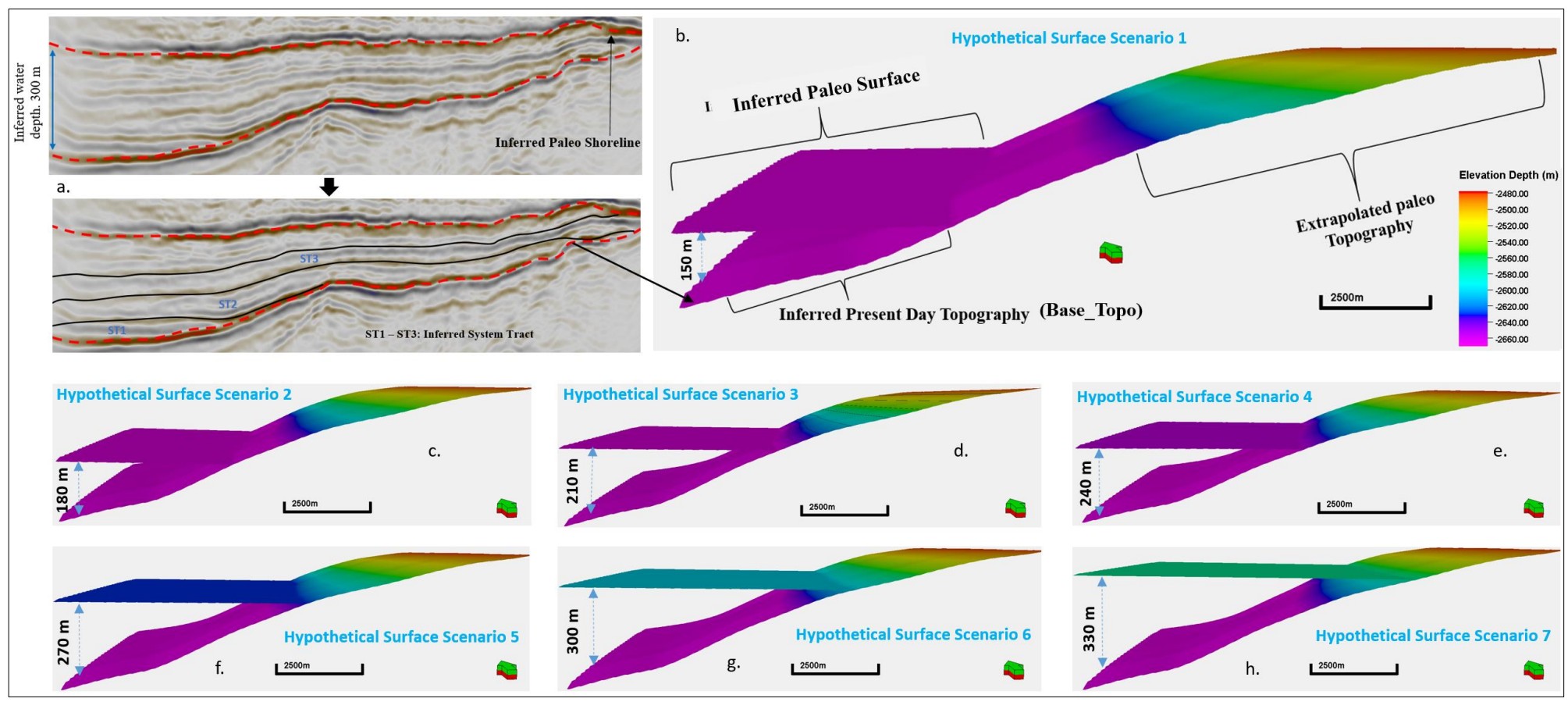

**Fig 4.** Illustrating a range of hypothetical initial topographic surfaces that were used to mitigate the uncertainty in selecting an initial topographic surface for the simulation. Considering that the topographic surface is a key control on stratigraphic sequence, different stratigraphic models are generated to attain a "best-fit" model.

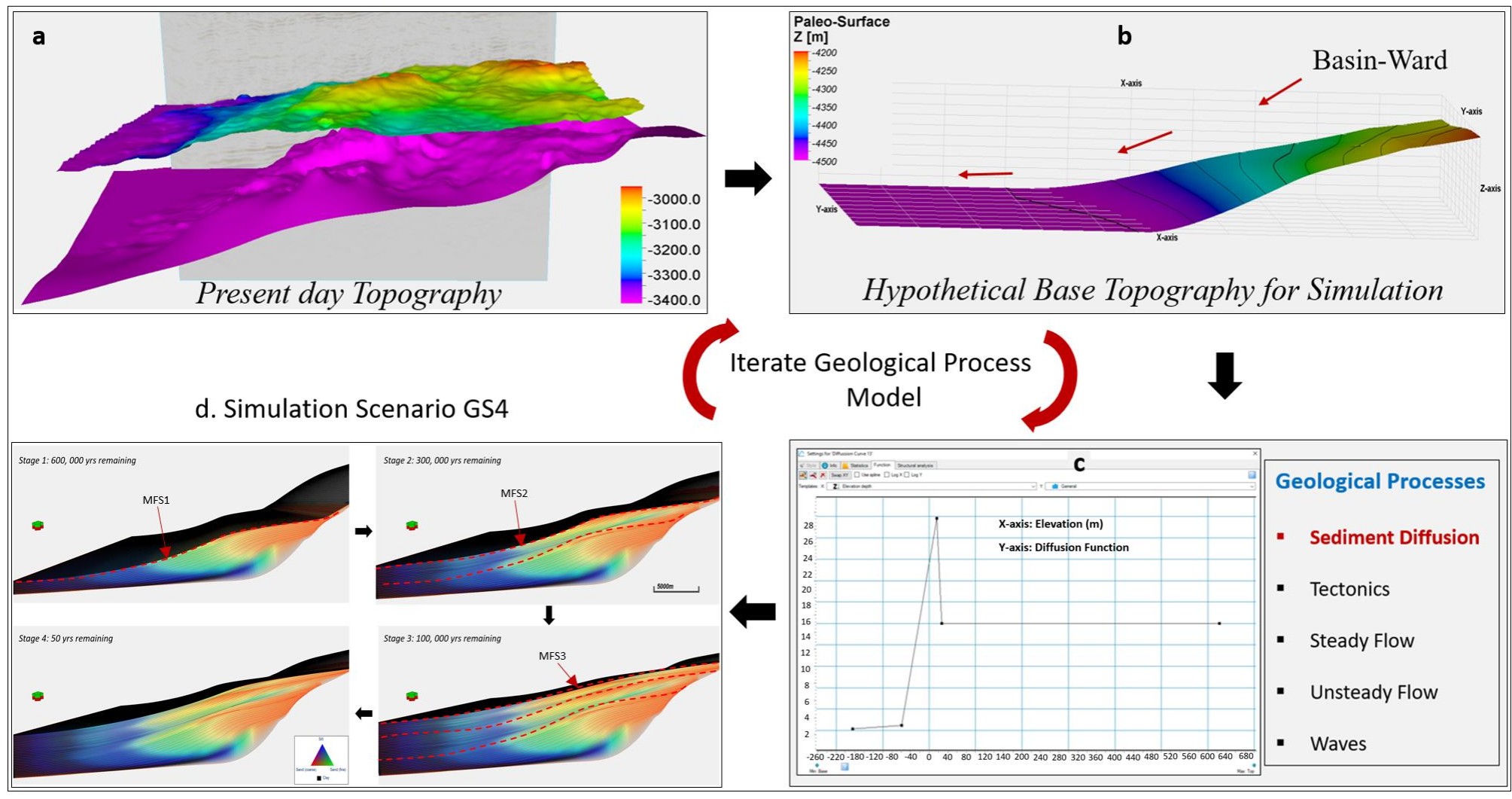

**Fig 5.** a. Present-day top and bottom topographic surfaces of the Hugin formation; b. hypothetical topographic surface after reprocessing of the base reservoir surface; c. stages of geological processes involved in the forward stratigraphic simulation; d. forward stratigraphic models at different time intervals of the simulation.

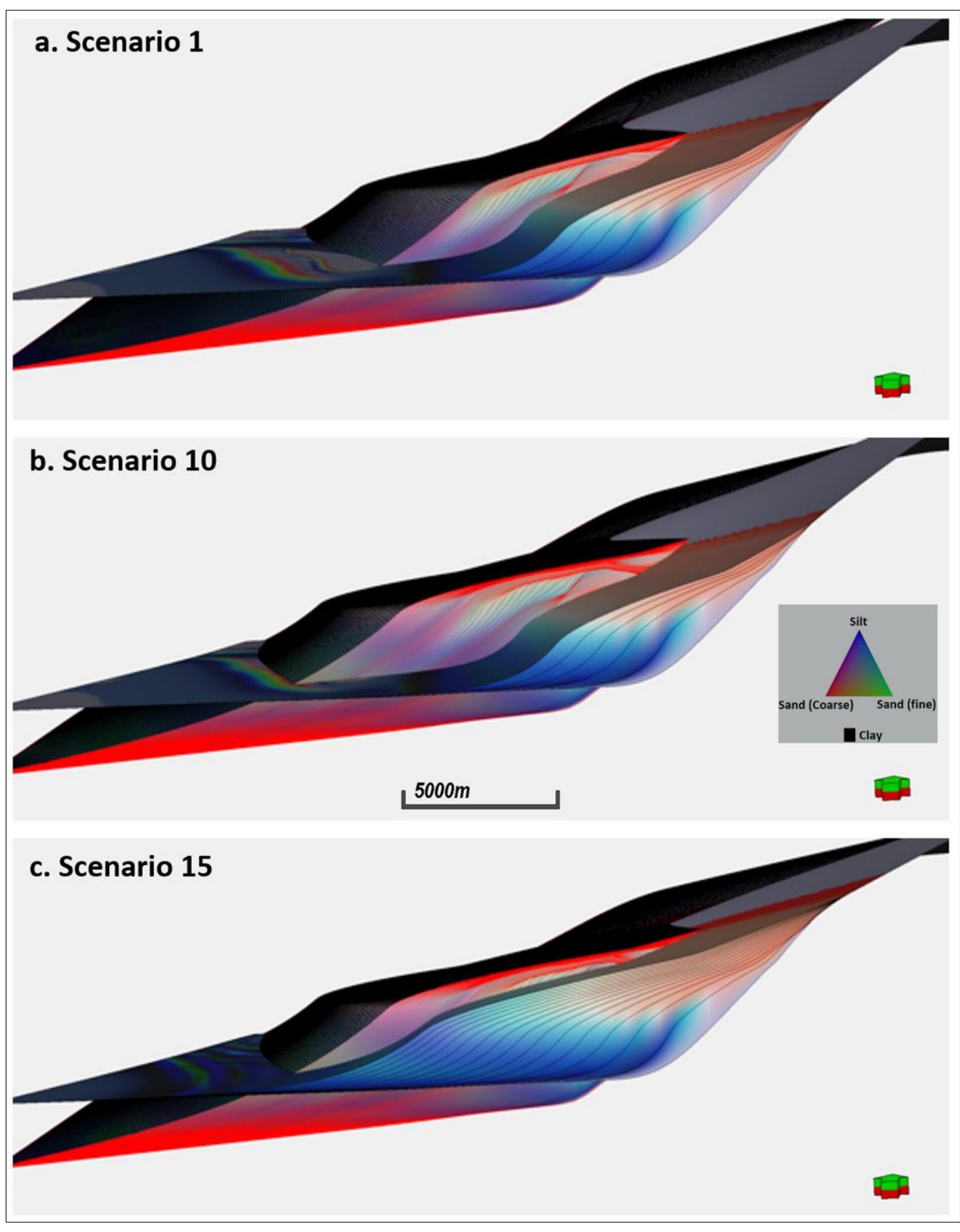

**Fig 6.** Example of stratigraphic simulation scenarios, from which the "best-fit" model was selected. **a.** involves the use of equal proportions of sediment supply, a relatively low subsidence rate and low water depth, **b.** applies a high proportions of fine sand and silt (70%) in the sediment mix, abrupt changes in subsidence rate, and a relatively high sea-level, **c**. involves very high proportions of fine sand and silt (80%), steady rate of subsidence and uplift in the sediment source area, and a relatively low water depth.

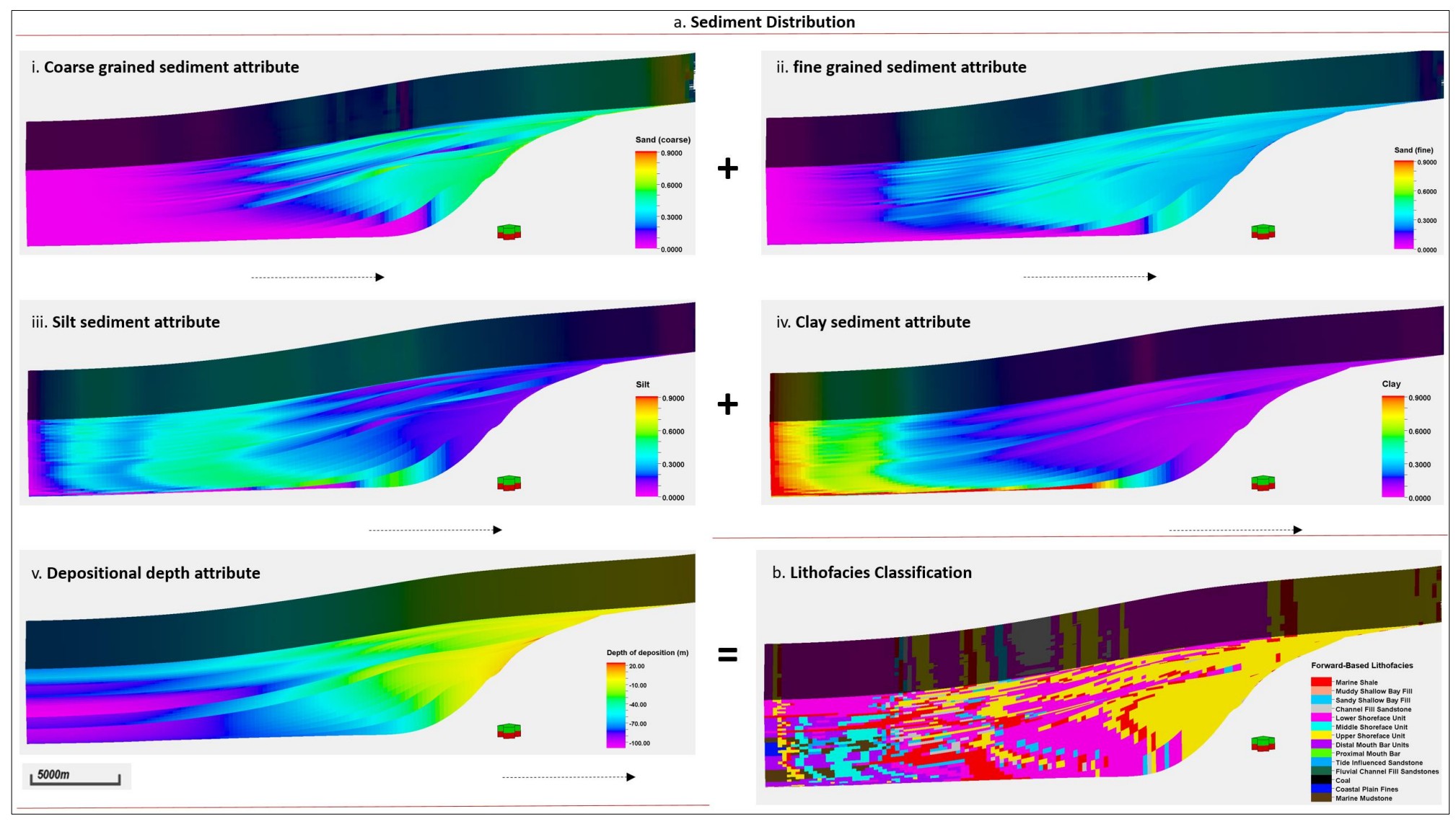

**Fig 7 a.** Sediment distribution patterns in the geological process modeling software. b. lithofacies classification using the property calculator tool in Petrel[TM].

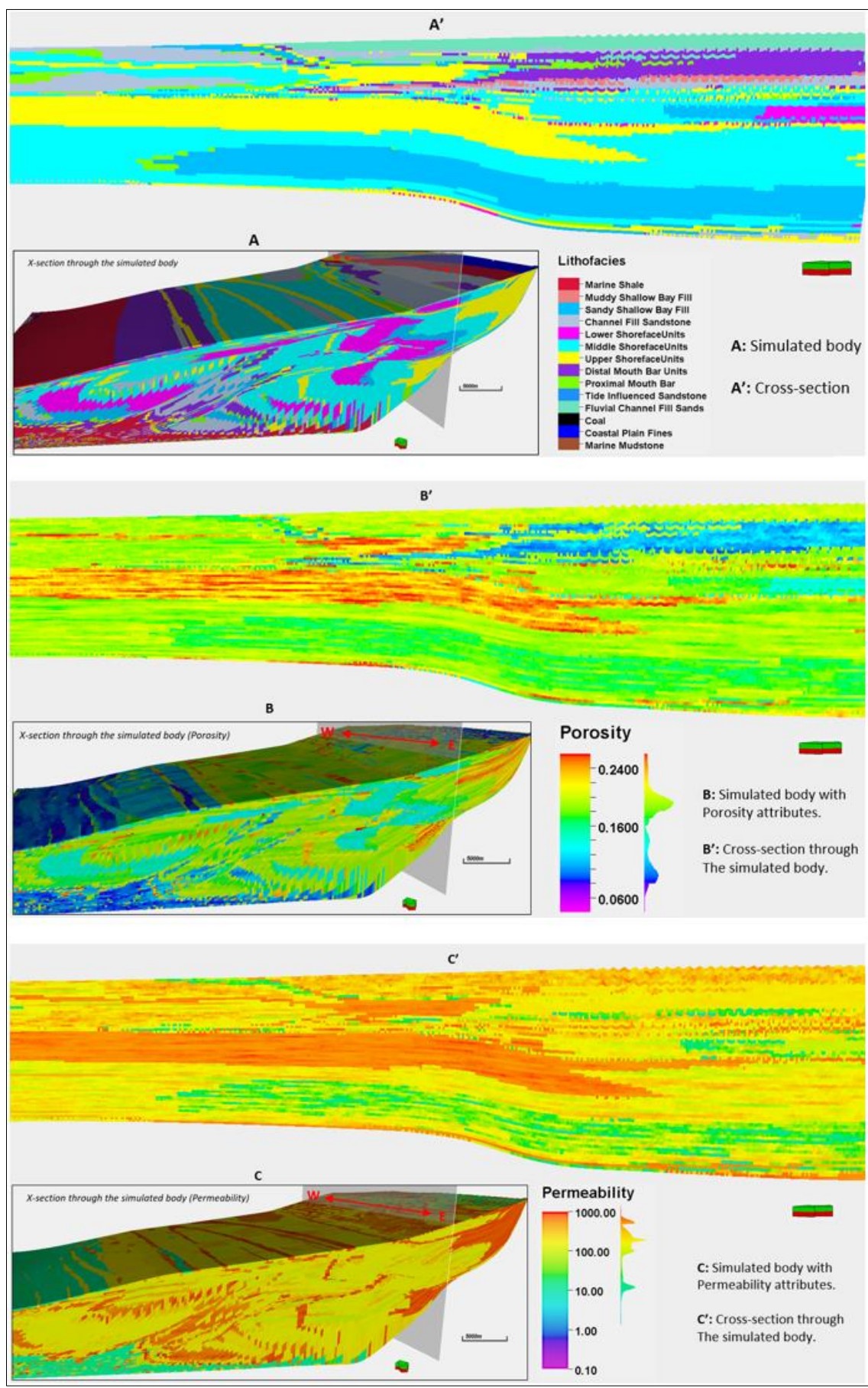

**Fig 8.** Lithofacies, porosity and permeability trends in the forward stratigraphic-based models.

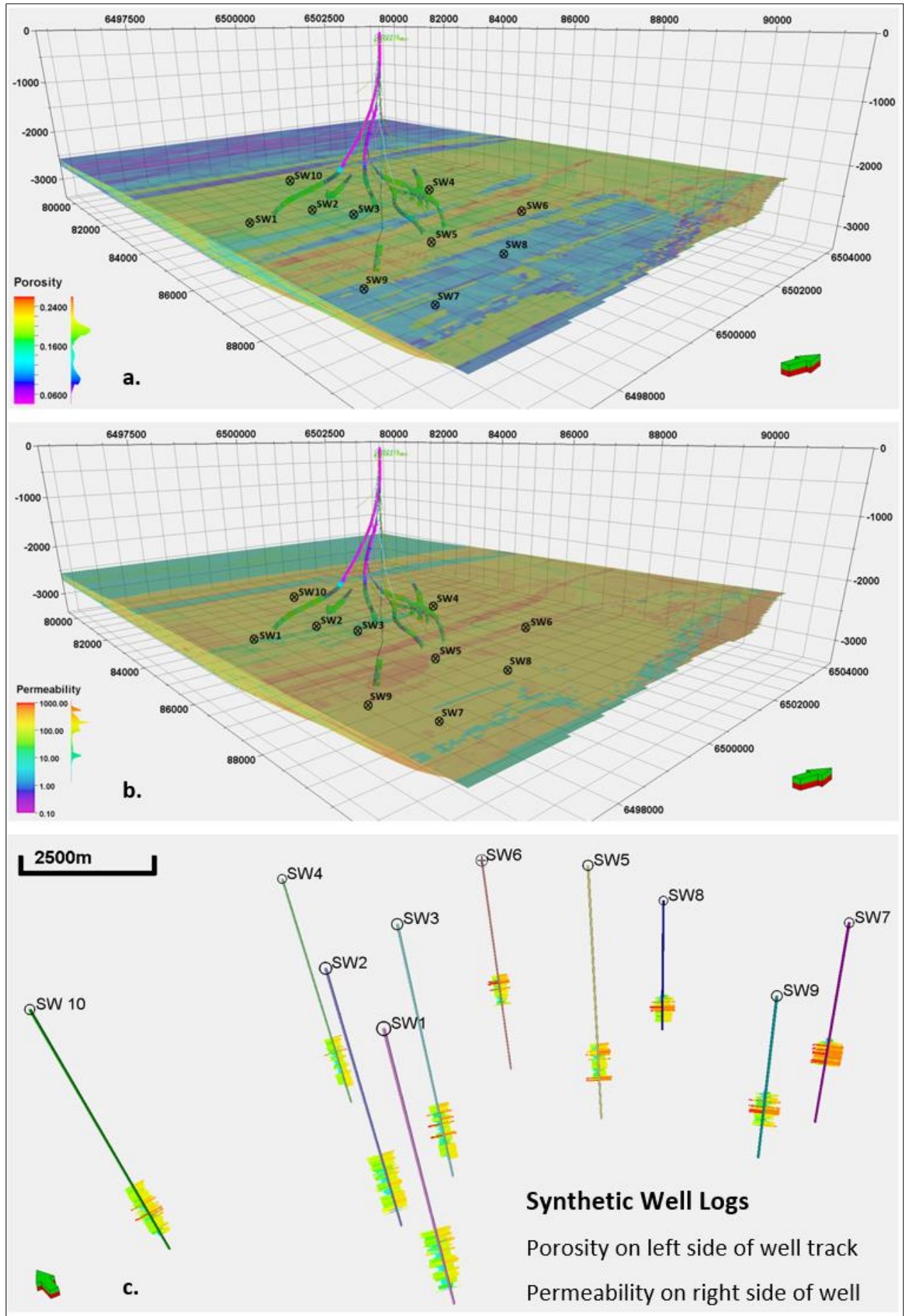

**Fig 9.** Synthetic wells forward stratigraphic-based porosity and permeability models. The average separation distance between wells is about 0.9 km.

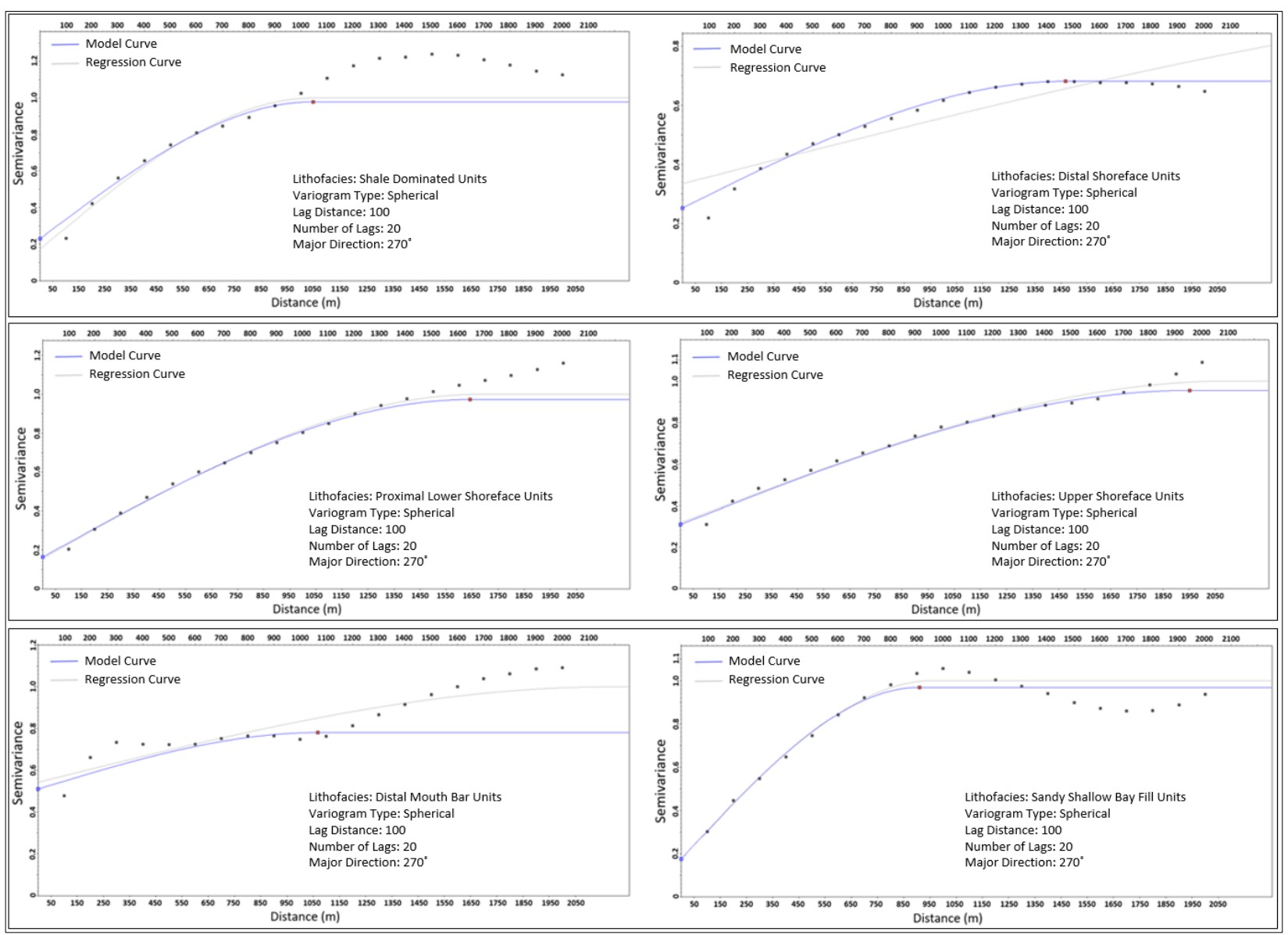

**Fig 10.** Variogram model of dominant lithofacies units from the forward stratigraphic model. The "dots" indicate the number of lags in the variogram through the major direction (NE-SW) of the stratigraphic model.

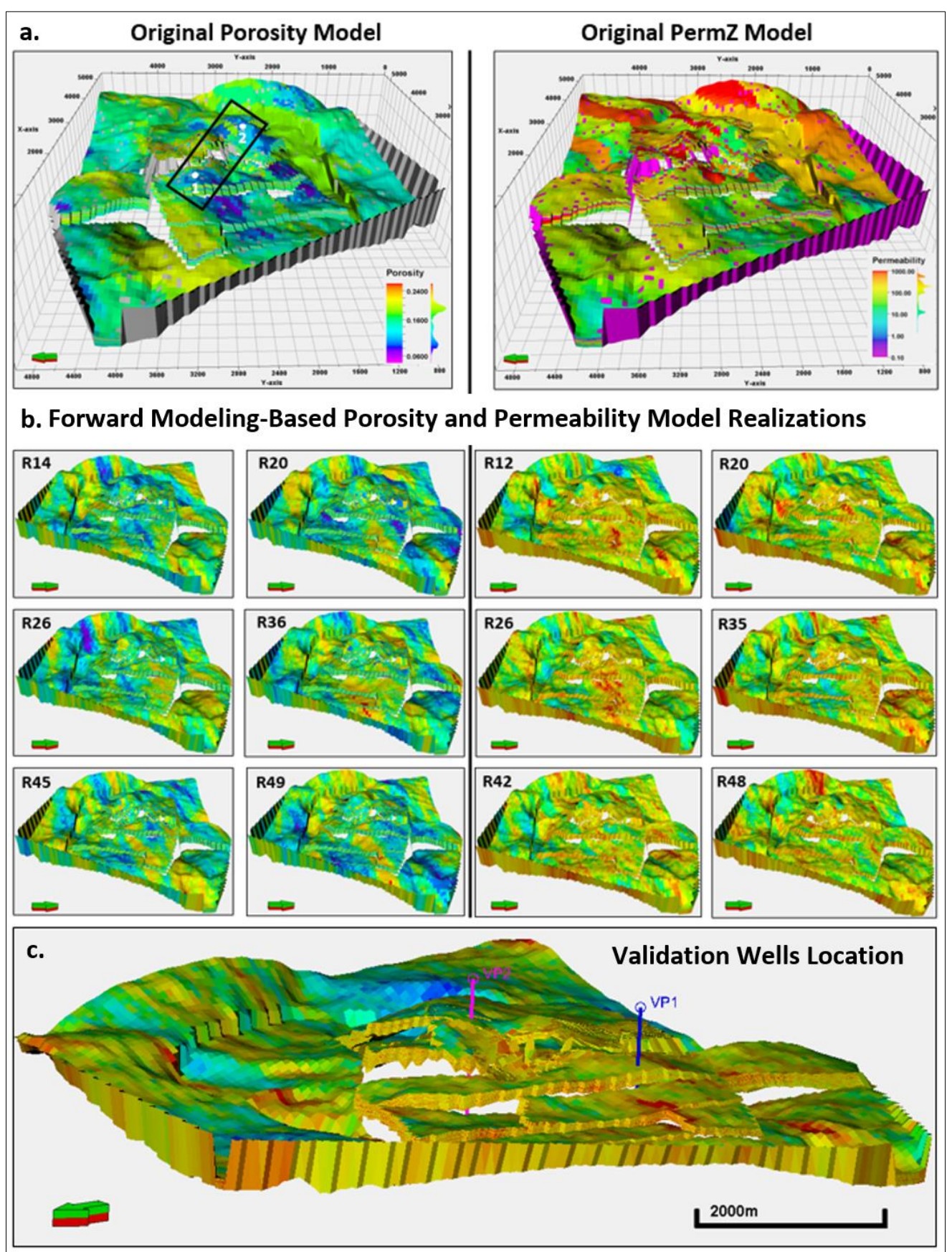

**Fig 11.** Comparing the Original Volve vs the forward modeling-based porosity and permeability models. Realizations 16, 20, 26, 36, 45, and 49 on the left half are porosity models, whiles realizations 12, 20, 26, 35, 42, and 48 on the right half are permeability models.

# a. Validation Well 1

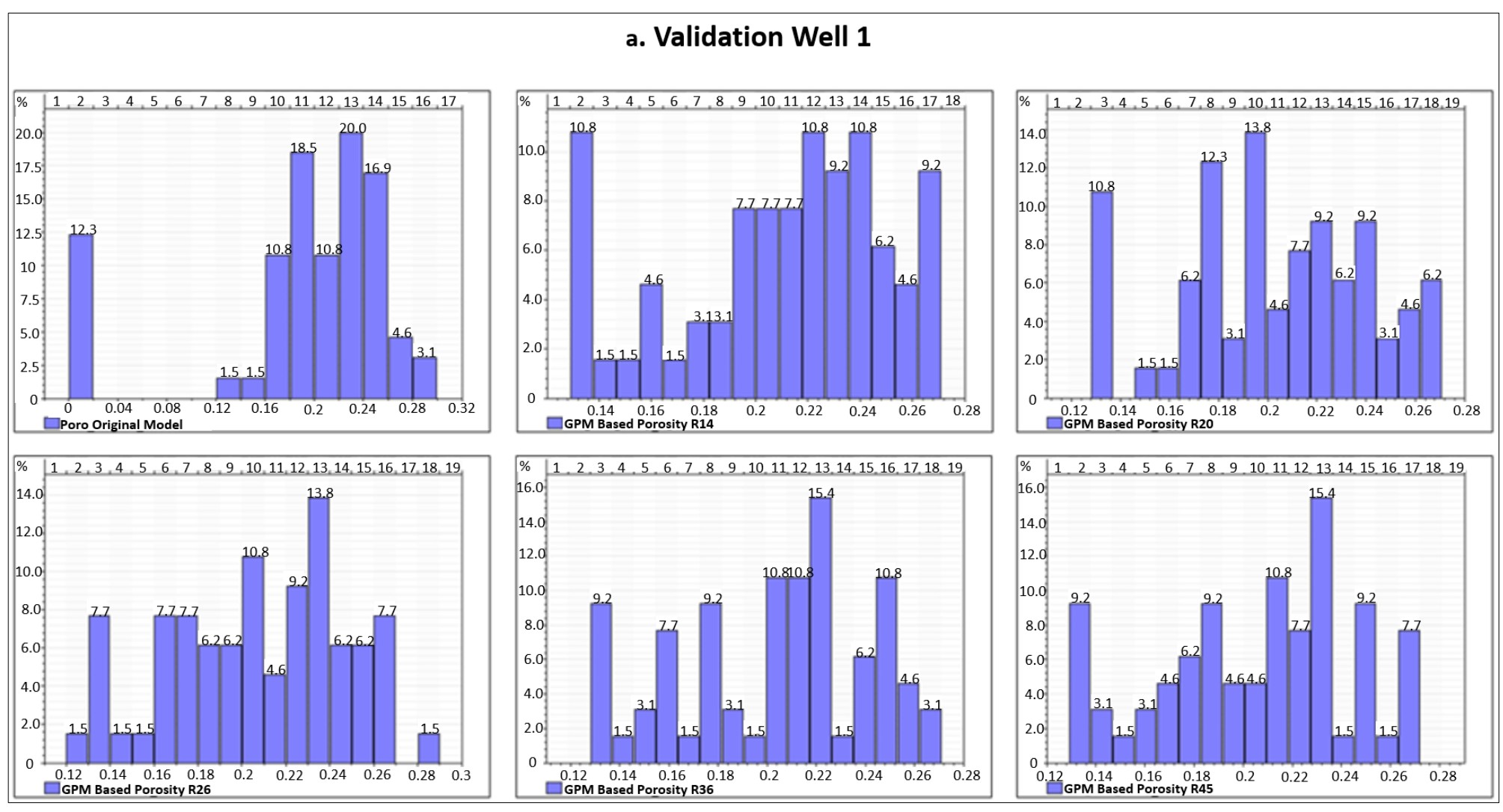

**Fig 12a.** Histogram illustrating porosity distribution in validation Well 1 of five stratigraphic-based realizations, and the original porosity model at identical vertical intervals.

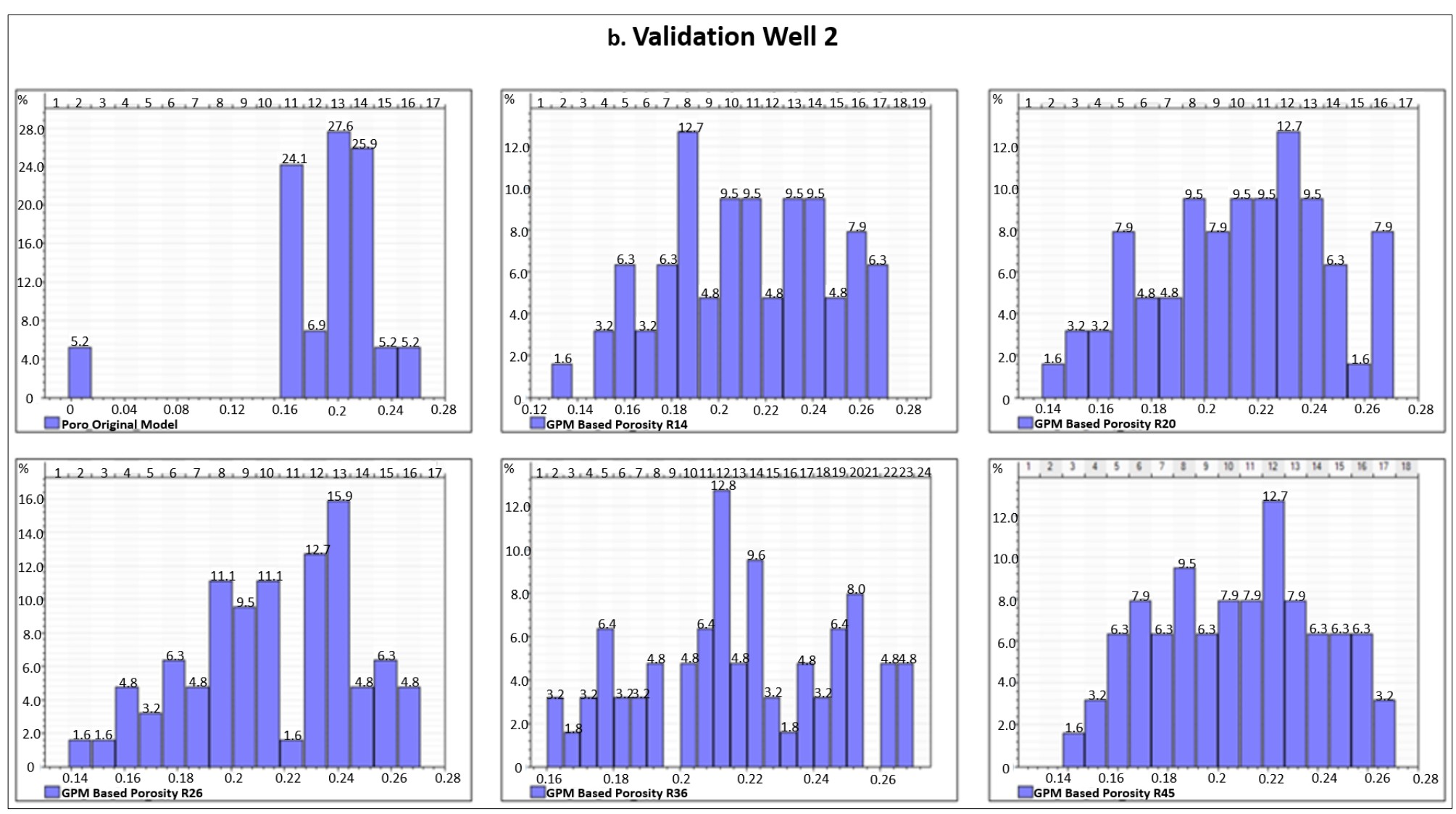

**Fig 12b.** Histogram illustrating porosity distribution in validation Well 2 of five stratigraphic-based realizations, and the original porosity model at identical vertical intervals.

**Table 1** Lithofacies-associations in the Hugin formation, Volve Field (after Kieft et al. 2011).

| Code | Facies | Description | Thickness (t); Extent (l) | Wireline-log Attribute | Interpretation |
|------|--------|-------------|---------------------------|------------------------|----------------|
| A | A1 | Parallel-laminated mudstone with occasional siltstone inputs. Monospecific pattern of disorder bivalves parallel to bedding. | t = 30 - 425 cm  l = 6 - 29 km | GR = 41 - 308 API<br>DT = 225 - 355 $\mu sm^{-1}$<br>NPHI = 0.17 - 0.45 v/v<br>RHOB = 2280 - 2820 $gcm^{-1}$ | Restricted marine shale |
| | A2 | Inter-bedded claystone and very fine-grained sandstone; non-parallel and wavy lamination. Scarecely bivalve shells oriented parallel to bedding. | t = 10 - 725 cm  l = 8 - 13 km | GR = 17 - 65 API<br>DT = 189 - 268 $\mu sm^{-1}$<br>NPHI =?<br>RHOB = 2280 - 2820 gcm-1 | Muddy hallow bay fill |
| | A3 | Fine to medium grained sandstone; moderately to well sorted grain. Wavy bedding, cross bedding, rare wave ripples. | t = 60 - 370 cm  l = 1 - 8 km | GR = 18 - 46 API<br>DT = 199 - 268 $\mu sm^{-1}$<br>NPHI = 0.07 - 0.52 v/v<br>RHOB = 1690 - 2745 gcm-1 | Sandy shallow bay fill |
| | A4 | Parallel-laminated mudstone with occasional siltstone inputs. Monospecific pattern of disorder bivalves parallel to bedding. | t = 30 - 425 cm  l = 6 - 29 km | GR = 7 - 35 API<br>DT = 175 - 230 $\mu sm^{-1}$<br>NPHI = 0.04 - 0.15 v/v<br>RHOB = 2280 - 2820 gcm-1 | Marine channel fill sandstone |
| B | B1 | Upward coarsening siltstone to fine-grained; moderatley sorted sandstone. Shell debris and quartz granules. | t = 30 - 480 cm  l = 1 - 2 km | GR = 18 - 80 API<br>DT = 168 - 291 $\mu sm^{-1}$<br>NPHI = 0.04 - 0.191 v/v<br>RHOB = 2322 - 2723 gcm-1 | Distal lower shoreface |
| | B2 | Very fine-fine grained sandstone. Moderate to well sorted; fine grained carbonaceous laminae, typically low angle cross beds. | t = 130 - 440 cm  l = 1.7 - 12 km | GR = 20 - 56 API<br>DT = 179 - 277 $\mu sm^{-1}$<br>NPHI = 0.05 - 0.168 v/v<br>RHOB = 2314 - 2696 gcm-1 | Proximal lower shoreface |
| | B3 | Coaesening upward, cross laminated, fine to medium grainned sandstone; consist of carbonaceous fragments. | t = 425 - 800 cm  l = 1.7 - 8 km | GR = 15 - 25 API<br>DT = 250 - 275 $\mu sm^{-1}$<br>NPHI = 0.09 - 0.113 v/v<br>RHOB = 2271 - 2342 gcm-1 | Upper shoreface |
| C | C1 | Highly bioturbated siltstone to very fine sandstone, with beds of rounded granules. | t = 175 - 1010 cm  l = 7.2 - 19.6 km | GR = 20 - 80 API<br>DT = 230 - 260 $\mu sm^{-1}$<br>NPHI = 0.08 - 0.169 v/v<br>RHOB = 2327 - 2521 gcm-1 | Distal mouth bar |
| | C2 | Very fine to fine grained sandstone, low angle cross bedding. | t = 290 - 775 cm  l = 1 - 5 km | GR = 12 - 58 API<br>DT = 167 - 397 $\mu sm^{-1}$<br>NPHI = 0.05 - 0.595 v/v<br>RHOB = 1612 - 2705 gcm-1 | Proximal mouth bar |
| D | D1 | Fining upward coarse to fine grained sandstone. Stacked fining upward beds with rare coarse grained stringers. | t = 740 - 820 cm  l = 1 - 2 km | GR = 8 - 134 API<br>DT = 235 - 335 $\mu sm^{-1}$<br>NPHI = 0.14 - 0.46 v/v<br>RHOB = 2284 - 2570 gcm-1 | Tidal influenced fluvial channel fill sandstone |
| | D2 | Fining upward coarse to medium grained sandstone. Carbonaceous laminae and fragments. Sharp and cohessive contact at base of bed. | t = 580 cm  l = < 2 km | GR = 9 - 34 API<br>DT = 241 - 297 $\mu sm^{-1}$<br>NPHI = 0.14 - 0.289 v/v<br>RHOB = 2168 - 2447 gcm-1 | fluvial channel fill sandstone |
| E | E1 | Coal and carbonaceous shale. Basal contact typically parallel, although maybe undulose. | t = 30 - 520 cm  l = 6 - 19.6 km | GR = 8 - 56 API<br>DT = 313 - 427 $\mu sm^{-1}$<br>NPHI = 0.24 - 0.529 v/v<br>RHOB = 1930 - 2225 gcm-1 | Coal |
| | E2 | Alternating dark grey mudstone/claystone and siltstone to very fine grained sandstone. Wavy to non-parallel lamination. | t = 60 cm  l = < 2 km | GR = 32 - 60 API<br>DT = 358 - 415 $\mu sm^{-1}$<br>NPHI = 0.43 - 0.49 v/v<br>RHOB = 1994 - 2148 gcm-1 | Coastal plain fines |
| F | F | Mudstone with rare siltstone beds. Parallel lamination, soft sediment deformation developed locally on top of beds. | t = section tot completely penetrated  l = 1.7 - 36.7 km | GR = 4 - 134 API<br>DT = 187 - 450 $\mu sm^{-1}$<br>NPHI = 0.114 - 0.618 v/v<br>RHOB = 1730 - 2925 gcm-1 | Open marine shale |

**Table 2.** Input parameters for forward stratigraphic simulations in GPM™

## Initial Conditions- GPM Input Parameters

| | | Simulation Duration | Sediment Type Proportion (%) | | | | Avg. Water Velocity | Avg. Sediment Velocity | Erodibility | Diffusion Coefficient | Avg. Sea Level | Turbidite Event Interval | Steady Flow Iteration | Sediment Movement |
|---|---|---|---|---|---|---|---|---|---|---|---|---|---|---|
| | | (Ma– 0a) Years | Sand (Coarse) | Sand (Fine) | Silt | Clay | (m/a) | (m/a) | | | Interval (m) | (/years) | (/hrs) | Coefficient |
| | S1 | 0.02 – 0 | 25 | 25 | 25 | 25 | 0.11 | 0.03 | 0.35 | 0.11 | 30 | 2500 | 10 | 0.001 |
| | S2 | 0.25 – 0 | 25 | 25 | 25 | 25 | 0.15 | 0.03 | 0.45 | 0.15 | 70 | 1000 | 15 | 0.012 |
| | S3 | 0.5 – 0 | 25 | 25 | 25 | 25 | 0.11 | 0.02 | 0.55 | 0.11 | 120 | 1000 | 20 | 0.012 |
| | S4 | 0.7 – 0.05 | 25 | 25 | 25 | 25 | 0.08 | 0.02 | 0.35 | 0.08 | 100 | 500 | 25 | 0.0011 |
| | S5 | 1.5 – 0 | 15 | 35 | 30 | 20 | 0.15 | 0.04 | 0.50 | 0.15 | 80 | 5000 | 20 | 0.001 |
| | S6 | 3.0 – 0 | 50 | 25 | 15 | 10 | 0.13 | 0.04 | 0.50 | 0.13 | 70 | 5000 | 30 | 0.0012 |
| | S7 | 3.5 – 0 | 50 | 25 | 15 | 10 | 0.11 | 0.04 | 0.50 | 0.11 | 70 | 10000 | 15 | 0.001 |
| | S8 | 4.0 – 0 | 50 | 25 | 15 | 10 | 0.13 | 0.04 | 0.50 | 0.13 | 90 | 5000 | 20 | 0.0015 |
| | S9 | 4.5 – 0 | 15 | 45 | 25 | 15 | 0.1 | 0.02 | 0.45 | 0.1 | 50 | 10000 | 30 | 0.0012 |
| | S10 | 5.0 – 0 | 15 | 45 | 25 | 15 | 0.12 | 0.02 | 0.45 | 0.12 | 55 | 10000 | 35 | 0.0013 |
| | S11 | 5.5 - 0 | 15 | 45 | 25 | 15 | 0.12 | 0.02 | 0.45 | 0.12 | 40 | 5000 | 40 | 0.0013 |
| | S12 | 6.0 – 0 | 15 | 45 | 25 | 15 | 0.1 | 0.02 | 0.45 | 0.1 | 60 | 10000 | 35 | 0.0011 |
| | S13 | 6.5 – 0 | 10 | 25 | 55 | 10 | 0.13 | 0.03 | 0.48 | 0.13 | 100 | 20000 | 50 | 0.0010 |
| | S14 | 7.0 – 0 | 10 | 25 | 55 | 10 | 0.16 | 0.03 | 0.48 | 0.16 | 40 | 20000 | 45 | 0.0011 |
| | S15 | 7.5 – 0 | 10 | 25 | 55 | 10 | 0.13 | 0.03 | 0.48 | 0.13 | 40 | 20000 | 40 | 0.0012 |
| | S16 | 8.0 – 0 | 10 | 25 | 55 | 10 | 0.15 | 0.03 | 0.48 | 0.15 | 30 | 10000 | 30 | 0.0010 |
| | S17 | 8.5 – 0 | 10 | 25 | 45 | 20 | 0.14 | 0.02 | 0.45 | 0.14 | 50 | 50000 | 50 | 0.0010 |
| | S18 | 9.0 – 0 | 30 | 30 | 18 | 22 | 0.13 | 0.02 | 0.52 | 0.13 | 60 | 25000 | 35 | 0.0012 |
| | S19 | 9.5 – 0 | 30 | 40 | 12 | 18 | 0.12 | 0.02 | 0.55 | 0.12 | 55 | 25000 | 20 | 0.0013 |
| | S20 | 10.0 - 0 | 30 | 42 | 18 | 10 | 0.11 | 0.01 | 0.40 | 0.11 | 50 | 5000 | 15 | 0.0011 |

*GPM Scenarios (GS)* (vertical label)

## Sediment Property

| Sediment Type | Diameter | Density | Initial Porosity | Initial Permeability | Compacted Porosity | Compaction | Compacted Permeability | Erodibility |
|---|---|---|---|---|---|---|---|---|
| Coarse Grained Sand | 1.0 mm | 2.70 g/cm³ | 0.21 m³/m³ | 500 mD | 0.25 m³/m³ | 5000 KPa | 50 mD | 0.6 |
| Fine Grained Sand | 0.1 mm | 2.70 g/cm³ | 0.3 m³/m³ | 100 mD | 0.15 m³/m³ | 2500 KPa | 5 mD | 0.45 |
| Silt | 0.01 mm | 2.65 g/cm³ | 0.38 m³/m³ | 50 mD | 0.12 m³/m³ | 1200 KPa | 2 mD | 0.3 |
| Clay | 0.001 mm | 2.65 g/cm³ | 0.48 m³/m³ | 5 mD | 0.05 m³/m³ | 500 KPa | 0.1 mD | 0.15 |

**Table 3.** Lithofacies classification in the forward stratigraphic model by using the property calculator tool in Petrel™.

| | | Lithofacies Classification |
|---|---|---|
| **Facies Code** | **Lithofacies** | **Command Used in Petrel's Property Calculator** |
| 0 | Marine Shale | If(Sand_fine>=0.19 And Sand_fine<=0.21 Or Silt>=0.19 And Silt<=0.2 Or Clay>=0.2 And Clay<=0.21 Or Depth_of_deposition>=-82 And Depth_of_deposition<=-78) |
| 1 | Muddy Shallow Bay Fill | If( Sand_fine>=0.36 And Sand_fine<=0.38 Or Silt>=0.18 And Silt<=0.2 Or Clay>0.18 And Clay<=0.19 Or Depth_of_deposition>=-30 And Depth_of_deposition<=-20) |
| 2 | Sandy Shallow Bay Fill | If(Sand_coarse>=0.65 And Sand_coarse<=0.73 Or Sand_fine>=0.18 And Sand_fine<=0.22 Or Silt>=0.18 And Silt<=0.2 Or Clay>=0.17 And Clay<=0.18 Or Depth_of_deposition>=-3 And Depth_of_deposition<=0) |
| 3 | Channel Fill Sandstone | If( Sand_coarse>=0.5 And Sand_coarse<=0.68 Or Sand_fine>=0.23 And Sand_fine<=0.25 Or Silt>=0.17 And Silt<=0.18 Or Depth_of_deposition>=0 And Depth_of_deposition<=2) |
| 4 | Lower Shoreface Units | If( Sand_coarse>=0.19 And Sand_coarse<=0.31 Or Sand_fine>=0.19 And Sand_fine<=0.24 Or Silt>=0.4 And Silt<=0.48 Or Clay>=0.19 And Clay<=0.31 Or Depth_of_deposition>=-83 And Depth_of_deposition<=50) |
| 5 | Middle Shoreface Units | If( Sand_coarse>=0.32 And Sand_coarse<=0.53 Or Sand_fine>=0.25 And Sand_fine<=0.32 Or Silt>=0.26 And Silt<=0.32 Or Clay>=0.19 And Clay<=0.21 Or Depth_of_deposition>=-38 And Depth_of_deposition<=-12) |
| 6 | Upper Shoreface Units | If(Sand_coarse>=0.53 And Sand_coarse<=0.72 Or Sand_fine>=0.28 And Sand_fine<=0.33 Or Silt>=0.16 And Silt<=0.21 Or Depth_of_deposition>=-10 And Depth_of_deposition<=6) |
| 7 | Distal Mouth Bar Units | If( Sand_fine>=0.23 And Sand_fine<=0.27 Or Silt>=0.38 And Silt<=0.43 Or Clay>=0.19 And Clay<=0.21 Or Depth_of_deposition>=-95 And Depth_of_deposition<=-80) |
| 8 | Proximal Mouth Bar Units | If( Sand_coarse>=0.53 And Sand_coarse<=0.71 Or Sand_fine>=0.27 And Sand_fine<=0.32 Or Silt>=0.16 And Silt<=0.21 Or Clay>=0.06 And Clay<=0.07 Or Depth_of_deposition>=-30 And Depth_of_deposition<=-27) |
| 9 | Tide Influenced Sandstones | If( Sand_coarse>=0.53 And Sand_coarse<=0.71 Or Sand_fine>=0.26 And Sand_fine<=0.31 Or Silt>=0.35 And Silt<=0.41 Or Depth_of_deposition>=-5 And Depth_of_deposition<=1) |
| 10 | Fluvial Channel Sandstones | If(Sand_coarse>=0.54 And Sand_coarse<=0.56 Or Sand_fine>=0.27 And Sand_fine<=0.29 Or Silt>=0.19 And Silt<=0.21 Or Depth_of_deposition>=-2 And Depth_of_deposition<=2) |
| 11 | Coal | Estimated as background attribute |
| 12 | Coastal plain fines | If( Silt>=0.31 And Silt<=0.43 Or Clay>=0.31 And Clay<=0.35 Or Depositional_depth>=-100 And Depositional_depth<=-40) |
| 13 | Marine Mudstone | If( Sand_fine>=0.36 And Sand_fine<=0.38 Or Silt>=0.4 And Silt<=0.52 Or Clay>=0.45 And Clay<=0.78 Or Depth_of_deposition>=-105 And Depth_of_deposition<=-90) |

**Table 4.** Porosity and Permeability estimates of lithofacies packages in the model area.

| Code | Lithofacies | Avg. NPHI | Density Porosity | Estimated Porosity | KLOGH (mD) |
|------|-------------|-----------|------------------|--------------------|------------|
| 0 | Marine Shale | 0.17 - 0.45 | 0.1 | 0.08 - 0.11 | 10.02 - 16.1 |
| 1 | Muddy Shallow Bay Fill | 0.17 - 0.42 | 0.1 | 0.08 - 0.13 | 23.85 - 102.3 |
| 2 | Sandy Shallow Bay Fill | 0.07 - 0.52 | 0.25 | 0.16 - 0.25 | 100.0 - 398.7 |
| 3 | Channel Fill Sandstone | 0.04 - 0.15 | 0.3 | 0.18 - 0.22 | 400.01 - 889.7 |
| 4 | Distal Lower Shoreface | 0.04 - 0.19 | 0.29 | 0.1 - 0.23 | 120.5 - 170.3 |
| 5 | Proximal Shoreface | 0.05 - 0.17 | 0.31 | 0.17 - 0.24 | 80.2 - 412.5 |
| 6 | Upper Shoreface | 0.09 - 0.11 | 0.28 | 0.21 - 0.26 | 650.2 - 1023.7 |
| 7 | Distal Mouth Bar | 0.08 - 0.17 | 0.27 | 0.09 - 0.17 | 170.5 - 223.1 |
| 8 | Proximal Mouth Bar | 0.05 - 0.59 | 0.12 | 0.19 - 0.21 | 130.5 - 314.3 |
| 9 | Tidal Influenced Sandstone | 0.14 - 0.46 | 0.26 | 0.15 - 0.20 | 220.0 - 512.6 |
| 10 | Fluvial Sandstones | 0.14 - 0.29 | 0.21 | 0.19 - 0.21 | 180.5 - 691.8 |
| 11 | Coal | 0.24 - 0.53 | 0.05 | 0.001 | 0.001 |
| 12 | Coastal Plain Fines | 0.43 - 0.49 | 0.06 | 0.04 - 0.12 | 5.2 - 34.6 |
| 13 | Marine Mudstone | 0.16 - 0.42 | 0.1 | 0.08 - 0.10 | 6.0 - 15.2 |

**Table 5.** A comparison of a) porosity, and b) permeability estimates from selected intervals in the original porosity/permeability models and forward modeling-based porosity and permeability models.

| | **a. Validation Well Position 1** | | | | |
|---|---|---|---|---|---|
| | **Depth (m)** | | | | |
| | **5 m** | **10 m** | **15 m** | **25 m** | **35 m** |
| **Models** | **Measured Porosity** | | | | |
| Original Model | 0.2 | 0.25 | 0.27 | 0.16 | 0.13 |
| R14 | 0.22 | 0.24 | 0.16 | 0.22 | 0.16 |
| R20 | 0.16 | 0.19 | 0.26 | 0.18 | 0.15 |
| R26 | 0.18 | 0.17 | 0.23 | 0.16 | 0.19 |
| R36 | 0.22 | 0.21 | 0.19 | 0.22 | 0.21 |
| R45 | 0.25 | 0.2 | 0.23 | 0.22 | 0.15 |
| R49 | 0.21 | 0.17 | 0.22 | 0.17 | 0.18 |
| | **Validation Well Position 2** | | | | |
| | **Depth (m)** | | | | |
| | **5 m** | **10 m** | **15 m** | **25 m** | **35 m** |
| **Models** | **Measured Porosity** | | | | |
| Original Model | 0.17 | 0.21 | 0.21 | 0.17 | 0.19 |
| R14 | 0.17 | 0.16 | 0.24 | 0.15 | 0.25 |
| R20 | 0.21 | 0.22 | 0.2 | 0.21 | 0.23 |
| R26 | 0.21 | 0.2 | 0.21 | 0.25 | 0.24 |
| R36 | 0.2 | 0.22 | 0.21 | 0.21 | 0.19 |
| R45 | 0.22 | 0.19 | 0.2 | 0.19 | 0.21 |
| R49 | 0.26 | 0.24 | 0.23 | 0.16 | 0.21 |

| | **b. Validation Well Position 1** | | | | |
|---|---|---|---|---|---|
| | **Depth (m)** | | | | |
| | **5 m** | **10 m** | **15 m** | **25 m** | **35 m** |
| **Models** | **Measured Permeability_Z (mD)** | | | | |
| Original Model | 352.74 | 312.38 | 201.08 | 199.76 | 508.2 |
| R14 | 163.95 | 312.38 | 69.84 | 310.16 | 508.2 |
| R20 | 290.84 | 315.09 | 105.66 | 273.04 | 200.63 |
| R26 | 375.92 | 203.81 | 166.23 | 189.92 | 348.12 |
| R36 | 418.03 | 203.27 | 190.9 | 168.9 | 370.56 |
| R45 | 337.6 | 412.67 | 199.66 | 156.71 | 305.92 |
| R49 | 370.89 | 129.33 | 291.77 | 175.53 | 551.18 |
| | **Validation Well Position 2** | | | | |
| | **Depth (m)** | | | | |
| | **5 m** | **10 m** | **15 m** | **25 m** | **35 m** |
| **Models** | **Measured Permeability_Z (mD)** | | | | |
| Original Model | 6.6 | 883.6 | 30.3 | 496.99 | 156.6 |
| R14 | 320.34 | 336.22 | 151.08 | 464.22 | 132.98 |
| R20 | 122.66 | 209.15 | 161.3 | 230.58 | 208.48 |
| R26 | 151.48 | 710.07 | 175.09 | 384.49 | 169.48 |
| R36 | 184.74 | 344.99 | 157.08 | 420.15 | 136.14 |
| R45 | 91.44 | 361.04 | 77.17 | 382.85 | 134.56 |
| R49 | 134.01 | 721.73 | 137.42 | 636.48 | 290.06 |