# Peer review of "Porosity and Permeability Prediction through Forward Stratigraphic Simulations Using GPMTM and PetrelTM: Application in Shallow Marine Depositional Settings."

_Geoscientific Model Development, 2020_

## Referee Comment (RC1) · Johannes Hommel (Referee) · 24 Jul 2020

Some of my comment and questions may result from being a bit outside this field of research and not knowing the conventions in this field. While it was definitely an interesting read, it took me quite some time to read through and understand it.

General comments

The author present an approach to decrease the uncertainty in the distribution of petrophysical properties such as porosity and permeability. The general concept is using stratigraphic simulations to create realistic lithofacies distributions which then can be populated with the petrophysical properties. The general concept is explained well but in my opinion lacks a few key issues.

A big general concern I have regarding the manuscript is that all results hinge on the realistic prediction of the sediment deposition by the stratigraphic model GPM, but GPM is in this manuscript described and treated as a kind of "black box". As a reader, without knowledge of how GPM works internally, there is no way to check, know or estimate, how, why, or if the given input parameters will yield the results presented in the manuscript.

I am missing at least some basic equations or general explanations how the geological processes (Sediment Diffusion, Tectonics, Steady Flow, Unsteady Flow and Waves) act in GPM on the given input parameters. I, as someone not having worked with GPM before, have no idea and no possibility to understand how exactly the simulation results result from the little descriptions in the text and the values given in Table 2. What does GPM assume how the geological processes (Sediment Diffusion, Tectonics, Steady Flow, Unsteady Flow and Waves) act on the sediments? Another issue I have with the manuscript is that it is not made clear enough how the 20 scenarios of the stratigraphic simulations are connected with the 50 realizations. What are the realizations? How are they generated? For which scenario(s) are they applied? How and why are parameters of the scenarios and realizations chosen?

In lines 86-87 the authors state "Simulations were constrained to twenty scenarios because the desired stratigraphic sequence and associated sediment patterns were achieved at the fourth simulation." I miss the discussion or details on how the authors determine what a desired stratigraphic sequence looks like and why they continued for 16 more scenarios, when Scenario 4 was already giving the desired stratigraphic sequence?

A final general comment is that the authors claim that the presented approach reduces the uncertainty in the distribution of petrophysical properties such as porosity and permeability. In the conclusion, the authors discuss that even with their approach, uncertainty in the distribution of petrophysical properties will remain. They might increase the impact of their paper and prove their claim by comparing their resulting distribution with ones that are generated by other, more classical methods. Although I acknowledge that this might be a too big of a topic to include in this manuscript, I would find it good to at least mention the possible interest of such a comparison in the conclusion.

Specific comments: lines 6-8: Something missing in this "Typically, reservoir modeling requires property-modifying coefficients in the form values to achieve a good match to known subsurface well data."

lines 25-27: Something missing in this "but the method tends to confine reservoir property models to known data and rarely realize geological realism to capture sedimentary that have led to reservoir formation" –> sedimentary processes??

line 39: Something missing in this "The sedimentary system, Hugin formation makes up the main reservoir interval in the Volve field."

line 69-70: Something missing in this "but the thickness have not been completely penetrated (Folkestad & Satur, 2006)."

lines 86-87: "the desired stratigraphic sequence and associated sediment patterns were achieved" How did you determine this? What was the criterion for this decision? and then, why did you add another 16 scenarios, if the 4th was already showing "the desired stratigraphic sequence"? The scenarios are never discussed in detail and hidden away in Table 2 and only hinted at in some short statements e.g. lines 126-127 "To mitigate this uncertainty, 5 paleo topographic surfaces were generated stochastically" or lines 148-149 "The simulation parameters applied (Table 2) were generated randomly using the initial run"

lines 99-114: As said in the general comments, I miss some detail on how the mentioned processes are implemented in GPM, how are they parameterized etc...

line 128: TPr is not defined. I assume it is the " paleo topographic surface" or something similar from the context.

lines 133-136: " the sediment entry point for this task was placed in the north-eastern section of the hypothetical paleo-topography. Since the exact sediment entry point is uncertain, multiple entry points were placed at 4 m radius around the primary location in (Figure 3c), in order to capture possible sediment source locations." Compared to the scale shown in Figure 3c or the area given in line 221 ($\sim$18km$^2$), a 4m radius seems to me just as the same location, as the modeled area seems kilometers wide. Or do you mean 4km? Could an sediment entry point actually be as narrow as 4m within such a relatively flat looking domain as shown in Figure 3c?

lines 139-140: What was the assumed sea level after 20000 years? Only the average sea levels are given later in Table 2.

lines 148-149 and following: "The simulation parameters applied (Table 2) were generated randomly" on what basis were they created? Was the simulation always constant with no changes? I did not find any boundary conditions, so how much sediment enters the study area? Is this constant over time? The following lines e.g. "A sudden change in subsidence rate tends" suggest that the (boundary) conditions changed over time.

line 157: "Shifting the source point to the mid-section of the topography" to where exactly? can you show that in Figure 3? Isn't the sediment entry point shown in figure 3 already somewhat in the mid section, at least when looking at figure 3? And previously, you wrote that you only look into changes within a 4m radius of the sediment entry point, so I do not understand how it can have such a big influence, see lines 133-136.

lines 176-178: "shoreface lithofacies units were characterized using medium-to-coarse grained sediments to that are proximal sediment source, whiles mudstone units are

constrained to the distal parts of the stratigraphic model, where fine grained sediments accumulate at the end of the simulation." –> coarse grained sediments that are proximal to the sediment sorce? "at the end of the simulation." Do you mean at the distal end of the simulation domain or towards the end of the simulated time? Time or space is not clear from the wording.

lines 179-180: "attributes, which is" –>"attributes, which are"

line 186: x not defined.

line 187: NPHI is already defined.

lines 197-218: inconsistent numbering

line 223: "and compressed by 75.27% of cell size" the verb is missing –> "and is compressed by 75.27% of the cell size"?

line 237: What are the length measures? Well lengths, distances, ...?

line 243: "populated" –> populate + How can wells be upscaled to the original structural model? Upscaling usually refers to representing something at a larger scale, not to extrapolate from lower dimensional objects (wells are practically 1D) to higher dimensions (the 3D structural model). I am confused here, but my guess is that the 1D to 3D extrapolation is meant here with upscaling. Please clarify. After rethinking, I do not even understand the purpose of the 10 synthetic wells, why do you use them? As I understand it, you should have from the previous steps already the full 3D structural stratigraphic information, so why throw away all that, keep only 10 locations and then reconstruct again everything? Couldn't you just directly populate the stratigraphic 3D domain?

lines 249-250: "Out of fifty model realizations, six realizations that showed some similarity to the original petrophysical model are presented" How did you generate the 50 realizations exactly? How did you quantify the similarity? For which scenario did you do the 50 realizations? All 20? Only scenario 4? Could you at some point specify this,

so for what scenarios did you do the model realizations? And I assume you mean the "Porosity and Permeability model", can you confirm?

lines 277-218: Did you do any of that what you write is "typically" done?

line 291: "multiple simulation scenarios" The 20 (GPM?) simulation scenarios defined in Table 2? How do they link to the poro-perm model realizations? See comment on lines 249-250.

line 298: "A porosity-permeability model that match the original petrophysical model was produced" –> A porosity-permeability model matching the original petrophysical model was produced

line 340: "will improve property prediction away from data" Away from data sounds weird to me, what do you mean with that exactly? Extrapolation away from points (wells) where there is data (well logs)?

line 355-358: How can you guarantee that the artificial neural network approach will not have similar biases, which only are better hidden as they are less understood? How do you provide training data without cognitive or sampling biases to ensure that the artificial neural network will not train to reproduce those biases?

Tables and figures in general: The tables are not consistently formatted, Table 3 could maybe be better a sideways table as it is quite wide. Some figures contain low resolution subfigures, in which e.g. the legend is made unreadable by the low resolution, e.g. Fig 7b. The figure caption of some figures start with just "Fig.", while other figure captions start with "Figure", some are bold and some not.

Figure 2: You could maybe add the number of scenarios and realizations into the figure.

Figure 4: unreadable depth legend, how were the scenarios chosen for this figure (also figure 6)?

Figure 5c blurry screenshot, what is the exact information content?

Figure 7b: unreadable legend, caption starting with a. a. in bold

Figure 10: What are the dots representing in the plots? What is the distance? Where does the distance measure start?

Figure 11: Why are different realizations shown for porosity and permeability? Why is the direction of view different for the original model and the forward-model based realizations? This is confusing, as one has to first notice it and mentally rotate the image before one can compare original and forward-model based porosity and permeability!

Figure 11 and 12: On what basis were the realizations chosen for this comparison?

Figure 12: Mention that you compare a porosity distribution in the caption.

Table 2: why does the simulation time end for scenario 4 end before reaching 0?

Table 3: Increase font size and make a sideways table?

Table 5: Can you give the values for all realizations, maybe in an appendix or make them otherwise available?

Questions given by GMD

1. Does the paper address relevant scientific modelling questions within the scope of GMD? Does the paper present a model, advances in modelling science, or a modelling protocol that is suitable for addressing relevant scientific questions within the scope of EGU?

Seems so.

2. Does the paper present novel concepts, ideas, tools, or data?

Seems so.

3. Does the paper represent a sufficiently substantial advance in modelling science?

Seems so.

4. Are the methods and assumptions valid and clearly outlined?

Yes, but only the used models/simulators are specified. There is no discussion of the actual equations of the models/simulators (at least the GPM).

5. Are the results sufficient to support the interpretations and conclusions?

Feels a bit unclear to me.

6. Is the description sufficiently complete and precise to allow their reproduction by fellow scientists (traceability of results)? In the case of model description papers, it should in theory be possible for an independent scientist to construct a model that, while not necessarily numerically identical, will produce scientifically equivalent results. Model development papers should be similarly reproducible. For MIP and benchmarking papers, it should be possible for the protocol to be precisely reproduced for an independent model. Descriptions of numerical advances should be precisely reproducible.

Yes, but only the used models/simulators are specified. There is no discussion of the actual equations of the models/simulators (at least the GPM). The data used is made available publically and the software (versions) used are given.

7. Do the authors give proper credit to related work and clearly indicate their own new/original contribution?

Yes.

8. Does the title clearly reflect the contents of the paper? The model name and number should be included in papers that deal with only one model.

Yes.

9. Does the abstract provide a concise and complete summary?

Yes.

10. Is the overall presentation well structured and clear?

Yes.

11. Is the language fluent and precise?

It seems to me that sometimes articles (or in some occasions verbs) are missing. Otherwise, the language seems precise.

12. Are mathematical formulae, symbols, abbreviations, and units correctly defined and used?

Only few are used and most are defined. Some definitions are missing, e.g. TPr in line 128 or x in line 186.

13. Should any parts of the paper (text, formulae, figures, tables) be clarified, reduced, combined, or eliminated?

The tables are not consistently formatted, Table 3 could maybe be better a sideways table as it is quite wide. Some figures contain low resolution subfigures, in which e.g. the legend is made unreadable by the low resolution, e.g. Fig 7b. The figure caption of some figures start with just "Fig.", while other figure captions start with "Figure", some are bold and some not.

14. Are the number and quality of references appropriate?

Seems so. The number is definitely sufficient, but being unfamiliar with most references, I don't want to comment on their quality, but at least on first glance they seem to be of quality.

15. Is the amount and quality of supplementary material appropriate? For model description papers, authors are strongly encouraged to submit supplementary material containing the model code and a user manual. For development, technical, and benchmarking papers, the submission of code to perform calculations described in the text is strongly encouraged.

It is appropriate. The data and code used is made available and the software (versions)

used are given.

---

## Author Comment (AC1) · 3 Aug 2020

Thank you for this detailed comment and suggestions on the manuscript. The response will follow the style (paragraph) as presented in your Comments.

General Comments

Paragraph 1: This is rightly captured in your comment. The idea is to use this forward

stratigraphic modeling approach to improve the representation porosity and permeability properties between wells, which in most cases are widely spaced in the early stage of exploration/development.

Paragraph 2: I agree with this comment that GPM software (at least the versions I used; 2017.1 to 2019.1 versions) acts as a "black box". However, the software tries to replicate a real world sedimentary process. For example, increasing sea level, and subsidence rate in GPM corresponds with an increase in accommodation. Similarly, high land-ward elevation, and erosion increases sediment supply into the basin. This scenarios can be likened to the natural order. In this work, we focused on producing a depositional sequence that is comparable to the seismic section (see Figure 3b).

Paragraph 3: I totally agree with the first part of this comment. This suggestion will be included in the manuscript. For example, the diffusion process is governed by the equation below;

$\partial z / \partial t = k â Ĺ Ğ^2 z$, where z is topographic elevation, k the diffusion coefficient, t for time, and $â Ĺ Ğ^2 z$ the Laplacian.

Furthermore, the 20 scenarios are because of the uncertainty associated with the input parameters used to for the simulation. Different inputs were used to obtain a most representative stratigraphic framework. The 50 realizations are generated in the property modeling stage (porosity and permeability) in Petrel software, where synthetic wells data from a simulation (in this instance scenario 4) are used in a geostatistical algorithm (i.e. sequential Gaussian simulation) to generate a range of property representations. This will allow us to compare which outcome(s) match the original Volve field model from Equinor.

Paragraph 4: A desired stratigraphic pattern in this contribution is one that exhibits similarity to the depositional sequence observed in the seismic section shown in Figure 3b. Additional 16 scenarios were generated as an attempt to enhance the results in scenario 4.

Paragraph 5: Thank you for the suggestion. A related work, which compares the forward stratigraphic-based modeling approach to a classical technique (e.g. pixel based modeling) is being worked on. Notwithstanding that, this suggestion will be highlighted in the concluding part of this manuscript.

Specific Comments

Lines 6-8: This statement will be revised to read: "Typically, reservoir modeling requires property-modifying coefficients in the form values to populate properties in inter-well regions to achieve a good match to known subsurface well data in different locations. It will be a reasonable assumption that closely spaced wells will control the effect of property-modifying coefficients in subsurface modeling, but the cost of acquiring subsurface data in deeper and complex geological basins limits the volume of quality datasets that could be obtained; hence reducing our perspective of reservoir property variation and its impact on fluid behaviour".

Lines 25-27: This statement will be modified to read "but the geostatistical-based method tends to confine reservoir property models to known data and rarely realize geological realism to capture sedimentary that have led to reservoir formation".

Lines 39: The correction will be made to read "The reservoir interval under study is located within the Hugin formation, which studies by Varadi et al. (1998); Kieft et al. (2011), suggest to be a complex depositional architecture of waves, tides and riverine processes; suggesting that a single depositional model will not be adequate to produce a realisitc lithofacies distributions model."

Line 69-70: This statement will be revised into "but the total thickness of code F lithofacies is not known (Folkestad & Satur, 2006)."

Line 86-87: The main criteria for evaluating the realistic nature of a stratigraphic model was to compare it to the depositional sequence observed in the seismic section in Figure 3b, and/or interpreted through well correlation.

[Figure]

Lines 99-114: In line with my response in paragraph 3 in the general comments, we will revise the manuscript to include details on the parameterization of the various geological processes involved in the simulation.

Line 128: Yes, TPr is the paleo-topographic surface. The necessary correction will be done in the manuscript.

Line 133-136: The distance should be 4 km and not 4m as stated in the manuscript. This was an error on my part. The necessary changes will be done in the manuscript. Thank you very much.

Line 139-140: The sea level curve used in the simulation followed the Haq global sea level curve generator as well as the Exxon global sea level curve generator formats. The sea level for year 20,000 was assumed to be 45 m, and decreased to 15 m by year zero. The sea level was not kept constant as it is a curve that covered a period of geological period (see figure 1). Averages were used in the manuscript to provide an insight into the mean sea level that was in the simulation scenarios.

Line 148-149: With scenario 1 (Figure 6a) beginning to show some resemblance to the target output (i.e. the depositional pattern observed in seismic section; Figure 3b), we generated input figures that were higher and lesser than those used in generating scenario 1. Example, based on a diffusion coefficient of 8 m2/a that was used in scenario 1, diffusion coefficients +/- 5 of 8 were generated with the aim to improve the development in scenario 1. Since the initial conditions (boundary conditions) at the time of deposition are unknown, an attempt was made to apply reasonable input parameters that will produce a comparable stratigraphic pattern to observation seen in the seismic section (Figure 3b). Aside the initial topography that was kept constant in a simulation run, other input parameters such as diffusion, wave event, steady/unsteady flow, tectonics use curve functions to provide variations within the simulated period.

Line 157: When the sediment source point was shifted to the mid-section (basin-ward, close to the basin slope; see modified image below; labelled Figure 2). It is our view

that, the location of the sediment source in the simulation will have a huge impact on the resultant stratigraphic architecture.

Line 176-178: Here, we mean the distal end of the simulation domain at the end of each run in the GPM simulator. The appropriate modification will be done in the manuscript to make the point clearer.

Line 179-180: Wireline-log attributes such as gamma ray, neutron porosity, sonic, and density logs outlined in Table 1 in the supplement.

Line 186: It is actually the multiplication symbol. It will be corrected in the manuscript.

Line 197-218: (i) and (ii) where used to show that the pillar gridding process, horizon, zoning and layering processes are all part of the structural modeling process. The numbering will be modified into a 1 to 4.

Line 223: The sentence will be corrected to read "The original petrophysical model has a grid dimension of 108 m x 100 m x 63 m, and is compressed by 75.27% of cell size" in the manuscript to conform with your correction.

Line 237: This statement will be revised to read; "Ten synthetic wells (SW); ranging between a relatively low of 80 m in length for SW8 and a 120 m for SW4 were positioned in the forward model to capture the distribution of porosity-permeability at different sections of the stratigraphic model. The average distance between these wells as shown in Figure 9c is about 0.9 km apart, with a maximum and minimum of 1.3 km and 0.65 km respectively" to provide more clarity.

Line 243: The synthetic wells derived from the stratigraphic model is to provide an additional well data for use in a traditional modeling workflow as was the case in the building of original Volve model. Using the same structural model was to attain a comparable framework for evaluating the modeling outputs. Upscaling the synthetic well data is a standard procedure to "transform" the data from 1-D into a 3-D framework to build the property model.
Line 249-250: The selection of six realizations was based on visual and statistical comparison of zones in the original Volve field model, and the stratigraphic-based porosity/permeability models. The statistical approach involved the comparison of summary statistics from the original Volve model, and the model realizations generated in the Petrel software. The visual comparison on the other hand looks at how geological realistic the output is, and if it conforms with our conceptual idea of the Volve field model

Line 277-218: No we didn't do that. The explanation is that a property model that has been used for production purposes would have gone through different phases of history matching, hence its adoption as a reasonable base model. The aim is to ascertain the practicability of using the forward stratigraphic modeling technique to predict property variation in a hydrocarbon reservoir.

Line 291: The 20 simulation scenarios generated are related to the depositional models (stratigraphic models). Out of the 20 scenarios, scenario 4 was adopted and populated with porosity and permeability attributes. So out of the 20 stratigraphic modeling scenarios only scenario 4 has a direct relationship to the 50 realizations produced in the property model.

Line 298: This will be revised to read "A porosity-permeability model matching the original petrophysical model was produced using synthetic porosity and permeability logs from the forward stratigraphic model as input datasets in the sequential Gaussian simulation algorithm".

Line 340: More conditioning data (well data) in a model area enhances the chance of attaining realistic distributions in inter-well regions. So with the forward stratigraphic-based property model providing a promising depositional framework in 3-D, we will be in the position to make realistic predictions in area were well logs are widely spaced ( e.g. >= 2 km). The data used here refers to wells that have well logs. This clarification will be made in the manuscript.

Line 355-358: In our view, the calculator approach used in estimating the lithofacies

proportions in the stratigraphic model were constrained to the extent to which we assume such distributions should go. Meanwhile, with an unsupervised machine learning via neural network, we can attain many outcomes that are not restricted by our cognitive biases. The neural network can be defined with varying components (e.g. weights) to attain different outcomes, from which a best fit vertical profile that is comparable to real well log can be adopted.

Comments on Figures and Tables

Changes will be made to reflect the suggestions with regards to figures and tables.

Figure 2: This will be replaced with the modified figure below (Labelled figure 3 here).

Figure 4 & 5: A landscape format has been used in both cases in the manuscript to make the figures and legends clearer. (labelled figure 4 and 5).

Figure 10: The points in the variogram indicate the experimental data. The lag distance is about 100 m. The figure show the Lags between sample pairs for calculating the variogram in the major direction (NE-SW) of the stratigraphic model. This modification will be included to the figure caption.

Figure 11: The 6 realizations of porosity and permeability as suggested previously in Line 249-250, were to depict the similarity of the forward stratigraphic-based model to the original model in terms of porosity and permeability distribution.

Figure 12: Your suggestion has been taken on board. Caption will be revised read; "Illustrating; a. validation well 1, and b. validation well 2 samples in the synthetic forward-based model and pseudo wells from the original Volve field to compare porosity distribution".

Table 2: For example, if the simulation interval is 1 Ma, but we reach $\frac{3}{4}$ of it, and notice a similarity to the depositional sequence under study is being attained, it can be truncated, without necessarily reaching "0 years". The full extent of the simulations conducted is achieved in the folder on Zenodo.

Table 3: The font size will be increased, and a landscape format used as shown in the supplement.

Table 5: Yes, and in tables i, ii, iii, and iv in the supplement.

Please also note the supplement to this comment:
https://gmd.copernicus.org/preprints/gmd-2020-37/gmd-2020-37-AC1-supplement.pdf

| | Stratigraphic curve spreadsheet for 'Trial Sea Level Curve. ' | — □ ✕ |
|---|---|---|

| | Geological age | Elevation depth m | |
|---|---|---|---|
| 0 | 0.00 | 15 | |
| 1 | 0.10 | 25 | |
| 2 | 0.15 | 38 | |
| 3 | 0.20 | 45 | |
| 4 | 0.25 | 75 | |
| 5 | 0.30 | 95 | |
| 6 | 0.35 | 92 | |
| 7 | 0.40 | 88 | |
| 8 | 0.45 | 100 | |
| 9 | 0.50 | 120 | |
| 10 | 0.55 | 92 | |
| 11 | 0.60 | 85 | |
| 12 | 0.70 | 100 | |

**Figure 1** Sea level curve used for simulation.

**Fig. 1.**

[Figure]

**Figure 2** 3-D seismic section of the study area, from which the hypothetical topographic surface was derived for the simulation. The sedimentary entry point into the basin is located in the North Eastern section, based on previous study in the model area (e.g. Kieft et al. 2011).

**Fig. 2.**

[Figure]

**Figure 3** Workflow used in this work

**Fig. 3.**

[Figure]

**Figure 4** Inferred paleo topographic surface from seismic, also illustrating different topographic surface scenarios used in the simulation.

**Fig. 4.**

[Figure]

**Figure 5** The figure 5c illustrates the diffusion curve used in the simulation of scenario. In addition figure 7 has been modified to make 7b clearer.

**Fig. 5.**

[Figure]

**Figure 7 a.** Sediment distribution patterns in the geological process modeling software. b. lithofacies classification using the property calculator tool in Petrel™

**Fig. 6.**

The diffusion equation stated may not come out as desired in the online version (preprint), So this pdf is to show the actual equation.

$\partial z/\partial t = k\nabla^2 z$, where z is topographic elevation, k the diffusion coefficient, t for time, and $\nabla^2 z$ the Laplacian.

**Fig. 7.**

**Supplement:**

**Table 3** Lithofacies classification in the forward stratigraphic model; showing the command used in the property calculator tool in Petrel$^{TM}$.

| Lithofacies Classification | | |
|---|---|---|
| **Facies Code** | **Lithofacies** | **Command Used in Petrel's Property Calculator** |
| 0 | Marine Shale | If(Sand_fine>=0.19 And Sand_fine<=0.21 Or Silt>=0.19 And Silt<=0.2 Or Clay>=0.2 And Clay<=0.21 Or Depth_of_deposition>=-82 And Depth_of_deposition<=-78) |
| 1 | Muddy Shallow Bay Fill | If( Sand_fine>=0.36 And Sand_fine<=0.38 Or Silt>=0.18 And Silt<=0.2 Or Clay>0.18 And Clay<=0.19 Or Depth_of_deposition>=-30 And Depth_of_deposition<=-20) |
| 2 | Sandy Shallow Bay Fill | If(Sand_coarse>=0.65 And Sand_coarse<=0.73 Or Sand_fine>=0.18 And Sand_fine<=0.22 Or Silt>=0.18 And Silt<=0.2 Or Clay>=0.17 And Clay<=0.18 Or Depth_of_deposition>=-3 And Depth_of_deposition<=0) |
| 3 | Channel Fill Sandstone | If( Sand_coarse>=0.5 And Sand_coarse<=0.68 Or Sand_fine>=0.23 And Sand_fine<=0.25 Or Silt>=0.17 And Silt<=0.18 Or Depth_of_deposition>=0 And Depth_of_deposition<=2) |
| 4 | Lower Shoreface Units | If( Sand_coarse>=0.19 And Sand_coarse<=0.31 Or Sand_fine>=0.19 And Sand_fine<=0.24 Or Silt>=0.4 And Silt<=0.48 Or Clay>=0.19 And Clay<=0.31 Or Depth_of_deposition>=-83 And Depth_of_deposition<=50) |
| 5 | Middle Shoreface Units | If( Sand_coarse>=0.32 And Sand_coarse<=0.53 Or Sand_fine>=0.25 And Sand_fine<=0.32 Or Silt>=0.26 And Silt<=0.32 Or Clay>=0.19 And Clay<=0.21 Or Depth_of_deposition>=-38 And Depth_of_deposition<=-12) |
| 6 | Upper Shoreface Units | If(Sand_coarse>=0.53 And Sand_coarse<=0.72 Or Sand_fine>=0.28 And Sand_fine<=0.33 Or Silt>=0.16 And Silt<=0.21 Or Depth_of_deposition>=-10 And Depth_of_deposition<=6) |
| 7 | Distal Mouth Bar Units | If( Sand_fine>=0.23 And Sand_fine<=0.27 Or Silt>=0.38 And Silt<=0.43 Or Clay>=0.19 And Clay<=0.21 Or Depth_of_deposition>=-95 And Depth_of_deposition<=-80) |
| 8 | Proximal Mouth Bar Units | If( Sand_coarse>=0.53 And Sand_coarse<=0.71 Or Sand_fine>=0.27 And Sand_fine<=0.32 Or Silt>=0.16 And Silt<=0.21 Or Clay>=0.06 And Clay<=0.07 Or Depth_of_deposition>=-30 And Depth_of_deposition<=-27) |
| 9 | Tide Influenced Sandstones | If( Sand_coarse>=0.53 And Sand_coarse<=0.71 Or Sand_fine>=0.26 And Sand_fine<=0.31 Or Silt>=0.35 And Silt<=0.41 Or Depth_of_deposition>=-5 And Depth_of_deposition<=1) |
| 10 | Fluvial Channel Sandstones | If(Sand_coarse>=0.54 And Sand_coarse<=0.56 Or Sand_fine>=0.27 And Sand_fine<=0.29 Or Silt>=0.19 And Silt<=0.21 Or Depth_of_deposition>=-2 And Depth_of_deposition<=2) |
| 11 | Coal | Estimated as background attribute |
| 12 | Coastal plain fines | If( Silt>=0.31 And Silt<=0.43 Or Clay>=0.31 And Clay<=0.35 Or Depositional_depth>=-100 And Depositional_depth<=-40) |
| 13 | Marine Mudstone | If( Sand_fine>=0.36 And Sand_fine<=0.38 Or Silt>=0.4 And Silt<=0.52 Or Clay>=0.45 And Clay<=0.78 Or Depth_of_deposition>=-105 And Depth_of_deposition<=-90) |

**Table i.** Porosity estimates for additional realizations in validation well 1 (VP1)

| Model Realization | 5 m | 10 m | 15 m | 20 m | 25 m | 30 m |
|---|---|---|---|---|---|---|
| R1 | 0.22 | 0.19 | 0.22 | 0.22 | 0.22 | 0.22 |
| R2 | 0.23 | 0.17 | 0.26 | 0.21 | 0.24 | 0.18 |
| R3 | 0.17 | 0.16 | 0.22 | 0.27 | 0.21 | 0.24 |
| R4 | 0.24 | 0.21 | 0.23 | 0.25 | 0.22 | 0.17 |
| R5 | 0.18 | 0.21 | 0.2 | 0.27 | 0.21 | 0.24 |
| R6 | 0.23 | 0.22 | 0.27 | 0.22 | 0.22 | 0.23 |
| R7 | 0.23 | 0.25 | 0.22 | 0.21 | 0.22 | 0.25 |
| R8 | 0.23 | 0.21 | 0.22 | 0.19 | 0.17 | 0.25 |
| R9 | 0.18 | 0.24 | 0.25 | 0.25 | 0.25 | 0.27 |
| R10 | 0.23 | 0.22 | 0.20 | 0.18 | 0.27 | 0.25 |
| R11 | 0.21 | 0.19 | 0.21 | 0.25 | 0.21 | 0.20 |
| R12 | 0.20 | 0.20 | 0.23 | 0.26 | 0.22 | 0.22 |
| R13 | 0.27 | 0.20 | 0.26 | 0.20 | 0.23 | 0.18 |
| R15 | 0.17 | 0.18 | 0.18 | 0.19 | 0.21 | 0.21 |
| R16 | 0.17 | 0.18 | 0.22 | 0.15 | 0.17 | 0.19 |
| R17 | 0.24 | 0.21 | 0.26 | 0.16 | 0.20 | 0.22 |
| R18 | 0.20 | 0.23 | 0.15 | 0.25 | 0.27 | 0.24 |
| R19 | 0.20 | 0.19 | 0.19 | 0.27 | 0.23 | 0.24 |
| R21 | 0.25 | 0.21 | 0.26 | 0.23 | 0.26 | 0.18 |
| R22 | 0.21 | 0.13 | 0.20 | 0.25 | 0.15 | 0.22 |
| R23 | 0.19 | 0.14 | 0.26 | 0.21 | 0.21 | 0.25 |
| R24 | 0.19 | 0.19 | 0.22 | 0.24 | 0.26 | 0.14 |
| R25 | 0.25 | 0.22 | 0.20 | 0.26 | 0.22 | 0.27 |
| R27 | 0.21 | 0.21 | 0.24 | 0.23 | 0.18 | 0.24 |
| R28 | 0.18 | 0.18 | 0.23 | 0.25 | 0.21 | 0.19 |
| R29 | 0.19 | 0.18 | 0.20 | 0.19 | 0.26 | 0.24 |
| R30 | 0.26 | 0.20 | 0.21 | 0.19 | 0.23 | 0.21 |
| R31 | 0.21 | 0.20 | 0.24 | 0.19 | 0.27 | 0.26 |
| R32 | 0.25 | 0.25 | 0.26 | 0.23 | 0.23 | 0.26 |
| R33 | 0.23 | 0.22 | 0.23 | 0.21 | 0.15 | 0.23 |
| R34 | 0.24 | 0.14 | 0.19 | 0.23 | 0.25 | 0.21 |
| R35 | 0.19 | 0.20 | 0.23 | 0.24 | 0.21 | 0.20 |
| R37 | 0.17 | 0.19 | 0.18 | 0.22 | 0.17 | 0.25 |
| R38 | 0.16 | 0.19 | 0.23 | 0.18 | 0.13 | 0.22 |
| R39 | 0.22 | 0.15 | 0.20 | 0.21 | 0.18 | 0.14 |
| R40 | 0.22 | 0.17 | 0.19 | 0.24 | 0.20 | 0.25 |
| R41 | 0.21 | 0.23 | 0.23 | 0.23 | 0.23 | 0.22 |
| R42 | 0.22 | 0.18 | 0.22 | 0.16 | 0.16 | 0.18 |
| R43 | 0.21 | 0.19 | 0.23 | 0.22 | 0.24 | 0.24 |
| R44 | 0.22 | 0.18 | 0.26 | 0.25 | 0.24 | 0.26 |
| R46 | 0.22 | 0.19 | 0.16 | 0.22 | 0.27 | 0.24 |
| R47 | 0.14 | 0.19 | 0.24 | 0.19 | 0.21 | 0.21 |
| R48 | 0.27 | 0.18 | 0.26 | 0.23 | 0.19 | 0.20 |
| R50 | 0.17 | 0.2 | 0.19 | 0.24 | 0.25 | 0.2 |

**Table ii.** Permeability estimates for additional realizations in validation well 1 (VP1)

| Model Realization | 5 m | 10 m | 15 m | 20 m | 25 m | 30 m |
|---|---|---|---|---|---|---|
| R1 | 379.31 | 260.66 | 185.88 | 185.19 | 329.09 | 328.35 |
| R2 | 384.34 | 195.97 | 59.06 | 163.29 | 200.77 | 136.18 |
| R3 | 118.83 | 174.67 | 172.82 | 178.25 | 110.80 | 241.07 |
| R4 | 187.04 | 166.46 | 104.98 | 70.26 | 204.86 | 373.47 |
| R5 | 201.90 | 175.34 | 83.27 | 66.96 | 167.18 | 103.90 |
| R6 | 264.63 | 148.19 | 104.12 | 211.01 | 115.75 | 183.53 |
| R7 | 219.98 | 138.02 | 263.07 | 62.54 | 163.68 | 275.60 |
| R8 | 248.14 | 385.98 | 237.65 | 311.84 | 359.39 | 427.61 |
| R9 | 498.38 | 232.63 | 227.79 | 76.59 | 84.16 | 116.22 |
| R10 | 398.47 | 242.10 | 56.23 | 122.51 | 112.90 | 372.70 |
| R11 | 151.35 | 318.87 | 98.52 | 127.63 | 503.26 | 141.26 |
| R12 | 352.74 | 312.38 | 61.37 | 179.67 | 310.16 | 203.55 |
| R13 | 245.65 | 125.45 | 123.16 | 155.68 | 236.63 | 272.13 |
| R15 | 183.90 | 385.07 | 134.21 | 113.91 | 276.87 | 140.77 |
| R16 | 849.34 | 102.83 | 195.27 | 222.62 | 174.72 | 298.23 |
| R17 | 122.66 | 253.21 | 122.16 | 219.00 | 241.98 | 179.78 |
| R18 | 370.54 | 166.05 | 74.44 | 116.93 | 95.25 | 90.68 |
| R19 | 261.38 | 95.90 | 233.35 | 82.87 | 230.17 | 174.61 |
| R21 | 253.12 | 154.08 | 68.50 | 134.08 | 136.13 | 212.08 |
| R22 | 347.55 | 145.00 | 137.37 | 367.13 | 220.15 | 137.84 |
| R23 | 365.95 | 172.31 | 63.81 | 203.99 | 183.25 | 176.33 |
| R24 | 195.54 | 198.98 | 147.49 | 98.15 | 177.45 | 338.20 |
| R25 | 589.85 | 96.96 | 60.41 | 162.41 | 393.90 | 215.91 |
| R27 | 183.77 | 234.90 | 180.59 | 75.82 | 276.10 | 222.50 |
| R28 | 165.27 | 157.76 | 217.18 | 63.85 | 323.63 | 295.20 |
| R29 | 310.78 | 326.10 | 103.89 | 446.21 | 215.21 | 146.43 |
| R30 | 419.11 | 209.80 | 208.55 | 81.47 | 128.24 | 215.83 |
| R31 | 393.50 | 233.34 | 128.40 | 231.34 | 170.30 | 307.12 |
| R32 | 278.48 | 149.37 | 116.93 | 59.62 | 134.66 | 166.98 |
| R33 | 756.08 | 209.10 | 61.51 | 173.62 | 62.08 | 243.39 |
| R34 | 176.57 | 157.58 | 65.83 | 65.75 | 220.10 | 216.13 |
| R35 | 418.03 | 203.27 | 103.22 | 69.08 | 168.93 | 181.13 |
| R37 | 348.58 | 260.47 | 242.77 | 40.5 | 496.53 | 299.73 |
| R38 | 187.07 | 183.05 | 113.43 | 163.43 | 94.32 | 427.80 |
| R39 | 291.87 | 118.40 | 108.90 | 236.13 | 337.48 | 161.13 |
| R40 | 185.04 | 192.87 | 169.04 | 209.17 | 459.44 | 182.28 |
| R41 | 323.40 | 272.29 | 166.90 | 99.29 | 316.77 | 343.22 |
| R42 | 337.60 | 106.86 | 206.92 | 206.12 | 51.21 | 211.49 |
| R43 | 376.63 | 396.70 | 66.75 | 141.02 | 127.55 | 258.62 |
| R44 | 161.55 | 140.87 | 68.15 | 248.66 | 162.83 | 190.65 |
| R46 | 452.95 | 160.70 | 79.39 | 141.91 | 166.07 | 113.03 |
| R47 | 127.18 | 115.43 | 168.50 | 78.86 | 102.83 | 149.16 |
| R48 | 370.88 | 129.33 | 292.19 | 333.32 | 175.53 | 354.65 |
| R50 | 245.45 | 407.80 | 167.88 | 157.98 | 327.08 | 248.71 |

**Table iii**. Porosity estimates for additional realizations in validation well 2 (VP2)

| Model Realization | 5 m | 10 m | 15 m | 20 m | 25 m | 30 m |
|---|---|---|---|---|---|---|
| R1 | 0.21 | 0.21 | 0.21 | 0.21 | 0.21 | 0.21 |
| R2 | 0.25 | 0.23 | 0.27 | 0.21 | 0.22 | 0.18 |
| R3 | 0.19 | 0.23 | 0.15 | 0.19 | 0.21 | 0.20 |
| R4 | 0.19 | 0.20 | 0.14 | 0.22 | 0.23 | 0.22 |
| R5 | 0.21 | 0.27 | 0.18 | 0.26 | 0.22 | 0.19 |
| R6 | 0.26 | 0.20 | 0.23 | 0.15 | 0.25 | 0.27 |
| R7 | 0.26 | 0.24 | 0.13 | 0.25 | 0.14 | 0.20 |
| R8 | 0.14 | 0.14 | 0.22 | 0.15 | 0.24 | 0.23 |
| R9 | 0.23 | 0.27 | 0.23 | 0.22 | 0.24 | 0.23 |
| R10 | 0.25 | 0.16 | 0.20 | 0.22 | 0.20 | 0.18 |
| R11 | 0.21 | 0.19 | 0.26 | 0.25 | 0.24 | 0.22 |
| R12 | 0.24 | 0.14 | 0.22 | 0.21 | 0.22 | 0.18 |
| R13 | 0.24 | 0.23 | 0.21 | 0.24 | 0.13 | 0.24 |
| R15 | 0.21 | 0.24 | 0.25 | 0.20 | 0.23 | 0.19 |
| R16 | 0.14 | 0.19 | 0.25 | 0.23 | 0.16 | 0.26 |
| R17 | 0.16 | 0.26 | 0.21 | 0.22 | 0.21 | 0.25 |
| R18 | 0.23 | 0.27 | 0.19 | 0.21 | 0.19 | 0.23 |
| R19 | 0.19 | 0.25 | 0.23 | 0.18 | 0.26 | 0.26 |
| R21 | 0.18 | 0.17 | 0.14 | 0.19 | 0.25 | 0.19 |
| R22 | 0.22 | 0.20 | 0.18 | 0.22 | 0.23 | 0.21 |
| R23 | 0.23 | 0.26 | 0.26 | 0.21 | 0.27 | 0.21 |
| R24 | 0.25 | 0.16 | 0.17 | 0.23 | 0.22 | 0.26 |
| R25 | 0.17 | 0.21 | 0.22 | 0.24 | 0.20 | 0.18 |
| R27 | 0.14 | 0.15 | 0.25 | 0.25 | 0.16 | 0.24 |
| R28 | 0.24 | 0.21 | 0.22 | 0.15 | 0.18 | 0.23 |
| R29 | 0.16 | 0.15 | 0.23 | 0.18 | 0.24 | 0.21 |
| R30 | 0.20 | 0.17 | 0.21 | 0.19 | 0.24 | 0.24 |
| R31 | 0.21 | 0.25 | 0.27 | 0.22 | 0.21 | 0.26 |
| R32 | 0.25 | 0.18 | 0.18 | 0.20 | 0.17 | 0.14 |
| R33 | 0.15 | 0.26 | 0.26 | 0.24 | 0.24 | 0.19 |
| R34 | 0.16 | 0.19 | 0.23 | 0.23 | 0.22 | 0.26 |
| R35 | 0.25 | 0.21 | 0.24 | 0.25 | 0.26 | 0.25 |
| R37 | 0.22 | 0.20 | 0.26 | 0.21 | 0.24 | 0.21 |
| R38 | 0.18 | 0.24 | 0.24 | 0.17 | 0.23 | 0.25 |
| R39 | 0.23 | 0.21 | 0.18 | 0.21 | 0.17 | 0.24 |
| R40 | 0.24 | 0.26 | 0.25 | 0.22 | 0.19 | 0.22 |
| R41 | 0.22 | 0.25 | 0.23 | 0.24 | 0.17 | 0.19 |
| R42 | 0.16 | 0.26 | 0.21 | 0.24 | 0.16 | 0.23 |
| R43 | 0.25 | 0.18 | 0.21 | 0.23 | 0.22 | 0.20 |
| R44 | 0.27 | 0.21 | 0.22 | 0.19 | 0.25 | 0.17 |
| R46 | 0.23 | 0.19 | 0.25 | 0.20 | 0.17 | 0.23 |
| R47 | 0.20 | 0.16 | 0.25 | 0.25 | 0.24 | 0.20 |
| R48 | 0.18 | 0.18 | 0.19 | 0.21 | 0.23 | 0.22 |
| R50 | 0.17 | 0.19 | 0.18 | 0.22 | 0.22 | 0.22 |

**Table iv.** Permeability estimates for additional realizations in validation well 2 (VP2)

| Model Realization | 5 m | 10 m | 15 m | 20 m | 25 m | 30 m |
|---|---|---|---|---|---|---|
| R1 | 249.86 | 225.22 | 246.61 | 242.39 | 247.71 | 248.00 |
| R2 | 95.20 | 285.68 | 248.40 | 129.46 | 259.32 | 225.79 |
| R3 | 210.15 | 249.43 | 188.74 | 169.56 | 153.44 | 150.78 |
| R4 | 261.12 | 166.09 | 230.64 | 141.59 | 162.11 | 425.82 |
| R5 | 789.17 | 168.73 | 258.06 | 305.51 | 141.85 | 261.91 |
| R6 | 222.61 | 236.47 | 383.56 | 333.38 | 161.19 | 614.10 |
| R7 | 376.97 | 156.81 | 277.21 | 188.69 | 170.06 | 382.66 |
| R8 | 324.07 | 231.58 | 319.35 | 117.89 | 144.11 | 326.74 |
| R9 | 314.09 | 292.84 | 325.72 | 341.66 | 182.97 | 170.39 |
| R10 | 206.30 | 199.17 | 162.96 | 221.21 | 206.78 | 214.83 |
| R11 | 498.74 | 352.02 | 111.43 | 220.14 | 204.01 | 73.44 |
| R12 | 320.34 | 364.45 | 151.08 | 382.86 | 271.56 | 237.48 |
| R13 | 310.92 | 168.93 | 101.24 | 903.17 | 294.32 | 432.77 |
| R15 | 232.79 | 255.19 | 268.45 | 85.29 | 234.97 | 252.18 |
| R16 | 282.88 | 240.28 | 197.54 | 170.48 | 126.56 | 316.02 |
| R17 | 339.86 | 168.33 | 147.77 | 309.62 | 153.24 | 172.51 |
| R18 | 314.12 | 219.46 | 183.26 | 161.30 | 228.06 | 290.04 |
| R19 | 255.03 | 144.89 | 253.03 | 253.29 | 80.39 | 331.35 |
| R21 | 230.07 | 148.77 | 135.18 | 191.74 | 224.57 | 352.67 |
| R22 | 108.40 | 353.74 | 259.81 | 128.19 | 356.51 | 476.83 |
| R23 | 314.54 | 146.52 | 294.84 | 359.42 | 154.85 | 213.29 |
| R24 | 249.64 | 93.13 | 295.11 | 162.75 | 312.53 | 382.09 |
| R25 | 65.74 | 325.92 | 210.69 | 159.77 | 274.75 | 216.15 |
| R27 | 180.03 | 98.51 | 272.17 | 343.36 | 126.86 | 276.55 |
| R28 | 605.71 | 284.30 | 475.18 | 76.64 | 183.92 | 245.18 |
| R29 | 424.41 | 479.04 | 299.31 | 357.95 | 295.87 | 323.31 |
| R30 | 404.93 | 203.29 | 194.22 | 166.63 | 369.01 | 160.38 |
| R31 | 146.73 | 457.63 | 355.88 | 310.11 | 185.30 | 208.33 |
| R32 | 314.20 | 143.63 | 177.56 | 273.10 | 154.19 | 482.85 |
| R33 | 344.27 | 347.94 | 302.71 | 168.65 | 104.95 | 112.74 |
| R34 | 274.43 | 167.98 | 159.12 | 97.46 | 130.56 | 220.56 |
| R35 | 264.95 | 238.51 | 283.41 | 159.71 | 226.63 | 470.54 |
| R37 | 229.77 | 190.45 | 146.58 | 423.58 | 172.59 | 111.28 |
| R38 | 203.36 | 403.54 | 178.82 | 110.61 | 90.45 | 281.38 |
| R39 | 432.48 | 144.75 | 462.12 | 172.01 | 226.16 | 188.94 |
| R40 | 125.30 | 197.07 | 222.82 | 262.67 | 216.51 | 267.67 |
| R41 | 339.85 | 164.82 | 155.85 | 576.46 | 296.95 | 145.37 |
| R42 | 91.44 | 155.31 | 268.61 | 134.41 | 336.38 | 101.83 |
| R43 | 145.94 | 302.80 | 366.89 | 210.76 | 241.69 | 297.30 |
| R44 | 285.47 | 101.72 | 175.48 | 199.55 | 291.47 | 229.55 |
| R46 | 125.20 | 133.55 | 194.17 | 386.73 | 146.38 | 142.73 |
| R47 | 117.57 | 337.95 | 541.82 | 143.06 | 162.88 | 458.12 |
| R48 | 396.18 | 313.38 | 137.42 | 141.91 | 636.48 | 560.63 |
| R50 | 387.84 | 173.45 | 57.70 | 385.97 | 183.59 | 318.98 |

---

## Referee Comment (RC2) · Brian Burham (Referee) · 2 Oct 2020

This review is for the manuscript entitled: "Porosity and Permeability Prediction through Forward Stratigraphic Simulations Using GPM™ and Petrel™: Application in Shallow Marine Depositional Settings" by Daniel Otoo and David Hodgetts

This manuscript describes and applies a good method to generate Forward Stratigraphic Models (FSM) to aid uncertainty reduction and complement stochastic reservoir modelling methods. The general concept presented is explained well, but there are some key issues that should be addressed. In my opinion, the manuscript requires major revisions and I have included several constructive comments below. It is my hope the authors take these onboard before final publication in Geoscientific Model Development.

General comments: There are many cases where imprecise language is used that resulted in superfluous words and unclear statements. Less is more when describing modelling methods as many readers will not be familiar with the software or methods used. I have made some suggestions in the Line comments below, but not all have been addressed in this review.

Care should be taken to re-read the manuscript carefully for grammatical errors, missing and misspelled words. Consistent English spellings should be used throughout the manuscript. Appropriate in-text citation style should be used and maintained throughout the manuscript. For example – line 88: ". . .in some studies (e.g. Delft3D-FlowTM; Rijin & Walstra, (2003); DIONISOSTM Burges et al. (2008)).

Where possible, the author should guide readers to appropriate figures. As it stands, not enough references to figures are made.

A paragraph with a detailed description of how GPM works would be beneficial to readers. This would fit well within the section title "Process Modeling in GPM". I have not used GPM, but if it follows similar principles to other FSM approaches, this should be explained. References to other FSM software is mentioned, but detail of how GPM generates the resultant models should be explained. For example, the 'steady flow process' should be explained like what has been done in Otoo & Hodgetts (2019). This should be used as guidance for the manuscript here. Including the diffusion equation as stated in the authors reply is a good step and should be integrated into this paragraph. I fully appreciate the ease of transitioning from GPM into Petrel, and is a valid reason, but more is required.

[Figure]

Statistical validation of why the number of modelling scenarios were chosen would be good to include. For this to be reproducible (which should be the aim), anyone that reads this should have a clear idea as to why 20 scenarios were chosen so this method can be repeated in other studies.

I feel the manuscript in its current form does not include enough technical detail to clearly describe GPM and the resultant models thereof. Further precise, succinct, and clear explanations are suggested to be added to this manuscript. If others will see the value in the methods discussed herein, then they will want to know how you achieved the results by giving the detail of the modelling software would be beneficial for reproducibility.

Finally, I would advise the authors to not mention any 'future studies' or further work. This manuscript should stand alone and showcase the modelling methods presented rather than putting a final statement about what they want/are going to do in the future.

Line comments:

Line 14: "where" should be used instead of "were"

Line 15: "accommodation space" is not a widely accepted term anymore, instead please use "accommodation".

Line 5: delete "can" and "these" from the statement.

Line: 6-10: These statements do not make much sense, even with the suggested revision by the author. I would suggest something like "Typically, reservoir modelling procedures require continued property modification until a satisfactory match to known subsurface data is achieved. However, acquisition of subsurface data is costly, thus prohibitive to data collection and reservoir model conditioning."

Line: 16: This is repeated throughout the manuscript, the statement 'most likely' does not fit with the assertions made in a scientific manuscript. Integration of FSM's with stochastic modelling techniques will improve reservoir characterisation because they

more accurately simulate the geology than other methods.

Line 41: I presume the author means 'tidal' processes. If so, "tidal" is a more appropriate term to use in this instance.

Line 41: Riverine is not a term used often in literature to describe fluvial processes. Please use the term 'fluvial' in stead of riverine.

Line 73: "twenty four suite of well data" should be changed to something like: "and a suite of 24 wells that comprise of.."

Line 76: The author states that a variety of geological features (grain size, sedimentary structures etc) "play a significant part of reservoir petrophysics". This is an important statement given the nature of the study, but this point should be elaborated. A sentence is all that is needed.

Line 101: The second sentence should start with "For example.."

Line 106: Remove "space" from "accommodation space" – 'space' is implied in 'accommodation'.

Line 108: This sentence should be broken into two, there isn't a need for a semicolon. In fact, I would recommend that the manuscript be carefully reviewed to exclude semicolons as they create long sentences.

Line 164: I am not sure the correct figure is cited here. Should it be Fig. 4d?

Line 164-166: This statement needs a reference. Which shallow marine depositional sequence? The one presented by Folkestad and Sature (2006)? If so, please note the appropriate reference.

Line 176-178: This sentence should be condensed. Careful attention should be paid to grammatical errors and misspellings.

Lines 180-183: This sentence needs to be condensed. High N/G zones are known to

be the best reservoir quality zones/units. Instead focus on what those zones are from the previous work and data.

Line 184: Replace "Statoil" with "Equinor".

Line 195: please change the current statement to '... extended to represent lithofacies...'

Line 197: Please use a colon to start the list

Line 197: Please use commas to aid sentence flow.

Line 203: please change the current statement to 'Typically, pillars join corresponding...' as there are more words than necessary

Line 203-205: This sentence should be condensed. For example, it's not necessary to include the nomenclature of 'corner point gridding' in this context.

Line 205: Please remove 'is' from the statement.

Line 205: What is the major direction that the cells are aligned? I.e. what is the major orientation of the faults?

Line 211: This sentence should be broken up into two statements and a clearer definition of layers is required.

Line 212: What is the cell thickness? Are they constant across the model? How were they defined to control the vertical scale?

Line 215: Colon instead of semicolon should be used.

Line 215: What is meant by 'finer' cells? Please be more precise with the scale you are referring to. Porosity and Permeability Modelling

Line 223: What is the original cell size if it was compressed by 75%? This statement is unclear.

Line 227: replace 'wells to correspond' with 'wells that correspond'.

Line 229: 'actual well' should be deleted and just what is in brackets should be used. 'known data'

Line 230: Where the petrophysical properties guided by any trend data? Facies information? How were the values populated in the model?

Line 237: Please be more explicit as to what you mean by 80m and 120m? Is this TD of the well? MD? Or are they 80m and 120m spaced apart?

Line 242: Please define SW. This I presume means Synthetic Well?

Lines 253-257: Figure 5 referenced need to have annotations which reflect the results discussed. For example, a line that indicates the MFS surface would be beneficial to readers.

Line 258: A reference is required here with this statement. Please also remove 'space' and only just 'accommodation'

Line 262: Singular 'literature' should be used.

Line 264: A word is missing here. Possibly 'Volve dataset' is meant here?

Line 265: Singular use of 'well' is suggested as a revision to the statement.

Line 269: A word is missing in between 'such model'.

Line 270: Singular use of 'validation' is suggested.

Line 272: I'm not sure what is mean by 'modal distribution'? Do the authors mean multi-modal distribution? Normal distribution? I would suggest calculated the statistical model of the original Volve porosity model and then the models of the validation wells.

Line 274-275: This statement needs to be reworded. Are the authors saying that stratigraphic inclination remains constant within the zones, or just other variogram parameters?

Line 277-281: This sentence is too long. Please break it into smaller, clearer sentences. Are the authors suggesting that the FSM is reasonable? Or are they suggesting that the permeability models should be conditioned to known subsurface data? This sentence should be a statement about the results of the model and what is suggested as uncertainties to consider when using these types of modelling methods.

Line 282: Singular use of Discussion

Line295-297: Sequence stratigraphy is a key component to lithofacies distribution characterisation, yes. This sentence should be condensed and reworded.

Line 298: Please use the plural 'matches' instead of 'match'.

Line 302: remove 'of'

Line 304: I would suggest you use 'data' instead of 'dataset'

Line 304: Please use the past tense of 'understand'.

Line 314-315: This sentence is too long. Please split into two statements for clarity.

Line 320: remote 'rather' from the sentence, this is an important statement in the manuscript.

Table comments:

Table 1: The extent of the identified facies should be clarified. For example, Facies A1 is currently stated to be less than 6 km and greater than 29km, when in fact it should be between 6 km and 29 km. ("l = 6 km to 29 km" is sufficient). This change should be made throughout the table.

Figure comments:

I would suggest changing the colourmap on several of the figures. The current colourmap (rainbow) can cause misinterpretation of the data (https://agilescientific.com/blog/2017/12/14/no-more-rainbows), and they are not

[Figure]

suitable for people who are have difficult seeing certain colours (colour blind). Please see these blog posts to why the 'jet' or 'rainbow' colour palette should not be used: https://mycarta.wordpress.com/2012/05/12/the-rainbow-is-dead-long-live-the-rainbow-part-1/. Further, please change the colours in the property models that are more colour blind friendly. The same font choice should be used for all text written in the figure. Currently there are at least two styles.

Figure 4 & 5: These figures are too small to see appropriate detail and is very important for the story of the manuscript. These should be a landscape-oriented figures . Figure 5: Annotations in 5d to guide the reader of the results would be beneficial here. A cartoon with annotations would also be good here that illustrate clinoform progradation events and the SB/MFS events.

Figure 6: There should be a legend and key associated with this figure - I don't know what the colours represent.

Figure 7: This figure is also too small and should be oriented to landscape. Please see my comment above about colourmap choice – 'jet' and 'rainbow' should be avoided if possible. Please consider revising the colourmaps.

Figure 10: The variograms are too small to read the text. I would suggest making this a full-page figure so the data can be read appropriately.

Figure 12: These histograms are too small and make the data difficult to read and interpret.

[Figure]

---

## Author Comment (AC2) · 9 Oct 2020

I appreciate the suggestions provided by the reviewer. Our responses will follow the format in which the reviewer's comments were presented.

NB: Lines will be referred to as "L" in our response. General Comments (GC)

GC1: A paragraph with a detailed description of how GPM works would be beneficial

to readers. This would fit well within the section title "Process Modeling in GPM".

Response 1: Additional information on how geological processes in GPMTM operate have been included. Below are the additions:

Steady Flow Process The steady flow process in GPM model flows that change slowly over a period; e.g. rivers at normal stage, and deltas. The steady flow process best depicts sediment transport scenarios where flow velocity and channel depth do not vary abruptly. The steady flow process settings can be specified to fit a task in the steady flow pane of the "run sedimentary simulation" dialog box in Petrel software (2017.1 version and above). To attain stability in the simulator before running the full simulation (i.e. entire depositional period), it is advisable to undertake preliminary runs to ascertain the appropriateness of the source intensity and flow behaviour. For steady flow, a boundary condition must be specified at the edges of the model. In an open flow system, negative integers (i.e. values below zero) should be assigned to the edges of the hypothetical paleo-surface to allow water to enter and leave the simulation area. Further information on the steady flow settings can be located in the GPM user manual (i.e. Guru in the Petrel software).

Unsteady Flow Process The unsteady flow process simulates flow that are periodic, and run for a limited time; example, in turbidites where velocity of flow and depth changes abruptly over time. The unsteady flow process involves fluid elements that are affected by gravity, and by friction against the hypothetical topographic surface. A previous study on the use of unsteady flow process for stratigraphic simulation is outlined in Otoo and Hodgetts, (2019).

Diffusion Process The diffusion process replicates sediment erosion from areas of higher slope (i.e. source location), and deposition to lower slope sections of the model area. Sediment dispersion is carried out through erosion and transportation processes that are driven by gravity. The diffusion process follows an assumption that sediments are transported downslope at a proportional rate to the topographic gradient; therefore

making fine grained sediments easily transportable than coarse grained sediments. The diffusion process is controlled by two parameters; (i) diffusion coefficient, which controls the strength of the diffusion, and (ii) diffusion curve that serves as a unitless multiplier in the algorithm. The mathematical equation for the diffusion geological process is: $\partial z/\partial t = k\nabla^2 z$, where z is topographic elevation, k the diffusion coefficient, t for time, and $\nabla^2 z$ the laplacian.

Sediment Accumulation This involves the deposition of sediment using an areal source location. In the GPMTM software, sediment source can be set to a point location or considered to emanate from a whole area. For example, where a lithology is interpreted to be uniformly distributed, the sediment accumulation process can replicate such depositional scenarios. The areal input rates (in mm/yr) for each sediment type must be specified in the settings. Specifying the areal rates for each sediment is important because the software is configured to use the value of the surface at each cell in the model and multiplies it by a value (i.e. value from a unitless curve) at each time step in the simulation to estimate the thickness of sediments accumulated or eroded from the cell.

GC2: Statistical validation of why the number of modelling scenarios were chosen would be good to include. For this to be reproducible (which should be the aim), anyone that reads this should have a clear idea as to why 20 scenarios were chosen so this method can be repeated in other studies.

Response 2: A major limitation in the FSM approach is that initial boundary conditions at the time of deposition, which is required for the simulation, are unknown. In our opinion, a better means to evaluate the stratigraphic scenarios selected should be the capacity of their resultant stratigraphic-based porosity and permeability property model to match known data. An initial simulation (labelled figure 6a) was undertaken to see if the outcome will mimic the depositional pattern observed in the seismic section (figure 3b). The 20 scenarios were derived by using different input parameters with Figure 6a as guide.

GC3: I would advise the authors to not mention any 'future studies' or further work. This manuscript should stand alone and showcase the modelling methods presented rather than putting a final statement about what they want/are going to do in the future.

Response 3: The suggestion will be discussed, and appropriate considered, and the necessary corrections made in the manuscript.

Line Comments (LC)

L15: The appropriate word "accommodation" will be used henceforth.

L6-10: The statement has been corrected to read "Typically, reservoir modeling tasks require continued property modification until an a appropriate match to known subsurface data is obtained. However, acquisition of subsurface datasets is costly, thus restricts data collection and subsurface modeling condition; hence reducing our perspective of reservoir property variation and its impact on fluid behaviour"

L16: The new statement now reads "Reservoir modeling techniques with the capacity to integrate forward stratigraphic simulation outputs with stochastic modeling techniques for subsurface property modeling will improve reservoir heterogeneity characterization, because they more accurately produce geological realism than the other methods (Singh et al. 2013)".

L41: The geological processes referred to as "tides" and "riverine" have been replaced with Tidal and Fluvial processes respectively.

L73: The statement has been changed into " Datasets include 3-D seismic data, and a suite of 24 wells that consist of formation pressure data, core data, and sedimentological logs"

L76: Additional statement have been added. This reads "Grain size, sediment matrix and the degree of sorting will typically drive the volume of void created, and therefore the porosity and permeability attributes".
L105-109: The statement has been changed into "Sediment deposition, and its response to post-depositional sedimentary and tectonic processes are significant in the ultimate distribution of subsurface lithofacies units. To attain stratigraphic outputs that fall within the depositional architecture interpreted from the seismic data, the input parameters were varied (see Table 2)".

L164: Figure 4d is the hypothetical topographic surface that was used to generate the "best fit" stratigraphic model in Figure 5d. So figure 5d as used in the manuscript is the appropriate figure.

L164-166: The statement has been reviewed into "This is because, when compared to depositional description in studies such as Folkestad and Sature (2006); kieft et al., (2011), it produced a stratigraphic sequence that mimics the depositional sequence in the shallow marine depositional environment under study"

L176-178: This has been corrected into "For example, shoreface lithofacies units were characterized using medium-to-coarse grained sediments, which accumulates at proximal distance to the sediment source. In contrast, mudstone units are associated to fine grained sediments that accumulate at distal section of the simulation domain".

L180-183: In line with the reviewer's comment, the following changes have been made: "The statement here has been changed to now read "In previous studies on the Sleipner Øst, and Volve field (e.g. Equinor, 2006; Kieft et al., 2011), Shoreface deposits were identified to make up the best reservoir units, whilst lagoonal deposits formed the worst reservoir units. Using this as guide, shoreface sandstone units and mudstone/shale units in the forward stratigraphic model were characterized as best and worst reservoir units respectively".

L195-205: Specific to L205, the sentence with respect to the fault directions has been modified into "The prominent orientation of faults (I-direction) within the model area trends generally in a N-S and NE-SW direction, so the "I-direction" was set to the NNE-SSW direction to align the grid cells".

L211-212: The statement has been revised into "Vertical layering on the other hand defines the thicknesses and orientation between the layers of the model. Layers in this context describes significant changes in particle size or sediment composition in a geological formation. Using the vertical layering scheme makes it possible to honour the fault framework, pillar grid and horizons that have been derived".

L215-218: The statement has been changed into: " Upscaling: involves the substation of fine grid cells with coarser grid cells. This is done to assign property values to cells in order to evaluate which discrete value suits each a selected data point. One advantage of the upscaling procedure is to make the modeling process faster".

L223-225: The statement has been revised into "The original petrophysical model has a grid dimension of 108 m x 100 m x 63 m, and is compressed by 75.27% of cell size (from an approximated original cell size 143 m x 133 m x 84 m)".

L230-233: The statement will now read, "For option 2 the best-fit forward stratigraphic model was populated with porosity, and permeability attributes using the stratigraphic orientation captured in the seismic data (i.e. NE-SW; 240‰) to control property distribution trends. Porosity and permeability were populated into the model by using the petrophysical modeling tool under property modeling process in PetrelTM".

L237-239: Modification has been made to the statement. New statement is "Ten synthetic wells (SW), ranging between 80 m and a 120 m in total depth (TD) were positioned in the forward model to capture the vertical distribution of porosity-permeability at different sections of the stratigraphic model". L253-257: As suggested, Fig. 5 has been modified to indicate the maximum flooding surfaces (MFS) in the forward stratigraphic model. Its orientation has also been changed into landscape format (Figure 5).

L258: References have been added as suggested. The new statement is "This is consistent with real-world scenario where sediment supply matchup with accommodation generated as a result of the relative constant sea level rise within a period (e.g. Muto

and Steel, 2000; Neal and Abreu, 2009)".

L262-270: The suggestions have been taken on board, and the necessary corrections will be done.

L272: According to the petrophysical evaluation report of the Volve field by Equinor, porosity in the reservoir is between 0.17-0.30. Vertical sampling in some selected models show more porosity values within this range (i.e. 0.17-0.30). This sentence is to illustrate how the FSM approach could generate outputs that are consistent with known data. I however, agree that more explained is required in the statement. The entire statement will be modified into "The vertical distribution (Figure 12 ) of porosity in selected model realizations shows a large set of porosity values that range between 0.18 – 0.24. This output is consistent to porosity figures captured in the petrophysical evaluation of the Volve field (Equinor, 2016)".

L274-275: The entire statement has been reviewed into ". In view of the limitation in making variations within a simulation run in GPMTM, the forward stratigraphic-based model was derived with an assumption that variogram parameters, stratigraphic inclination within zones are constant in each simulation run. In contrast, the original petrophysical model involve other measured attributes within the stratigraphic zone, hence the variations noted in Table 5b".

L277-281: The statement has been re-arranged into "Typically, a petrophysical model like the Sleipner Øst and Volve field model will take into account other sources of data. For example, data from a special core analysis (SCAL) will improve the reservoir petrophysics assessment. On the basis that the FSM approach did not involve these additional information from the formation, it is reasonable to suggest that the forward stratigraphic-based porosity and permeability models have been adequately conditioned to known subsurface data".

L295-297: The entire statement has been amended into "As indicated in other studies, (e.g. Allen and Posamentier, 1993; Ghandour and Haredy, 2019) sequence stratigraphy is vital in the characterization of lithofacies in shallow marine settings. Aimed at replicating stratigraphic sequence formation in 3-D, the forward stratigraphic modeling approach in GPMTM provide a good framework to analyse petrophysical property variations in a reservoir".

L314-315: The statement has been revised into "In reality, sediment deposition into a geological basin is also controlled by mechanical and geochemical processes that tend to modify a formations petrophysical attributes (Warrlich et al. 2010). Therefore, using different geological processes and initial conditions to generate depositional scenarios, will help to produce a best fits stratigraphic framework of the reservoir under study".

Table Comments The correction has been done as suggested. Please see updated table 1 in the supplement file section. Figure Comments Figure 4, 5, 7 & 12: The figures have been changed into a landscape orientation to make them more visible. Figure 6: A key has been provided to this figure.

Please also note the supplement to this comment:
https://gmd.copernicus.org/preprints/gmd-2020-37/gmd-2020-37-AC2-supplement.pdf

———————————————————

[Figure]

Figure 5.

**Fig. 1.**

[Figure]

Figure 6.

[Figure]

**Fig. 2.**

[Figure]

Figure 10.

**Fig. 3.**

---

## Author Response (AR1)

**Author's Response**

We appreciate all the comments provided by the reviewers. The responses will follow the format in which both reviewers presented their comments.

**NB:** Lines will be referred to as "L" in Author's Response (AR), General Comments as GC, and Author's Changes as AC

**Reviewer 2**

**GC1**: A paragraph with a detailed description of how GPM works would be beneficial to readers. This would fit well within the section title "Process Modeling in GPM".

**AR1**: We agree with this comment from the reviewer. Additional information on how geological processes in  $GPM^{TM}$  operate have been included in the manuscript.

**AC1:** Changes that have been made in the manuscript include:

**Steady Flow Process**

The steady flow process in GPM model flows that change slowly over a period; e.g. rivers at normal stage, and deltas. The steady flow process best depicts sediment transport scenarios where flow velocity and channel depth do not vary abruptly. The steady flow process settings can be specified to fit a task in the steady flow pane of the "run sedimentary simulation" dialog box in Petrel software (2017.1 version and above). To attain stability in the simulator before running the full simulation (i.e. entire depositional period), it is advisable to undertake preliminary runs to ascertain the appropriateness of the source intensity and flow behaviour. For steady flow, a boundary condition must be specified at the edges of the model. In an open flow system, negative integers (i.e. values below zero) should be assigned to the edges of the hypothetical paleo-surface to allow water to enter and leave the simulation area. Further

information on the steady flow settings can be located in the GPM user manual (i.e. Guru in the Petrel software).

**Unsteady Flow Process**

The unsteady flow process simulates flow that are periodic, and run for a limited time; example, in turbidites where velocity of flow and depth changes abruptly over time. The unsteady flow process involves fluid elements that are affected by gravity, and by friction against the hypothetical topographic surface. A previous study on the use of unsteady flow process for stratigraphic simulation is outlined in Otoo and Hodgetts, (2019).

**Diffusion Process**

The diffusion process replicates sediment erosion from areas of higher slope (i.e. source location), and deposition to lower slope sections of the model area. Sediment dispersion is carried out through erosion and transportation processes that are driven by gravity. The diffusion process follows an assumption that sediments are transported downslope at a proportional rate to the topographic gradient; therefore making fine grained sediments easily transportable than coarse grained sediments. The diffusion process is controlled by two parameters; (i) diffusion coefficient, which controls the strength of the diffusion, and (ii) diffusion curve that serves as a unitless multiplier in the algorithm. The mathematical equation for the diffusion geological process is:

 $\frac{\partial z}{\partial t} = k \nabla^2 z$ , where z is topographic elevation, k the diffusion coefficient, t for time, and  $\nabla^2 z$  the laplacian.

**Sediment Accumulation**

This involves the deposition of sediment using an areal source location. In the  $\text{GPM}^{\text{TM}}$  software, sediment source can be set to a point location or considered to emanate from a

whole area. For example, where a lithology is interpreted to be uniformly distributed, the sediment accumulation process can replicate such depositional scenarios. The areal input rates (in mm/yr) for each sediment type must be specified in the settings. Specifying the areal rates for each sediment is important because the software is configured to use the value of the surface at each cell in the model and multiplies it by a value (i.e. value from a unitless curve) at each time step in the simulation to estimate the thickness of sediments accumulated or eroded from the cell. Sediment accumulation can be expressed as:

**GC2:** Statistical validation of why the number of modelling scenarios were chosen would be good to include. For this to be reproducible (which should be the aim), anyone that reads this should have a clear idea as to why 20 scenarios were chosen so this method can be repeated in other studies.

**AR 2:** A major limitation in the FSM approach is that initial boundary conditions at the time of deposition, which is required for the simulation, are unknown. In our opinion, a better means to evaluate the stratigraphic scenarios selected should be the capacity of their resultant stratigraphic-based porosity and permeability property model to match known data.

**AC2**: An initial simulation (labelled figure 6a) was undertaken to see if the outcome will mimic the depositional pattern observed in the seismic section (figure 3b). The 20 scenarios were derived by using different input parameters with Figure 6a as guide.

**GC3**: I would advise the authors to not mention any 'future studies' or further work. This manuscript should stand alone and showcase the modelling methods presented rather than putting a final statement about what they want/are going to do in the future.

**AR 3:** The suggestion will be considered, and the necessary corrections made in the manuscript.

**AC 3**: Further explanations on how results were achieved in GPM and integrated into the property modeling workflow in Petrel have been included in the manuscript. In addition, the mention of future studies in the manuscript have been removed.

**Line Comments (LC)**

L14: "where" should be used instead of "were"

AR14: The appropriate word will be used in the manuscript.

**AC14**: The new statement reads "Reservoir modeling techniques with the capacity to integrate forward stratigraphic simulation outputs with stochastic modeling techniques for subsurface property modeling"

L15: "accommodation space" is not a widely accepted term anymore, instead please use "accommodation".

AR15: The appropriate word "accommodation" will be used henceforth.

AC15: "Accommodation" has been used instead of "Accommodation space" anywhere it was found in the manuscript.

L6-10: These statements do not make much sense, even with the suggested revision by the author. I would suggest something like "Typically, reservoir modelling procedures require continued property modification until a satisfactory match to known subsurface data is achieved. However, acquisition of subsurface data is costly, thus prohibitive to data collection and reservoir model conditioning."

AR 6-10: The suggestion has been accepted and incorporated into the manuscript.

AC 6-10: Typically, reservoir modeling tasks require continued property modification until an a appropriate match to known subsurface data is obtained. However, acquisition of subsurface datasets is costly, thus restricts data collection and subsurface modeling condition; hence reducing our perspective of reservoir property variation and its impact on fluid behaviour.

**L 16**: This is repeated throughout the manuscript, the statement 'most likely' does not fit with the assertions made in a scientific manuscript. Integration of FSM's with stochastic modelling techniques will improve reservoir characterisation because they more accurately simulate the geology than other methods.

**AR 16**: We agree with the reviewer's suggestion, and will make the changes in the manuscript.

**AC 16**: The correction will be made to read "Reservoir modeling techniques with the capacity to integrate forward stratigraphic simulation outputs with stochastic modeling techniques for subsurface property modeling will improve reservoir heterogeneity characterization, because they more accurately produce geological realism than the other methods (Singh et al. 2013)".

**L 41:** I presume the author means 'tidal' processes. If so, "tidal" is a more appropriate term to use in this instance.

AR 41: The appropriate changes will be made to conform with the reviewer's comment.

AC 41: complex depositional architecture of waves, tidal and fluvial processes; suggesting that a single.

**L 73**: "twenty four suite of well data" should be changed to something like: "and a suite of 24 wells that comprise of.."

AR 73: We agree with the suggestion, and will make the corrections in the manuscript.

**AC 73:** Datasets include 3-D seismic data, and a suite of 24 wells that consist of formation pressure data, core data, and sedimentological logs.

L 76: The author states that a variety of geological features (grain size, sedimentary structures etc.) "play a significant part of reservoir petrophysics". This is an important statement given the nature of the study, but this point should be elaborated. A sentence is all that is needed.

**AR 76**: Grain composition, and structure does control petrophysical attributes in a reservoir, hence some additional explanation to make the statement clearer.

**AC 76**: Grain size, sediment matrix and the degree of sorting generally controls the volume of voids created in a sedimentary formation, and therefore the porosity and permeability attributes.

L 101 & 106: The second sentence should start with "For example.." and Remove "space" from "accommodation space" – 'space' is implied in 'accommodation' respectively.

AR 101 & 106: The corrections will be made as suggested.

AC 101 & 106: As indicated previously, "Accommodation space" has been replaced with "Accommodation" in the manuscript.

L 108: This sentence should be broken into two, there isn't a need for a semicolon. In fact, I would recommend that the manuscript be carefully reviewed to exclude semicolons as they create long sentences.

AR 108: The suggestion is accepted, and the relevant changes will be done.

**AC 108**: Sediment deposition, and its response to post-depositional sedimentary and tectonic processes are significant in the ultimate distribution of subsurface lithofacies units. To attain

stratigraphic outputs that fall within the depositional architecture interpreted from the seismic data, the input parameters were varied (see Table 2).

Lv164: I am not sure the correct figure is cited here. Should it be Fig. 4d?

**AR 164:** Figure 4d is the hypothetical topographic surface that was used to generate the "best fit" stratigraphic model in Figure 5d. So figure 5d as used in the manuscript is the appropriate figure.

L164: As a result, no changes will be made to this comment.

L 164-166: This statement needs a reference. Which shallow marine depositional sequence? The one presented by Folkestad and Sature (2006)? If so, please note the appropriate reference.

AR 164-166: We are agree, and will provide the references to support this statement.

**AC 164-166**: This is because, when compared to depositional description in studies such as Folkestad and Sature (2006); kieft et al., (2011), it produced a stratigraphic sequence that mimics the depositional sequence in the shallow marine depositional environment under study.

L 176-178: This sentence should be condensed. Careful attention should be paid to grammatical errors and misspellings.

AR 176-178: The comment has been considered and corrections made.

**AC 176-178:** For example, shoreface lithofacies units were characterized using medium-tocoarse grained sediments, which accumulates at proximal distance to the sediment source. In contrast, mudstone units are associated to fine grained sediments that accumulate at distal section of the simulation domain. **L 180-183:** This sentence needs to be condensed. High N/G zones are known to be the best reservoir quality zones/units. Instead focus on what those zones are from the previous work and data.

**AR 180-183**: In line with the reviewer's comment, appropriate changes will be made in the manuscript.

**AC 180-183**: In previous studies on the Sleipner Øst, and Volve field (e.g. Equinor, 2006; Kieft et al., 2011), Shoreface deposits were identified to make up the best reservoir units, whilst lagoonal deposits formed the worst reservoir units. Using this as guide, shoreface sandstone units and mudstone/shale units in the forward stratigraphic model were characterized as best and worst reservoir units respectively.

L 184: Replace "Statoil" with "Equinor"

**AR 184**: We agree to the suggestion, and will make the corrections.

AC 184: Anywhere "Statoil" was used in the manuscript will be replaced with "Equinor".

L 195: Please change the current statement to '... extended to represent lithofacies...'

AR 195: The relevant modification will be done.

**AC 195**: The workflow (**Figure 2b**) used for subsurface property (e.g. lithofacies, and petrophysical) modeling in PetrelTM is extended to represent lithofacies, porosity, and permeability properties in the forward stratigraphic model.

**L 203**: Please change the current statement to 'Typically, pillars join corresponding...' as there are more words than necessary.

**AR 203**: We are in agreement with the reviewer's comment, and will make the appropriate changes.

**AC 203**: Typically, pillars join corresponding corners of every grid cell of the adjacent grid to form the foundation.

L 205: This sentence should be condensed. For example, it's not necessary to include the nomenclature of 'corner point gridding' in this context.

**AR 205**: We agree with the reviewer on this comment, and will be the necessary changes in the manuscript.

**AC 205:** The sentence with respect to the fault directions has been modified into "The prominent orientation of faults (I-direction) within the model area trends generally in a N-S and NE-SW direction, so the "I-direction" was set to the NNE-SSW direction to align the grid cells.

L 211: This sentence should be broken up into two statements and a clearer definition of layers is required.

AR 211: The comment has been taken on-board, and will be included in the manuscript.

**AC 211**: Vertical layering on the other hand defines the thicknesses and orientation between the layers of the model. Layers in this context describes significant changes in particle size or sediment composition in a geological formation.

**L 212**: What is the cell thickness? Are they constant across the model? How were they defined to control the vertical scale?

**AR 212**: A constant cell thickness of 1 (one) was used. This was done to attain an identical thickness as that generated from the stratigraphic simulation in all zones.

**AC 212**: Using the vertical layering scheme makes it possible to honour the fault framework, pillar grid and horizons that have been derived in the model.

**L 215:** What is meant by 'finer' cells? Please be more precise with the scale you are referring to. Porosity and Permeability Modelling.

**AR 215**: The use "finer cells" is to describe cells with smaller dimension. However, it has been modified to make it more meaningful to readers of this manuscript.

**AC 215**: The statement has been changed into: " Upscaling: involves the substation of fine grid cells with coarser grid cells. This is done to assign property values to cells in order to evaluate which discrete value suits each a selected data point. One advantage of the upscaling procedure is to make the modeling process faster.

L 223: What is the original cell size if it was compressed by 75%? This statement is unclear.

AR 223: The original cell size from our deduction is 143 m x 133 m x 84 m.

AC 223: The statement has been revised into "The original petrophysical model has a grid dimension of 108 m x 100 m x 63 m, and is compressed by 75.27% of cell size (from an approximated original cell size 143 m x 133 m x 84 m)".

L 227 & 229: replace 'wells to correspond' with 'wells that correspond' and 'actual well' should be deleted and just what is in brackets should be used. 'known data' respectively.

AR 227 & 229: The suggestion have been incorporated into the manuscript.

AC 227 & 229: Option 1 was to assign porosity and permeability values to the synthetic lithofacies wells that correspond to known facies-associations as indicated in **Table 4**. The synthetic wells with porosity and permeability data are placed in-between known data locations to guide porosity and permeability property distribution in the model.

L 230: Where the petrophysical properties guided by any trend data? Facies information? How were the values populated in the model?

**AR 230**: Yes the petrophysical properties were guided with trend map that where derived from the major orientation of the stratigraphic framework. In this instance, the orientation was in a NE-SW direction.

**AC 230**: For option 2 the best-fit forward stratigraphic model was populated with porosity, and permeability attributes using the stratigraphic orientation captured in the seismic data (i.e. NE-SW; 240°) to control property distribution trends. Porosity and permeability were populated into the model by using the petrophysical modeling tool under property modeling process in PetrelTM.

L 237: Please be more explicit as to what you mean by 80m and 120m? Is this TD of the well? MD? Or are they 80m and 120m spaced apart?

**AR 237**: 80 m and 120 m are the range of total depths (TD) of synthetic wells used in the study.

**AC 237**: Ten synthetic wells (SW), ranging between 80 m and a 120 m in total depth (TD) were positioned in the forward model to capture the vertical distribution of porosity-permeability at different sections of the stratigraphic model.

**L 253:** Figure 5 referenced need to have annotations which reflect the results discussed. For example, a line that indicates the MFS surface would be beneficial to readers.

AR 253: We agree totally with this comment, and have made some modifications to figure 5.

**AC 253**: Figure. 5 has been modified to indicate the maximum flooding surfaces (MFS) in the forward stratigraphic model. Its orientation has also been changed into landscape format (Figure 5).

L 258: A reference is required here with this statement. Please also remove 'space' and only just 'accommodation'

AR 258: We have done the necessary correction on this line to include the references.

**AC 258:** This is consistent with real-world scenario where sediment supply matchup with accommodation generated as a result of the relative constant sea level rise within a period (e.g. Muto and Steel, 2000; Neal and Abreu, 2009).

L 262-270: This contains comments that are related to spelling mistakes.

**AR 262-270:** The suggestions have been taken on board, and the necessary corrections will be done.

**AC 262-270:** The impact of the stratigraphic simulation on porosity and permeability representation in the model was evaluated by comparing its outcomes to the original porosity and permeability models of the Volve dataset using two synthetic well prefixed VP1 and VP2. The synthetic well were sampled at a 5 m intervals vertically to estimate the distribution of porosity and permeability attributes along wells. Considering that the original porosity and permeability model (**Figure 11a**) have undergone phases of history matching to enable well planning and production strategies in the Volve field, it is reasonable to assume that porosity and permeability distribution in the Volve field petrophysical model will be geologically realistic and less uncertain.

L 272: I'm not sure what is mean by 'modal distribution'? Do the authors mean multi-modal distribution? Normal distribution? I would suggest calculated the statistical model of the original Volve porosity model and then the models of the validation wells.

**AR 272:** According to the petrophysical evaluation report of the Volve field by Equinor, porosity in the reservoir is between 0.17-0.30. Vertical sampling in some selected models

show more porosity values within this range (i.e. 0.17-0.30). This sentence is to illustrate how the FSM approach could generate outputs that are consistent with known data. I however, agree that more explained is required in the statement.

AC 272: The vertical distribution (Figure 12) of porosity in selected model realizations shows a large set of porosity values that range between 0.18 - 0.24. This output is consistent to porosity figures captured in the petrophysical evaluation of the Volve field (Equinor, 2016).

L 274-275: This statement needs to be reworded. Are the authors saying that stratigraphic inclination remains constant within the zones, or just other variogram parameters?

AR 274-275: We have made some changes to the statement as suggested by the reviewer.

**AC 274-275:** In view of the limitation in making variations within a simulation run in  $GPM^{TM}$ , the forward stratigraphic-based model was derived with an assumption that variogram parameters, stratigraphic inclination within zones are constant in each simulation run. In contrast, the original petrophysical model involve other measured attributes within the stratigraphic zone, hence the variations noted in Table 5b.

L277-281: This sentence is too long. Please break it into smaller, clearer sentences. Are the authors suggesting that the FSM is reasonable? Or are they suggesting that the permeability models should be conditioned to known subsurface data? This sentence should be a statement about the results of the model and what is suggested as uncertainties to consider when using these types of modelling methods.

AR 277-281: The sentence has been condensed to make it clearer to a reader.

**AC 277-281:** Typically, a petrophysical model like the Sleipner Øst and Volve field model will take into account other sources of data. For example, data from a special core analysis

(SCAL) will improve the reservoir petrophysics assessment. On the basis that the FSM approach did not involve these additional information from the formation, it is reasonable to suggest that the forward stratigraphic-based porosity and permeability models have been adequately conditioned to known subsurface data.

L 295-297: Sequence stratigraphy is a key component to lithofacies distribution characterisation, yes. This sentence should be condensed and reworded.

AR 295-297: Suggestion has been taken on board, and the appropriate modifications made.

**AC 295-297:** Indicated in previous studies, (e.g. Allen and Posamentier, 1993; Ghandour and Haredy, 2019) sequence stratigraphy is vital in the characterization of lithofacies in shallow marine settings. Aimed at replicating stratigraphic sequence formation in 3-D, the forward stratigraphic modeling approach in GPMTM provide a good framework to analyse petrophysical property variations in a reservoir.

L 302-304: These lines refer to spelling mistakes.

AR 302-304: The corrections have been made as suggested in the comment.

**AC 302-304**: we concede that there is a possibility to overestimate and or underestimate porosity and permeability properties as observed in some sampled intervals of the validation wells. In view of this, it is our suggestion that forward stratigraphic simulation outputs should be applied as additional data to understand sediment distribution patterns, and associated vertical and horizontal petrophysical trends in the depositional environment than using its outputs as an absolute conditioning data in subsurface property modeling.

L 314-315: This sentence is too long. Please split into two statements for clarity.

AR 314-315: The sentence has been condensed as suggested by the reviewer.

**AC 314-315:** In reality, sediment deposition into a geological basin is also controlled by mechanical and geochemical processes that tend to modify a formations petrophysical attributes (Warrlich et al. 2010). Therefore, using different geological processes and initial conditions to generate depositional scenarios, will help to produce a best fits stratigraphic framework of the reservoir under study.

**Reviewer 1**

**GC1:** A big general concern I have regarding the manuscript is that all results hinge on the realistic prediction of the sediment deposition by the stratigraphic model GPM, but GPM is in this manuscript described and treated as a kind of "black box". As a reader, without knowledge of how GPM works internally, there is no way to check, know or estimate, how, why, or if the given input parameters will yield the results presented in the manuscript.

**AR 1:** We agree with the reviewer that the GPM software (at least the versions I used; 2017.1 to 2019.1 versions) acts as a "black box". However, the software tries to replicate a real world sedimentary process. For example, increasing sea level, and subsidence rate in GPM corresponds with an increase in accommodation. Similarly, high land-ward elevation, and erosion increases sediment supply into the basin.

**AC 1**: The focus of this study is to produce a depositional sequence that mimic the pattern in the seismic section in Figure 3b. Throughout the manuscript, further information will be provided in relevant sections to bring clarity on how the GPM software works.

**GC 2**: I am missing at least some basic equations or general explanations how the geological processes (Sediment Diffusion, Tectonics, Steady Flow, Unsteady Flow and Waves) act in GPM on the given input parameters. I, as someone not having worked with GPM before, have no idea and no possibility to understand how exactly the simulation results result from the little descriptions in the text and the values given in Table 2. What does GPM assume how the geological processes (Sediment Diffusion, Tectonics, Steady Flow, Unsteady Flow and Waves) act on the sediments? Another issue I have with the manuscript is that it is not made clear enough how the 20 scenarios of the stratigraphic simulations are connected with

the 50 realizations. What are the realizations? How are they generated? For which scenario(s) are they applied? How and why are parameters of the scenarios and realizations chosen?

**AR 2**: This comment is very similar to the observations of reviewer 2. In addition, 20 scenarios are because of the uncertainty associated with the input parameters used to for the simulation. Different inputs were used to obtain a most representative stratigraphic framework. The 50 realizations are generated in the property modeling stage (porosity and permeability) in Petrel software, where synthetic wells data from a simulation (in this instance scenario 4) are used in a geostatistical algorithm (i.e. sequential Gaussian simulation) to generate a range of property representations. This will allow us to compare which outcome(s) match the original Volve field model from Equinor.

**AC 2**: We totally agree with the comment and the modification in the manuscript you follow the same style as the one presented in AC 1 under reviewer 2.

**GC 3**: In lines 86-87 the authors state "Simulations were constrained to twenty scenarios because the desired stratigraphic sequence and associated sediment patterns were achieved at the fourth simulation." I miss the discussion or details on how the authors determine what a desired stratigraphic sequence looks like and why they continued for 16 more scenarios, when Scenario 4 was already giving the desired stratigraphic sequence?

**AR 3**: A desired stratigraphic pattern in this contribution is one that exhibits similarity to the depositional sequence observed in the seismic section shown in Figure 3b. Additional 16 scenarios were generated as an attempt to enhance the results in scenario 4.

**AC 3**: Further explanations will be made in relevant sections to make the statement clearer to readers of this manuscript.

**GC 4**: A final general comment is that the authors claim that the presented approach reduces the uncertainty in the distribution of petrophysical properties such as porosity and permeability. In the conclusion, the authors discuss that even with their approach, uncertainty in the distribution of petrophysical properties will remain. They might increase the impact of their paper and prove their claim by comparing their resulting distribution with ones that are generated by other, more classical methods. Although I acknowledge that this might be a too big of a topic to include in this manuscript, I would find it good to at least mention the possible interest of such a comparison in the conclusion.

**AR 4**: In a related work, which compares the forward stratigraphic-based modeling approach to a classical technique (e.g. pixel based modeling) is being worked on. Notwithstanding that, this suggestion will be highlighted in the concluding part of this manuscript.

**Line Comments**

**L 6-8:** Something missing in this "Typically, reservoir modeling requires property-modifying coefficients in the form values to achieve a good match to known subsurface well data."

**AR 6-8:** The statement is not conclusive and not clear, so the appropriate changes will be made.

**AC 6-8**: Typically, reservoir modeling require continued property modification until an a appropriate match to known subsurface data is obtained.

L 25-27: Something missing in this "but the method tends to confine reservoir property models to known data and rarely realize geological realism to capture sedimentary that have led to reservoir formation" -> sedimentary processes??

**AR 25-27**: Again, we agree the statement is not conclusive. The corrections will be done as suggested.

**AC 25-27**: This statement will be modified to read "but the geostatistical-based method tends to confine reservoir property models to known data and rarely realize geological realism to capture sedimentary that have led to reservoir formation".

L 39: Something missing in this "The sedimentary system, Hugin formation makes up the main reservoir interval in the Volve field."

**AR 39**: We have taken the suggestion on board, and modification will be made to reflect the comment.

**AC 39**: The reservoir interval under study is located within the Hugin formation, which studies by Varadi et al. (1998); Kieft et al. (2011), have shown to be a complex depositional architecture of waves, tidal and fluvial processes; suggesting that a single depositional model will not be adequate to produce a realistic lithofacies distributions model.

**L 69-70**: Something missing in this "but the thickness have not been completely penetrated (Folkestad & Satur, 2006).

AR 69-70: The relevant changes will be made in the manuscript.

**AC 69-70**: This statement will be revised into "but the total thickness of code F lithofacies is not known (Folkestad & Satur, 2006)."

**L 86-87**: The desired stratigraphic sequence and associated sediment patterns were achieved" How did you determine this? What was the criterion for this decision? and then, why did you add another 16 scenarios, if the 4th was already showing "the desired stratigraphic sequence"? The scenarios are never discussed in detail and hidden away in Table 2 and only hinted at in some short statements e.g. lines 126-127 "To mitigate this uncertainty, 5 paleo topographic surfaces were generated stochastically" or lines 148-149 "The simulation parameters applied (Table 2) were generated randomly using the initial run.

**AR 86-87**: The main criteria for evaluating the realistic nature of a stratigraphic model was to compare it to the depositional sequence observed in the seismic section in Figure 3b, and/or interpreted through well correlation.

AC 86-87: The changes made in these lines was to include the Author's response (AR) above.

L 99-114: As said in the general comments, I miss some detail on how the mentioned processes are implemented in GPM, how are they parameterized etc...

**AR 99-114**: This is related to GC2, so the explanation provided will suffice for this comment. **AC 99-114**: The changes that have been made include details on the parameterization of the geological processes used in simulation.

L 128: TPr is not defined. I assume it is the "paleo topographic surface" or something similar from the context.

AR 128: Yes, TPr is the paleo-topographic surface of the model area.

**AC 128**: To mitigate this uncertainty, 5 paleo topographic surfaces (TPr) were generated by adding or subtracting elevations from the inferred paleo topographic surface or base topography (see Figure 4g).

L 133-136: The sediment entry point for this task was placed in the north-eastern section of the hypothetical paleo-topography. Since the exact sediment entry point is uncertain, multiple entry points were placed at 4 m radius around the primary location in (Figure 3c), in order to capture possible sediment source locations." Compared to the scale shown in Figure 3c or the area given in line 221 ( $\sim$ 18km2 ), a 4m radius seems to me just as the same location, as the modelled area seems kilometres wide. Or do you mean 4km? Could an sediment entry point actually be as narrow as 4m within such a relatively flat looking domain as shown in Figure 3c.

**AR 133-136**: The distance should be 4 km and not 4m as stated in the manuscript. The correction will be made in the manuscript.

**AC 133-136**: Based on regional well correlations in previous studies (e.g. Kieft et al. 2011), and seismic interpretation of the basin structure, the sediment entry point was placed in the north-eastern section of the hypothetical paleo-topography. Since the exact sediment entry point is not known, multiple entry points were placed at 4 km radius around the primary location in (Figure 3c), in order to capture possible sediment source locations. The source position is characterised by positive integers (i.e. values greater than zero) to enable fluid flow to other parts of the simulation surface.

L 139-140: What was the assumed sea level after 20000 years? Only the average sea levels are given later in Table 2

**AR 139-140**: The sea level curve used in the simulation followed the Haq global sea level curve generator as well as the Exxon global sea level curve generator formats. The sea level for year 20,000 was assumed to be 45 m, and decreased to 15 m by year zero. The sea level was not kept constant as it is a curve that covered a period of geological period (see figure 1). Averages were used in the manuscript to provide an insight into the mean sea level that was in the simulation scenarios.

**AC 139-140**: To attain stability in the simulator, we assumed a sea level that range between 15 m to 45 m; averaging 30 m for short simulation runs, e.g. 5000 to 20000 years. The sea level was varied with increasing duration of the simulation (illustrated in Table 2). The peak sea-level in the simulation represents the maximum flooding surface (Figure 5d), and therefore the inferred sequence boundary in the geological process model.

L 148-149: and following: "The simulation parameters applied (Table 2) were generated randomly" on what basis were they created? Was the simulation always constant with no changes? I did not find any boundary conditions, so how much sediment enters the study area? Is this constant over time? The following lines e.g. "A sudden change in subsidence rate tends" suggest that the (boundary) conditions changed over time.

**AR 148-149**: With scenario 1 (Figure 6a) beginning to show resemblance to the target output (i.e. the depositional pattern observed in seismic section; Figure 3b), we generated input figures that were higher and lesser than those used in generating scenario 1. Example, based on a diffusion coefficient of 8 m2/a that was used in scenario 1, diffusion coefficients  $\pm$  5 of 8 were generated with the aim to improve the development in scenario 1. Since the initial

conditions (boundary conditions) at the time of deposition are unknown, an attempt was made to apply input parameters that will produce a comparable stratigraphic pattern to what is observed in the seismic section (Figure 3b). Aside the initial topography that was kept constant in a simulation run, other input parameters such as diffusion, wave event, steady/unsteady flow, tectonics use curve functions to provide variations within the simulated period.

AC 148-149: Modifications made in the manuscript is the same as the author's response (AR) above.

L 157: "Shifting the source point to the mid-section of the topography" to where exactly? can you show that in Figure 3? Isn't the sediment entry point shown in figure 3 already somewhat in the mid-section, at least when looking at figure 3? And previously, you wrote that you only look into changes within a 4m radius of the sediment entry point, so I do not understand how it can have such a big influence, see lines 133-136.

**AR 157**: The "mid-point" used in the manuscript is the middle section of the entire topography (i.e. basin-ward, close to the basin slope; see modified image below; labelled Figure 2). The point was made to show that the location of the sediment source in the simulation will have a huge impact on the resultant stratigraphic architecture.

**AC 157**: Shifting the source point to the mid-section of the topography (i.e. the mid-point of the topography in a basin-ward direction) resulted in the accumulation of distal elements that are identical to turbidite lobe systems.

L 176-178: "shoreface lithofacies units were characterized using medium-to-coarse grained sediments to that are proximal sediment source, whiles mudstone units are constrained to the distal parts of the stratigraphic model, where fine grained sediments accumulate at the end of

the simulation." -> coarse grained sediments that are proximal to the sediment source? "at the end of the simulation." Do you mean at the distal end of the simulation domain or towards the end of the simulated time? Time or space is not clear from the wording.

**AR 176-178**: Here, we mean the distal end of the simulation domain at the end of each run in the GPM simulator. The appropriate modification will be done in the manuscript to make the point clearer.

**AC 176-178**: For example, shoreface lithofacies units were characterized using medium-tocoarse grained sediments, which accumulate at proximal distance to the sediment source. In contrast, mudstone units were restricted to fine grained sediments that accumulate at distal section of the simulation domain.

L 179-180: "attributes, which is" ->"attributes, which are"

**AR 179-180**: These are Wireline-log attributes such as gamma ray, neutron porosity, sonic, and density logs outlined in Table 1 in the supplement.

**AC 179-180**: Using published studies by Kieft et al., (2011), porosity and permeability variations in the stratigraphic model were estimated from wireline-log attributes such as gamma ray, neutron, sonic, and density logs outlined in Table 1.

L 186: x not defined.

**AR 186**: "x" as used here is the multiplication symbol.

AC 186:  $\emptyset_{er} = \emptyset_D + \alpha$ . (*NPHI* -  $\emptyset_D$ ) +  $\beta$ ; where  $\emptyset_{er}$  is the estimated porosity range,  $\emptyset_D$  is density porosity,  $\alpha$  and  $\beta$  are regression constants; ranging between -0.02 - 0.01 and 0.28 - 0.4 respectively, *NPHI* is neutron porosity.

Line 197-218: inconsistent numbering

**AR 197-218**: (i) and (ii) where used to show that the pillar gridding process, horizon, zoning and layering processes are all part of the structural modeling process. The numbering will be modified into a 1 to 4.

**AC 197-218:**

- 1. Structure modelling: identified faults within the study area are modelled together with interpreted surfaces from seismic and well data to generate the main structural framework, within which the entire property model will be built. The procedures involve modification of fault pillars and connecting fault bodies to one another to attain the kind of fault framework interpreted from seismic and core data.
- (2) Pillar gridding: a "grid skeleton" that is made up of a top, middle and base architectures. Typically, pillars join corresponding corners of every grid cell of the adjacent grid to form the foundation for each cell within the model. The prominent orientation of faults (i.e. I-direction) within the model area generally trends in a N-S and NE-SW direction, so the "I-direction" was set to the NNE-SSW direction to capture the structural description.

L 223: "and compressed by 75.27% of cell size" the verb is missing -> "and is compressed by 75.27% of the cell size"?

**AR 223**: The sentence shall be corrected.

**AC 223**: The original petrophysical model has a grid dimension of 108 m x 100 m x 63 m, and is compressed by 75.27% of cell size (from an approximated cell size 143 m x 133 m x 84 m). To achieve a comparable model resolution as the original porosity and permeability

model, the forward stratigraphic output with initial resolution of 90 m x 78 m x 45 m was upscaled to a cell size of 107 m x 99 m x 63 m.

L 237: What are the length measures? Well lengths, distances, ...?

AR 237: This statement will be corrected to make its meaning clearer.

**AC 237**: Ten synthetic wells (SW), ranging between 80 m and a 120 m in total depth (TD) were positioned in the forward model to capture the vertical distribution of porositypermeability at different sections of the stratigraphic model. The average distance between these wells as shown in Figure 9c is about 0.9 km apart, with a maximum and minimum of 1.3 km and 0.65 km respectively.

**L 243**: "populated" –> populate + How can wells be upscaled to the original structural model? Upscaling usually refers to representing something at a larger scale, not to extrapolate from lower dimensional objects (wells are practically 1D) to higher dimensions (the 3D structural model). I am confused here, but my guess is that the 1D to 3D extrapolation is meant here with upscaling. Please clarify. After rethinking, I do not even understand the purpose of the 10 synthetic wells, why do you use them? As I understand it, you should have from the previous steps already the full 3D structural stratigraphic information, so why throw away all that, keep only 10 locations and then reconstruct again everything? Couldn't you just directly populate the stratigraphic 3D domain?

**AR 243**: The synthetic wells derived from the stratigraphic model is to provide an additional well data for use in a traditional modeling workflow as was the case in the building of

original Volve model. Using the same structural model was to attain a comparable framework for evaluating the modeling outputs. Upscaling the synthetic well data is a standard procedure to "transform" the data from 1-D into a 3-D framework to build the property model.

**AC 243:** The variogram model (**Figure 10**), of dominant lithofacies units in the formation served as a guide in the estimation of variogram parameters from the forward model.

L 249-250: "Out of fifty model realizations, six realizations that showed some similarity to the original petrophysical model are presented" How did you generate the 50 realizations exactly? How did you quantify the similarity? For which scenario did you do the 50 realizations? All 20? Only scenario 4? Could you at some point specify this, so for what scenarios did you do the model realizations? And I assume you mean the "Porosity and Permeability model", can you confirm?

**AR 249-250**: The selection of six realizations was based on visual and statistical comparison of zones in the original Volve field model, and the stratigraphic-based porosity/permeability models. The statistical approach involved the comparison of summary statistics from the original Volve model, and the model realizations generated in the Petrel software. The visual comparison on the other hand looks at how geological realistic the output is, and if it conforms with our conceptual idea of the Volve field model.

AC 249-250: Out of fifty model realizations, six realizations that showed some similarity to the original petrophysical model are presented (Figure 11). This was achieve through visual and statistical comparison of zones in the original Volve field model, and the stratigraphic-based porosity/permeability models. The statistical approach involved the comparison of

summary statistics from the original Volve model, and the model realizations generated in the Petrel software. The visual comparison on the other hand looks at how geological realistic the output is, and if it conforms with our conceptual idea of the Volve field model.

L 277-278: Did you do any of that what you write is "typically" done?

**AR 277-278**: No we didn't do that. The explanation to this is that a property model that has been used for production purposes would have gone through different phases of history matching, hence its adoption as a reasonable base model. The aim is to ascertain the practicability of using the forward stratigraphic modeling technique to predict property variation in a hydrocarbon reservoir.

**AC 277-278**: Typically, a petrophysical model like the Sleipner Øst and Volve field model will take into account other sources of data. For example, data from a special core analysis (SCAL) will improve the reservoir petrophysics assessment. Considering that the FSM approach did not involve these additional information from the formation, it is reasonable to suggest that the forward stratigraphic-based porosity and permeability models have been adequately conditioned to known subsurface data.

**L 291**: "multiple simulation scenarios" The 20 (GPM?) simulation scenarios defined in Table 2? How do they link to the poro-perm model realizations? See comment on lines 249-250.

**AR 291**: The 20 simulation scenarios generated are related to the depositional models (stratigraphic models). Out of the 20 scenarios, scenario 4 was adopted and populated with porosity and permeability attributes. So out of the 20 stratigraphic modeling scenarios only scenario 4 has a direct relationship to the 50 realizations produced in the property model.

**AC 291**: Since the initial conditions of this basin is uncertain, multiple simulation scenarios were carried out to account for the range of bathymetries that may have influenced sediment transportation to form the present day Hugin formation. The simulation produced well defined clinoforms and sequence boundaries that depict the pattern observed in the seismic data. Clinoforms in this context, are sloping depositional surfaces in a stratigraphic architecture (Patruno & Hansen, 2018).

**L 298**: "A porosity-permeability model that match the original petrophysical model was produced" -> A porosity-permeability model matching the original petrophysical model was produced.

**AR 298**: We agree to the suggestion from the reviewer, and will make corrections to that effect.

**AC 298**: A porosity-permeability model matching the original petrophysical model was produced using synthetic porosity and permeability logs from the forward stratigraphic model as input datasets in the sequential Gaussian simulation algorithm.

L 340: "will improve property prediction away from data" Away from data sounds weird to me, what do you mean with that exactly? Extrapolation away from points (wells) where there is data (well logs)?

**AR 340**: More conditioning data (well data) will enhances the chance of attaining realistic distributions in the model area. So with the forward stratigraphic-based property model providing a realistic stratigraphic framework, synthetic wells can be obtained to control property modeling of the reservoir. In addition the term "data" used in the manuscript refers to well logs.

**AC 340**: The good match obtained from validation wells in the original and stratigraphicbased petrophysical model, leads us to the suggestion that an integration of variogram parameters from well data and forward stratigraphic simulation outputs will improve property prediction in inter-well zones. This suggestion is supported by the idea that more conditioning data (well data) will increase the chance of producing realistic property distribution in the model area.

**Line 355-358**: How can you guarantee that the artificial neural network approach will not have similar biases, which only are better hidden as they are less understood? How do you provide training data without cognitive or sampling biases to ensure that the artificial neural network will not train to reproduce those biases?

**AR 355-358**: In our view, the calculator approach used in estimating the lithofacies proportions in the stratigraphic model were constrained to the extent to which we assume such distributions should go. Meanwhile, with an unsupervised machine learning via neural network, we can attain many outcomes that are not restricted by our cognitive biases. The neural network can be defined with varying components (e.g. weights) to attain different outcomes, from which a best fit vertical profile that is comparable to real well log can be adopted.

**List of Figures**

Generally, comments on figures had to do with its clarity and caption. The appropriate corrections have been done, such that the orientation of some of the figures have been changed into landscape to make them clearer.

Also, Figure 12 have been divided into two (i.e. Figure 12a, and Figure 12b), in order to make it clearer and readable.

---

## Author Response (AR2)

**Authors Response**

**Reviewer 1**

**General Comment**

The reviewer expressed concerns about the unavailability of mathematical equations to support the geological processes that were applied in the forward stratigraphic simulation. Also, the reviewer suggests that proper formatting and modification is done to Table 1, Table, and Figure 12. Finally, the reviewer called for necessary corrections done to formatting styles and spelling mistakes.

Author Response: We agree with comments from the reviewer. Additional information on how geological processes in GPMTM operate have been included in the manuscript.

Author Changes: Changes made in the manuscript include:

**Steady and Unsteady Flow Process**

Appropriate equations for fluid/sediment movement has been provided to illustrate the mathematical basis for these processes:

The simplified Navier-Stokes comprises of two key parameters that partly rely on channel geometry and flow velocity. The Navier-Stokes equation combines the continuity equation (2) and the momentum equation (3) to generate the equation on which the steady and unsteady flow processes evolve.

The continuity equation integrates the conservation of mass:

$$\frac{\partial \rho}{\partial t} + \nabla . \rho \mathbf{q} = 0 \tag{1}$$

Where  $\rho$  is fluid density, t is time, and q the flow velocity vector.

The equation that shows the changes in momentum by the fluid:

$$p.(\frac{\partial q}{\partial t} + (q, \nabla)q) = -\nabla\rho + \nabla . \mu U + \rho(g + \Omega q)$$
(2)

Where P is pressure, t is time,  $\mu$  is fluid viscosity, and U is the Navier Stokes tensor.

Keeping density ( $\rho$ ) and viscosity ( $\mu$ ) as constant, a simple flow equation is obtained:

$$\frac{\partial q}{\partial t} + (q \cdot \nabla)q = -\nabla\Phi + v\nabla^2 q + g \qquad (3)$$

Where,  $\Phi$  is the ratio of pressure to constant density (i.e. P/ $\rho$ ), and v is the kinematic viscosity (i.e.  $\mu/\rho$ )

The solution of the framework formed in (3) is completely obtained by specifying various boundary conditions that are used in the steady and or unsteady flow processes.

**Diffusion Process**

Like the unsteady/steady flow process, the guiding equation for sediment diffusion was also provided in the manuscript:

$$F_{e} = \alpha_{e} M_{e} + \alpha_{e} \Phi_{D} \frac{U_{fi} - U_{ei}}{T_{p}}$$
(4)

 $M_e$  is the resultant force of other forces with the exception of drag force,  $T_p$  stokes relation time, expressed as:  $T_p = \rho_\rho D^2/(18\rho_f V_f)$ , with  $\rho_f$  and  $V_f$  as density and viscosity of fluid respectively.  $\Phi_D$  is a coefficient that accounts for the non-linear dependence of drag force on grain slip Reynolds number ( $R_p$ ).

$$\Phi_{\rm D} = \frac{{\rm Rp}}{24} C_D$$
(5), with CD sediment grain coefficient.

With the flow component in place, the diffusion coefficient  $(D_i)$  is deduced from the Einstein equation. Using an assumption that the diffusion coefficient decreases with increasing grain size and rise in temperature, and that the coefficient f is known, the expression for  $D_i$  is:

$$\mathbf{D}_{\mathrm{i}} = \frac{K_B T}{f} \tag{6}$$

Meanwhile, f is a function of the dimension of the spherical particle involved at a particular time (t). In accounting for f, the equation for Di changes into:

$$\mathbf{D}_{\mathrm{i}} = \frac{K_B T}{6.\pi.\eta_o.r} \tag{7}$$

The rate diffusion of diffusion relative to topography in the simulator is achieved through;

$$\frac{\partial z}{\partial t} = D_i \nabla^2 \mathbf{z} \tag{8}$$

where z is topographic elevation, k the diffusion coefficient, t for time, and  $\nabla^2 z$  is the laplacian.

**Sediment Accumulation**

Based on Tetzlaff & Harbaugh (1989), sediment accumulation in the stratigraphic simulator is also supported with the following equations:

$$(\mathbf{H} - \mathbf{Z})\frac{Dl_{KS}}{Dt} = f(Q, \nabla H, \nabla Z, L, F, K_s, k(Z))$$
(9)

Where;

H is the free surface elevation to sea level, Z is the topographic elevation for sea level,  $K_s$  is the sediment type,  $l_{ks}$  is the volumetric sediment concentration of a specific type (k), L is the vector that defines sediment concentration of each type, F is the matrix of coefficients that define each sediment type, and t is the time.

Sediment accumulation relies on (i) basin geometry and tectonics (Bajpai et al. 2001) (ii) erosion and volume of sediment transported (Cheng, et al. 2018), (iii) prevailing accommodation.

Based on Cheng et al. (2018), sediment accumulation over a period (Ar) is:

$$A_r = V_{er} - V_{es} \tag{10}$$

 $V_{es}$ , is the total volume of sediments that may escapes from the basin.  $V_{er}$  is the total volume of sediments eroded into the basin.  $V_{er} = A_{er} \times R_{er} \times t$ ; where  $A_{er}$  is the average erosion area,  $R_{er}$  is the average erosion rate, and t, time.

Because source position for the sediment accumulation process is areal, the volume of sediments accumulated in a specific layer (k) in the basin; excluding porosity, is expressed as:

$$\mathbf{A}_{\mathbf{r}} = \sum_{k=1}^{n} A_{rk} \tag{11}$$

Taking into account the impact of porosity ( $\phi$ ) in this process, the equation for the sediment accumulation is:

$$A_{r} = \sum_{k=1}^{n} [(1 - \phi_{0} * e^{-c * z_{k}}) X V_{observed_{k}}$$
(12)

Where;  $V_{observedk}$  is the volume of sediment and porosity observed in a specific layer (k),  $\phi_0$  is the surface porosity, c is the porosity-depth coefficient (after Sclater & Christie, 1980), and  $Z_k$  is the average depth of the layer k.

**Tables and Figures: Changes include Table 1, Table 5, and Figure 12.**

| Code | Facies | Description                                                                               | Thickness (t); Extent (l)              | Wireline-log Attribute               | Interpretation                                     |
|------|--------|-------------------------------------------------------------------------------------------|----------------------------------------|--------------------------------------|----------------------------------------------------|
|      | A1     | Parallel-laminated mudstone                                                               |                                        | GR = 41 - 308 API                    |                                                    |
|      |        | with occasional siltstone inputs.                                                         | + = 20 - 425 cm   = 5 - 29 km          | DT = 225 - 355 µsm -1     | Pastricted marine shale                            |
|      |        | Monospecific pattern of disorder                                                          | t = 30 - 425 till 1 = 0 - 25 kill      | NPHI = 0.17 - 0.45 v/v               | Restricted marine share                            |
|      |        | bivalves parallel to bedding.                                                             |                                        | RHOB = 2280 - 2820 gcm -1 |                                                    |
|      |        | Inter-bedded claystone and very                                                           |                                        | GR = 17 - 65 API                     |                                                    |
|      | A2     | fine-grained sandstone; non-                                                              | t=10_725 cm l=9_12 km                  | DT = 189 - 268 µsm -1     | Muddy ballow bay fill                              |
|      |        | Scarecely bivalve shells oriented                                                         | t = 10 - 725 cm 1 = 8 - 13 km          | NPHI =?                              | woody narrow bay init                              |
| A    |        | parallel to bedding.                                                                      |                                        | RHOB = 2280 - 2820 gcm-1             |                                                    |
|      |        | Fine to medium grained                                                                    |                                        | GR = 18 - 46 API                     |                                                    |
|      | A3     | sandstone; moderately to well                                                             | t = 60 - 370 cm   = 1 - 8 km           | DT = 199 - 268 µsm -1     | Sandy shallow bay fill                             |
|      |        | sorted grain. Wavy bedding,                                                               |                                        | NPHI = 0.07 - 0.52 v/v               | ,,,                                                |
|      |        | cross bedding, rare wave ripples.                                                         |                                        | RHOB = 1690 - 2745 gcm-1             |                                                    |
|      | Α4     | Parallel-laminated mudstone                                                               | t = 30 - 425 cm   = 6 - 29 km          | GR = 7 - 35 API                      | Marine channel fill                                |
|      |        | Monospecific pattern of disorder                                                          |                                        | DI = 175 - 230 µsm                   | sandstone                                          |
|      |        | bivalves parallel to bedding.                                                             |                                        | RHOB = 2280 - 2820 gcm-1             | Sundstone                                          |
|      |        | Upward coarsening siltstone to                                                            | t = 30 - 480 cm   = 1 - 2 km           | GR = 18 - 80 API                     |                                                    |
|      | B1     | fine-grained; moderatley sorted                                                           |                                        | DT = 168 - 291 usm -1     | Distal lower shoreface                             |
|      |        | sandstone. Shell debris and                                                               |                                        | NPHI = 0.04 - 0.191 v/v              |                                                    |
|      |        | quartz granules.                                                                          |                                        | RHOB = 2322 - 2723 gcm-1             |                                                    |
|      | B2     | Very fine-fine grained sandstone.                                                         | t = 130 - 440 cm   = 1.7 - 12 km       | GR = 20 - 56 API                     |                                                    |
| в    |        | Moderate to well sorted; fine                                                             |                                        | DT = 179 - 277 µsm -1     | Proximal lower                                     |
| -    |        | grained carbonaceous laminae,                                                             |                                        | NPHI = 0.05 - 0.168 v/v              | shoreface                                          |
|      |        | typically low angle cross beds.                                                           |                                        | RHOB = 2314 - 2696 gcm-1             |                                                    |
|      |        | Coaesening upward, cross                                                                  |                                        | GR = 15 - 25 API                     |                                                    |
|      | B3     | laminated, fine to medium                                                                 | t = 425 - 800 cm   = 1.7 - 8 km        | DT = 250 - 275 µsm -1     | Upper shoreface                                    |
|      |        | carbonaceous fragments                                                                    |                                        | NPHI = 0.09 - 0.113 v/v              |                                                    |
|      | C1     |                                                                                           | t = 175 - 1010 cm   = 7.2 - 19.6
km | GR = 20 - 80 API                     |                                                    |
|      |        | Highly bioturbated siltstone to
very fine sandstone, with beds of
rounded granules. |                                        | DT = 230 - 260 µsm -1     | Distal mouth has                                   |
|      |        |                                                                                           |                                        | NPHI = 0.08 - 0.169 v/v              | Distal mouth bar                                   |
| c    |        |                                                                                           |                                        | RHOB = 2327 - 2521 gcm-1             |                                                    |
| _    |        | Very fine to fine grained                                                                 |                                        | GR = 12 - 58 API                     |                                                    |
|      | C2     | sandstone, low angle cross                                                                | t = 290 - 775 cm   = 1 - 5 km          | DT = 167 - 397 µsm -1     | Proximal mouth bar                                 |
|      |        | bedding.                                                                                  |                                        | PHOB = 1612 - 2705 gcm-1             |                                                    |
|      | D1     | Fining upward coarse to fine                                                              |                                        | GR = 8 - 134 API                     |                                                    |
|      |        | grained sandstone. Stacked fining                                                         | t = 740 - 820 cm l = 1 - 2 km          | DT = 235 - 335 µsm -1     | Tidal influenced fluvial
channel fill sandstone |
|      |        | upward beds with rare coarse                                                              |                                        | NPHI = 0.14 - 0.46 v/v               |                                                    |
| _    |        | grained stringers.                                                                        |                                        | RHOB = 2284 - 2570 gcm-1             |                                                    |
| U U  | D2     | Fining upward coarse to medium                                                            | t = 580 cm   = < 2 km                  | GR = 9 - 34 API                      |                                                    |
|      |        | grained sandstone.                                                                        |                                        | DT = 241 - 297 µsm -1     | fluvial channel fill                               |
|      |        | fragments. Sharp and cohessive                                                            |                                        | NPHI = 0.14 - 0.289 v/v              | sandstone                                          |
|      |        | contact at base of bed.                                                                   |                                        | RHOB = 2168 - 2447 gcm-1             |                                                    |
| F    | E1     | Coal and carbonaceous shale                                                               |                                        | GR = 8 - 56 API                      |                                                    |
|      |        | Basal contact typically parallel.                                                         | t = 30 - 520 cm   = 6 - 19.6 km        | DT = 313 - 427 µsm -1     | Coal                                               |
|      |        | although maybe undulose.                                                                  |                                        | NPHI = 0.24 - 0.529 v/v              |                                                    |
|      |        | Alternating dark grov                                                                     |                                        | кнов = 1930 - 2225 gcm-1             |                                                    |
|      |        | mudstone/claystone and                                                                    |                                        | GR = 32 - 60 API                     |                                                    |
|      | E2     | siltstone to very fine grained                                                            | t = 60 cm   = < 2 km                   | DT = 358 - 415 µsm -1     | Coastal plain fines                                |
|      |        | sandstone. Wavy to non-parallel                                                           |                                        | NPHI = 0.43 - 0.49 v/v               |                                                    |
|      |        | lamination.                                                                               |                                        | KHOB = 1994 - 2148 gcm-1             |                                                    |
|      | F      | Mudstone with rare siltstone                                                              |                                        | GR = 4 - 134 API                     |                                                    |
| F    |        | beds. Parallel lamination, soft                                                           | t = section tot completely             | DT = 187 - 450 µsm -1     | Open marine shale                                  |
|      |        | sediment deformation developed                                                            | penetrated I = 1.7 - 36.7 km           | NPHI = 0.114 - 0.618 v/v             |                                                    |
|      |        | locally on top of beds.                                                                   |                                        | KHOB = 1730 - 2925 gcm-1             |                                                    |

**Table 1**. Lithofacies-associations in the Hugin formation, Volve Field (after Kieft et al. 2011).

|                |                               | a. Validation \ | Nell Position 1  |         |        |  |  |  |
|----------------|-------------------------------|-----------------|------------------|---------|--------|--|--|--|
|                | Depth (m)                     |                 |                  |         |        |  |  |  |
|                | 5 m                           | 10 m            | 15 m             | 25 m    | 35 m   |  |  |  |
| Models         | Measured Porosity             |                 |                  |         |        |  |  |  |
| Original Model | 0.2                           | 0.25            | 0.27             | 0.16    | 0.13   |  |  |  |
| R14            | 0.22                          | 0.24            | 0.16             | 0.22    | 0.16   |  |  |  |
| R20            | 0.16                          | 0.19            | 0.26             | 0.18    | 0.15   |  |  |  |
| R26            | 0.18                          | 0.17            | 0.23             | 0.16    | 0.19   |  |  |  |
| R36            | 0.22                          | 0.21            | 0.19             | 0.22    | 0.21   |  |  |  |
| R45            | 0.25                          | 0.2             | 0.23             | 0.22    | 0.15   |  |  |  |
| R49            | 0.21                          | 0.17            | 0.22             | 0.17    | 0.18   |  |  |  |
|                |                               | Validation W    | ell Position 2   |         |        |  |  |  |
|                | Depth (m)                     |                 |                  |         |        |  |  |  |
|                | 5 m                           | 10 m            | 15 m             | 25 m    | 35 m   |  |  |  |
| Models         |                               | N               | leasured Porosit | Y       |        |  |  |  |
| Original Model | 0.17                          | 0.21            | 0.21             | 0.17    | 0.19   |  |  |  |
| R14            | 0.17                          | 0.16            | 0.24             | 0.15    | 0.25   |  |  |  |
| R20            | 0.21                          | 0.22            | 0.2              | 0.21    | 0.23   |  |  |  |
| R26            | 0.21                          | 0.2             | 0.21             | 0.25    | 0.24   |  |  |  |
| R36            | 0.2                           | 0.22            | 0.21             | 0.21    | 0.19   |  |  |  |
| R45            | 0.22                          | 0.19            | 0.2              | 0.19    | 0.21   |  |  |  |
| R49            | 0.26                          | 0.24            | 0.23             | 0.16    | 0.21   |  |  |  |
|                |                               | h Maltdaday A   | Mall Desition 1  |         |        |  |  |  |
|                | b. Validation Well Position 1 |                 |                  |         |        |  |  |  |
|                | 5                             | 10              | Depth (m)        | 25      | 25     |  |  |  |
|                | 5 m                           | 10 m            | 15 m             | 25 m    | 35 M   |  |  |  |
| Wodels         | 252.74                        | Measur          | ed Permeability  | _Z (mD) | 500.0  |  |  |  |
| Original Model | 352.74                        | 312.38          | 201.08           | 199.76  | 508.2  |  |  |  |
| R14            | 163.95                        | 312.38          | 69.84            | 310.16  | 508.2  |  |  |  |
| R20            | 290.84                        | 315.09          | 105.66           | 273.04  | 200.63 |  |  |  |
| R26            | 375.92                        | 203.81          | 166.23           | 189.92  | 348.12 |  |  |  |
| K36            | 418.03                        | 203.27          | 190.9            | 168.9   | 370.56 |  |  |  |
| R45            | 337.6                         | 412.6/          | 199.66           | 156./1  | 305.92 |  |  |  |
| R49            | 370.89                        | 129.33          | 291.77           | 175.53  | 551.18 |  |  |  |
|                |                               | Validation W    | ell Position 2   |         |        |  |  |  |
|                |                               | 1               | Depth (m)        |         | 1      |  |  |  |
|                | 5 m                           | 10 m            | 15 m             | 25 m    | 35 m   |  |  |  |
| Models         |                               | Measur          | ed Permeability  | _Z (mD) | 1      |  |  |  |
| Original Model | 6.6                           | 883.6           | 30.3             | 496.99  | 156.6  |  |  |  |
| R14            | 320.34                        | 336.22          | 151.08           | 464.22  | 132.98 |  |  |  |
| R20            | 122.66                        | 209.15          | 161.3            | 230.58  | 208.48 |  |  |  |
| R26            | 151.48                        | 710.07          | 175.09           | 384.49  | 169.48 |  |  |  |
| R36            | 184.74                        | 344.99          | 157.08           | 420.15  | 136.14 |  |  |  |
| R45            | 91.44                         | 361.04          | 77.17            | 382.85  | 134.56 |  |  |  |
| R49            | 134.01                        | 721.73          | 137.42           | 636.48  | 290.06 |  |  |  |

**Table 5.** A comparison of a) porosity, and b) permeability estimates from selected intervals in the original porosity/permeability models and forward modeling-based porosity and permeability models.

---

## Author Response (AR3)

**Response**

**Reviewer 1**

**Comment**

The reviewer wants further explanation of how geological processes used in  $\text{GPM}^{\text{TM}}$  transport/deposit sediments, and the mathematical equations that guide these processes. For example, on sediment accumulation process, the reviewer asked the following questions:

- 1. How does the accumulation process account for different sediment types?
- 2. Does the accumulation process use mass balance for each sediment type entering and leaving a cell?
- 3. What are the actual processes and equations used in GPM for the accumulation process?

Author Response: To begin, GPMTM is commercial software developed by Schlumberger to simulate clastic and carbonate sedimentation in a deep or shallow marine environment. It is made up of geological processes such as steady and unsteady flow, sediment diffusion, wave action, tectonics, and sediment accumulation that rely on physical equations and assumptions to replicate the process of sedimentation in a geological basin. A realistic realization of a stratigraphic pattern as observed in seismic or well data will provide a 3-dimensional framework to constrain subsurface property representation that conforms with the real-world trend. In clastic sedimentation, the movement of sediments relies on equations from the original SEDSIM developed in Stanford University. Sediment transportation, erosion and deposition is governed by a simplified Navier Stokes equation. It is termed "simplified" because the Navier-Stokes equation in its original form define sediment movement in a 3-

dimensional differential form, while the flow equation used in GPMTM is 2-dimensional with an arbitrary input of flow depth.

Following a review of the original Stanford SEDSIM (Harbaugh, 1993) from which the steady and unsteady flow process in the GPM software are derived, and also through personal communication with Daniel Tetzlaff (GPM software), further details of the GPM processes involved in this work is given to answer the questions asked by the reviewer. Lastly, due to software copyright constrains I do not have access to the code/ algorithms that control each step of simulation in the software. However, the general guiding equations and assumptions used in computing the movement of sediments are provided in the manuscript.

**Author Changes:** The changes made in the manuscript includes further details of how sediment movement and deposition occur under the steady, unsteady, diffusion, and especially the accumulation processes in the GPMTM software.

**Steady and Unsteady Flow Process**

Fluid/sediment movement in the steady and unsteady flow process in GPM relies distantly on the Navier-Stokes flow equation. As indicated earlier, the Navier Stokes equation deals with flow in a 3-Dimensional framework, but the simplified flow equation used in GPM defines a 2-Dimensional system with an arbitrary dimension that accounts for flow depth (Tetzlaff, D. personal communication, February 2021).

Although the steady and unsteady flow governing equations distantly rely on the Navier-Stokes equations, the steady flow is quite distinct, as it uses a finite difference numerical method for faster computation and to also depict the frequency of flow that is characteristic of flow in channel such as rivers. The finite difference method use an assumption that flow velocity is constant from channel bottom to surface. On the other hand, the unsteady flow uses the particle method from SEDSIM3 to solve the sediment concentration in flow and sediment transport capacity (Tetzlaff & Harbaugh 1989). The simplified flow equation in the GPM software attempts to solve the problem of "shallow-water free-surface flow" over an arbitrary topography surface (Tetzlaff, D. personal communication, February 2021). "Shallow water" in this context indicates the instance where only the vertically-averaged flow velocity and flow depth are applied and kept track of as a function of two horizontal coordinates.

The flow equations in the steady and unsteady processes are expressed through:

$$\frac{\partial h}{\partial t} + \nabla . hQ = 0 \tag{1}$$

Where: h is flow depth, t is time, and Q the horizontal flow velocity vector.

$$\left(\frac{\partial Q}{\partial t} = -(g\nabla)H + \frac{c_2}{\rho}\nabla^2 Q - \frac{c_2 Q/Q}{h}\right)$$
(2)

Where:  $\frac{\partial Q}{\partial t}$  is the Lagrangian derivative of flow relative to time, g is gravity, H is the water surface elevation, c2 is the fluid friction coefficient,  $\rho$  is the water density, c1 is the water friction coefficient and h is the flow depth.

The Manning's equation is applied to relate flow, slope, flow depth and hydraulic radius channels with a constant cross-section for the steady flow process. Manning's formula states:

$$V = \frac{k}{n} R_h^{2/3} S^{1/2}$$
(3)

Where: V is the flow velocity, k is the unit conversion factor, n is the Manning's coefficient which depends on channel rugosity,  $R_h$  is the hydraulic radius and S is the slope.

As mentioned earlier, the unsteady flow process uses the particle method equation, which relies on the assumption that erosion and deposition depend on the balance between the flow's transport capacity and the "effective sediment concentration". The equation for multiple-sediment transport in flow is given as follows:

$$A_{\rm em} = \sum_{k_s} \frac{l_{Ks}}{f_{1k_s}} \tag{4}$$

Where:  $A_{em}$  is the effective sediment concentration of mixture,  $l_{ks}$  is the sediment concentration of each type, and  $f_{1}$ ,  $k_{s}$  is the transportability of each sediment type.

The transport capacity of a sediment type is expressed by equations (5) and (6). Let consider

$$\mathbf{R} = (\mathbf{A} - \mathbf{A}_{\rm em})\mathbf{f}_{2,\mathbf{k}_{\rm s}} \tag{5}$$

Where  $f_{2,k_s}$  is the erosion-deposition rate coefficient for sediment type  $k_s$ . For every sediment type  $k_s$ , the formula for transporting sediment of different grain sizes is given as:

$$(H-Z)\frac{Dl_{Ks}}{Dt} = \begin{cases} R & \text{if } R > 0 \text{ and } \tau_0 \ge f_{3,k_s} \text{ and } k(x,y,z) = K_s \\ & \text{or } R < 0 \text{ and } K_s = 1 \text{ or } l_{k_s-1} = 0 \\ 0 & \text{otherwise} \end{cases}$$
(6)

Where;

H is the free surface elevation to sea level, Z is the topographic elevation for sea level,  $K_s$  is the sediment type,  $l_{ks}$ , is the volumetric sediment concentration of a specific type (k).

**Diffusion Process**

The diffusion process simulate sediment movement from a higher slope (source location) and deposition into a lower elevation of the model through gravity. Sediment diffusion runs on the assumption that sediments are transported downslope at a proportional rate to the topographic gradient, making fine-grained sediments easily transportable than coarse-grained sediments. Sediment diffusion depends on three parameters: (i) sediment grain size and turbulence in the flow, (ii) diffusion curve, which is a unitless multiplier in the algorithm and,

(iii) diffusion coefficient. The diffusion coefficient, among other variables depend on the type of sediment and "energy" of the depositional environment. In this contribution, the highest depth-dependent diffusion coefficient occurs near sea level, where the "energy" is highest over a geological time (Dashtgard et al. 2007).

In GPM, sediment diffusion is computed using:

$$\frac{\partial z}{\partial t} = D_i \nabla^2 z + S_n \tag{7}$$

where z is topographic elevation,  $D_i$  is the diffusion coefficient, t for time, and  $\nabla^2 z$  is the laplacian of z, and  $S_n$  is the sediment source term.

In other studies such as From Dade & Friend (1998); and Zhong (2011), sediment diffusion is defined through a considering that the grain size for each sediment component (coarse sand, fine sand, silt, and clay) are known. Also an assumption that these particles have a uniform diameter (D) in the flow mix. In that case, external fore (Fe), which consist of drag, lift, virtual mass, and Basset history force is given as:

$$F_{e} = \alpha_{e} M_{e} + \alpha_{e} \Phi_{D} \frac{U_{fi} - U_{ei}}{T_{p}}$$
(8)

 $M_e$  is the resultant force of other forces with the exception of drag force,  $T_p$  stokes relation time, expressed as:  $T_p = \rho_\rho D^2/(18\rho_f V_f)$ , with  $\rho_f$  and  $V_f$  as density and viscosity of fluid respectively.  $\Phi_D$  is a coefficient that accounts for the non-linear dependence of drag force on grain slip Reynolds number ( $R_p$ ).

$$\Phi_{\rm D} = \frac{{\rm Rp}}{24} C_D$$
 (9), with CD sediment grain coefficient.

With the flow component in place, the diffusion coefficient  $(D_i)$  is deduced from the Einstein equation. Using an assumption that the diffusion coefficient decreases with increasing grain size and rise in temperature, and that the coefficient f is known, the expression for  $D_i$  is:

$$\mathbf{D}_{\mathbf{i}} = \frac{K_B T}{f} \tag{10}$$

Meanwhile, f is a function of the dimension of the spherical particle involved at a particular time (t). In accounting for f, the equation for Di changes into:

$$\mathbf{D}_{\mathbf{i}} = \frac{K_B T}{6.\pi.\eta_o r} \tag{11}$$

**Sediment Accumulation**

The sediment accumulation process in GPM is designed to produce an arbitrary amount of sediment representing the artificial vertical thickness of a uniform lithology as interpreted in a well or outcrop data (Tetzlaff, D., personal communication, February 2021).

The areal input rates for each sediment type (coarse-grained, fine-grained sediments) use the value of the map (topographic surface) at each cell in the model and multiply it by a value from a unitless curve at each time step in the simulation to estimate the thickness of sediments accumulated or eroded from a cell to another. In doing so, the accumulation process accounts for the different sediments involved in the simulation.

Sediment accumulation in the GPM software requires other processes such as steady flow and diffusion to account for sediment transport (sediment entering or leaving a cell). In line with the principle of mass balance the accumulation process uses a deposition/year (mm/yr) function to artificially produce the height of sediment deposited per cell.

The equation that guide sediment accumulation in the GPM software is given as:

$$A_{T} = \sum_{s=1}^{n} \left[ (M_{\nu 1} * S_{c1}), \_\_n \right]$$
(12)

Where;

 $A_T$  is the total sediment accumulated in a cell over a period, S is the sediment type,  $M_v$  is the map value of sediment in each cell, and  $S_C$  is the sediment supply curve as a function of topographic elevation.

**Summary Answers to Specific Questions on Accumulation Process.**

1. How does the accumulation process account for different sediment types?

**Answer**: It account for different sediment types by using curves (sediment type as a function of time that is not less than zero) and sediment probability maps.

2. Does the accumulation process use mass balance for each sediment type entering and leaving a cell?

**Answer**: Yes it uses Mass balance. It applies deposition height/year (mm/yr) as input in each cell and requires other processes such as diffusion to account for sediment transport.

3. What are the actual processes and equations used in GPM for the accumulation process?

**Answer**: It is solely the sum of all sediments (map value in cell \* sediment supply curve) to produce the height of the deposited layer per cell.